# EMBO *reports*

# Local nuclear to cytoplasmic ratio regulates H3.3 incorporation via cell cycle state during zygotic genome activation

Anusha D Bhatt [ID], Madeleine G Brown, Aurora B Wackford, Yuki Shindo & Amanda A Amodeo [ID] 

## Abstract

Early embryos often have unique chromatin states prior to zygotic genome activation (ZGA). In *Drosophila*, ZGA occurs after 13 reductive nuclear divisions during which the nuclear to cytoplasmic (N/C) ratio grows exponentially. Previous work found that histone H3 chromatin incorporation decreases while its variant H3.3 increases leading up to ZGA. In other cell types, H3.3 is associated with sites of active transcription and heterochromatin, suggesting a link between H3.3 and ZGA. Here, we test what factors regulate H3.3 incorporation at ZGA. We find that H3 nuclear availability falls more rapidly than H3.3 leading up to ZGA. We generate H3/H3.3 chimeric proteins at the endogenous H3.3 A locus and observe that chaperone binding, but not gene structure, regulates H3.3 behavior. We identify the N/C ratio as a major determinant of H3.3 incorporation. To isolate how the N/C ratio regulates H3.3 incorporation we test the roles of genomic content, zygotic transcription, and cell cycle state. We determine that cell cycle regulation, but not H3 availability or transcription, controls H3.3 incorporation. Overall, we propose that local N/C ratios control histone variant usage via cell cycle state during ZGA.

**Keywords** Nuclear to Cytoplasmic Ratio; Histones; Chromatin; Transcription; Cell Cycle
**Subject Categories** Cell Cycle; Chromatin, Transcription & Genomics; Development

## Introduction

Genome accessibility can be dynamically regulated through controlled incorporation of variant histones (Khorasanizadeh, 2004; Talbert and Henikoff, 2021, 2017). In most tissues, replication-dependent (RD) histones, produced during S-phase, generate the majority of nucleosomes (Loppin and Berger, 2020; Talbert and Henikoff, 2017; Weber and Henikoff, 2014). RD histones have unusually high copy number, lack introns, and contain specialized UTRs to facilitate their rapid production during

S-phase (Dominski and Tong, 2021; Lifton et al, 1978; Marzluff et al, 2008; McKay et al, 2015; Talbert and Henikoff, 2021). Conversely, replication-independent (RI), "variant" histones are made throughout the cell cycle and incorporated into specific genomic regions (Talbert and Henikoff, 2010; Weber and Henikoff, 2014). The exchange of RD and RI histones on chromatin is a common feature of early embryonic development, especially during zygotic genome activation (ZGA)(Dimitrov et al, 1993; Johnson et al, 2018; Müller et al, 2002; Pérez-Montero et al, 2013; Shindo et al, 2022; Smith et al, 1988; Wibrand and Olsen, 2002). In most organisms, ZGA happens in multiple waves, but the chromatin undergoes extensive remodeling to facilitate bulk transcription during the major wave of ZGA (hereafter referred to as ZGA) (Blythe and Wieschaus, 2016; Hug et al, 2017; McKnight and Miller, 1977; Shermoen et al, 2010; Tadros and Lipshitz, 2009; Vastenhouw et al, 2019; Zhang et al, 2014). In *Drosophila*, these changes include refinement of nucleosomal positioning, partitioning of euchromatin and heterochromatin, and formation of topologically associated domains (Blythe and Wieschaus, 2016; Hug et al, 2017; McKnight and Miller, 1977; Seller et al, 2019).

The pre-ZGA cell cycles in many organisms depend on maternally supplied components, including histones (Adamson and Woodland, 1974; Horard and Loppin, 2015; Shindo and Amodeo, 2019; Woodland and Adamson, 1977). These cycles are unusual since they oscillate between S and M without growth phases, leading to an exponential increase in the nuclear to cytoplasmic (N/C) ratio (Blythe and Wieschaus, 2015a; Farrell and O'Farrell, 2014; Kane and Kimmel, 1993; Newport and Kirschner, 1982a, 1982b; Shindo and Amodeo, 2019). The N/C ratio, in turn, controls the timing of cell cycle slowing and ZGA (Edgar et al, 1986; Edgar and Schubiger, 1986; Kane and Kimmel, 1993; Newport and Kirschner, 1982a, 1982b; Syed et al, 2021). Titration of maternal histones against the increasing amount of DNA has been proposed to contribute to N/C ratio sensing in the early embryo (Almouzni et al, 1991, 1990; Almouzni and Wolffe, 1995; Amodeo et al, 2015; Chari et al, 2019; Joseph et al, 2017; Prioleau et al, 1994; Shindo and Amodeo, 2021). Another hallmark of ZGA is histone variant exchange on chromatin. In many organisms, maternally supplied, embryonic-specific linker histone variants are replaced by RD H1s during ZGA (Dimitrov et al, 1993; Müller et al, 2002; Pérez-Montero et al, 2013; Smith et al, 1988; Wibrand and Olsen, 2002). Concurrently, the RD nucleosomal H2A is also

Department of Biological sciences, Dartmouth College, Hanover, NH 03755, USA. ✉E-mail: amanda.amodeo@dartmouth.edu

replaced by RI H2Av as a consequence of the lengthened interphase in cycles leading up to ZGA in *Drosophila* (Johnson et al, 2018; Li et al, 2014). Similarly, we have previously shown that RD H3.2 (hereafter referred to as H3) is replaced by RI H3.3 during these same cycles, though the cause remains unclear (Shindo and Amodeo, 2019).

H3.3 is essential for proper embryonic development in mice, *Xenopus*, and zebrafish (Delaney et al, 2023; Jang et al, 2015; Klein and Knoepfler, 2023; Santenard et al, 2010; Sitbon et al, 2020). In *Xenopus*, the H3.3-specific S31 residue is required for gastrulation, while its chaperone binding site is dispensable (Sitbon et al, 2020). In *Drosophila*, H3.3 nulls survive until adulthood using maternal H3.3 but are sterile (Sakai et al, 2009). Flies expressing H3 from the H3.3 enhancer generated conflicting results as to whether H3.3 protein or simply a source of replication-independent H3 is required for fertility (Hödl and Basler, 2009; Sakai et al, 2009). Nonetheless, H3.3 is required to complete development when H3 copy number is reduced (McPherson et al, 2023). It was recently shown that the H3.3-specific chaperone, Hira, is an important regulator of chromatin accessibility and transcription during *Drosophila* ZGA (Zhang et al, 2024). This is consistent with the observation that in other contexts, H3.3 is often enriched at sites of active transcription and in heterochromatin, which are both established during ZGA (Szenker et al, 2011; Talbert and Henikoff, 2017; Weber and Henikoff, 2014).

Here, we examine the factors that contribute to H3.3 incorporation at ZGA in *Drosophila*. We identify a more rapid decrease in the nuclear availability of H3 than H3.3 over the final pre-ZGA cycles. We find that chaperone binding, not gene expression, controls incorporation patterns using H3/H3.3 chimeric proteins at the endogenous H3.3 A locus. The increase in H3.3 incorporation depends on the N/C ratio. Since the N/C ratio affects many parameters of embryogenesis, we further test the contributions of genomic content, zygotic transcription, and cell cycle states. We identify cell cycle regulation, but not H3 availability or transcription, as a major determinant of H3.3 incorporation. Overall, we propose a model in which local N/C ratios regulate chromatin composition via cell cycle state during ZGA.

## Results

### The interphase nuclear availability of H3 decreases more rapidly than H3.3 over the pre-ZGA cycles

To understand the in vivo dynamics of the H3/H3.3 pair during ZGA in *Drosophila*, we previously tagged H3 and H3.3 with a photoconvertible Dendra2 protein (H3-Dendra2 and H3.3-Dendra2) at a pseudo-endogenous H3 locus and the endogenous H3.3A locus, respectively (Fig. EV1A,B) (Shindo and Amodeo, 2019). In *Drosophila*, ZGA occurs in two waves. The minor wave starts as early as the seventh cycle, while major ZGA occurs after 13 rapid syncytial nuclear cycles (NCs) and is accompanied by cell cycle slowing and cellularization (Fig. 1A, B). During the pre-ZGA cycles (NC10-13), the maximum volume that each nucleus attains decreases in response to the doubling number of nuclei with each division (Fig. 1C). These divisions are driven by maternally provided components, and the total amount of H3-type histones

do not keep up with the pace of new DNA produced (Shindo and Amodeo, 2019). We have previously shown that with each NC, the pool of free H3 in the nucleus is depleted, and its levels on chromatin during mitosis decrease (Fig. 1D and EV1C,D) (Shindo and Amodeo, 2019). In contrast, H3.3 mitotic chromatin levels increase during the same cycles (Fig. 1D and EV1C-D) (Shindo and Amodeo, 2019). To test if changes in the relative nuclear availability of H3 and H3.3 mirror the observed chromatin incorporation trends, we measured the nuclear intensities of H3-Dendra2 and H3.3-Dendra2 in each interphase. We observed that H3 nuclear intensities decreased by ~40% between NC10 and NC13 as previously shown (Fig. 2A,B) (Shindo and Amodeo, 2019). However, when we measured H3.3-Dendra2 nuclear intensities, we found that they decreased by only ~20% between NC10 and NC13 (Fig. 2A-B). We note that these differences are not due to photobleaching, as our measurements on imaged and unimaged embryos indicate that photobleaching is negligible under our experimental conditions (see methods, Fig. EV1G,H).

The reduction in nuclear accumulation could be due to a decrease in nuclear import, an increase in nuclear export, or both. To test these possibilities, we quantified the rate of nuclear export by photo-converting Dendra2 during interphase and measuring red Dendra2 signal over time. Using this method, we have previously shown that nuclear export of H3 is negligible (Shindo and Amodeo, 2019). Here, we find that export of H3.3 is also negligible (Fig. EV1E). These data suggest that the distinct behavior of H3 and H3.3 nuclear availability are due to their import dynamics. To further assess how nuclear uptake dynamics changed during these cycles, we tracked total nuclear H3 and H3.3 in each cycle (Fig. 2C,D). Since nuclear export is effectively zero, we attribute the increase in total H3.3 over time solely to import, and therefore the slope of total H3.3 over time corresponds to the import rate. Though the change in initial import rates between NC10 and NC13 are similar between the two histones (Fig. EV1F), we observed a notable difference in their behavior in NC13. H3 nuclear accumulation plateaus ~5 min into NC13, whereas H3.3 nuclear accumulation merely slows (Fig. 2C,D). These changes in nuclear import and incorporation result in a less complete loss of the free nuclear H3.3 pool (~70% free in NC11 to ~30% in NC13) than previously seen for H3 (~55% free in NC11 to ~20% in NC13) (Fig. 2E)(Shindo and Amodeo, 2019).

### Chaperone binding sites regulate the differences in H3 and H3.3 chromatin incorporation

We next investigated what differences between H3 and H3.3 caused the observed trends in chromatin incorporation. There are two major differences between H3 and H3.3: protein sequence and expression pattern. H3 differs from H3.3 by four amino acids, which create an additional phosphosite in H3.3 and generate differing affinities for specific H3-family histone chaperones (Tagami et al, 2004). H3 is also generally expressed at much higher levels and in a replication-dependent manner. There are ~100 copies of H3 in the Drosophila genome, but only 2 of H3.3 (H3.3A and H3.3B) (Horard and Loppin, 2015). To determine which factor controls nuclear availability and chromatin incorporation, we genetically engineered flies to express Dendra2-tagged H3/H3.3 chimeras at the endogenous H3.3A locus, keeping the H3.3B locus intact. These chimeras include (i) H3.3's phosphosite replaced

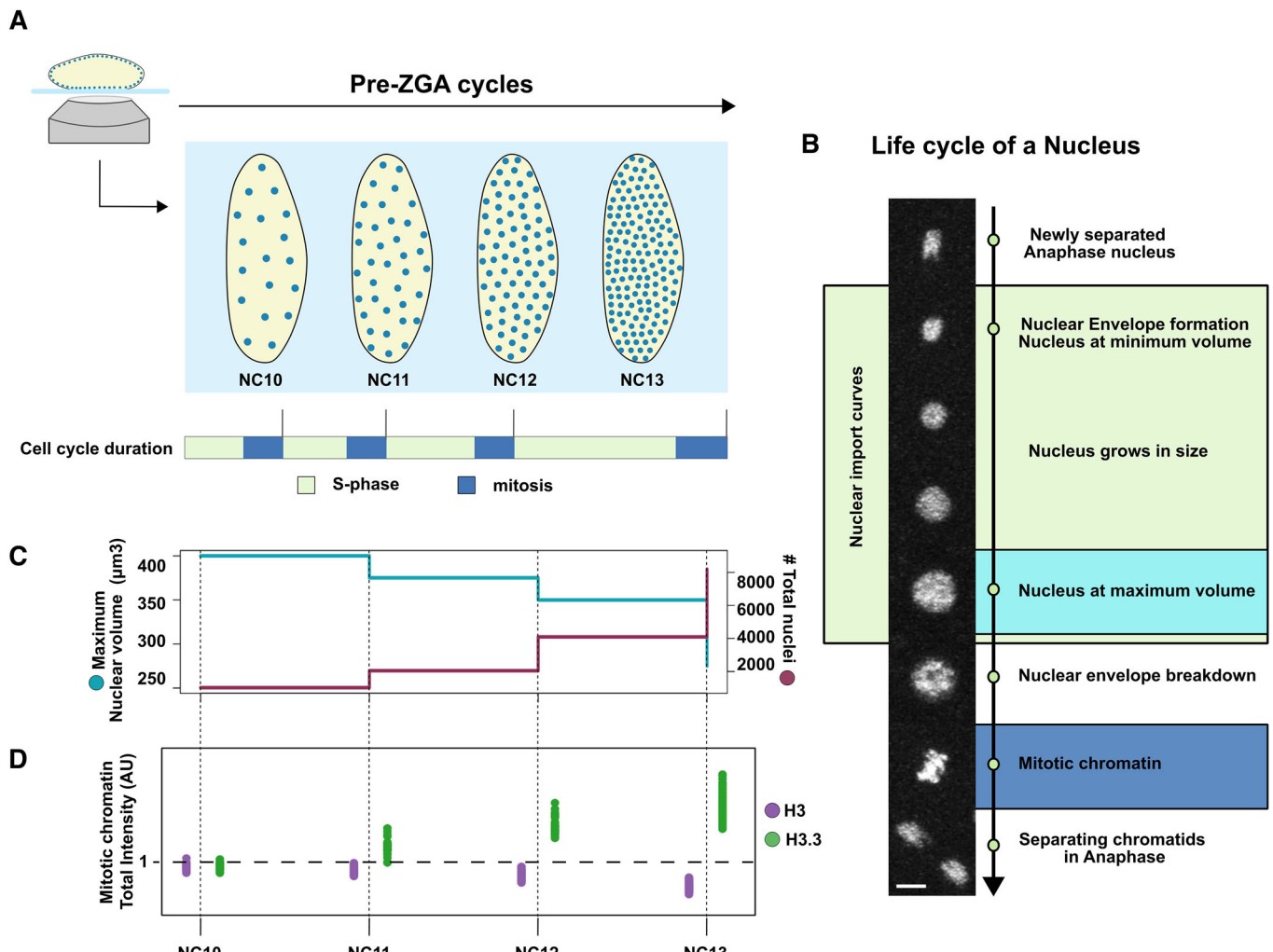

**Figure 1. Nuclear dynamics in *Drosophila* embryos during the pre-ZGA cycles.**

(A) Schematic representation of a *Drosophila* embryo imaged during nuclear cycle (NC) 10–13. The number of nuclei within the embryo doubles with each nuclear cycle. The relative durations of S-phase (mint) and mitosis (blue) for each nuclear cycle are depicted below. (B) Life cycle of a nucleus during each NC. The separated anaphase nucleus forms a nuclear envelope, grows to a maximum volume, and undergoes nuclear envelope breakdown in the subsequent mitotic division. The stages used in later figures for measurements of nuclear import curves (mint), nuclear concentration (cyan), and mitotic chromatin (blue) are indicated. Scale bar 5 μm. (C) Maximum nuclear volume attained by each nucleus (cyan) reduces as the total number of nuclei (maroon) doubles with each NC. (D) Total amount of histone H3-Dendra2 (purple) and its variant H3.3-Dendra2 (green) on mitotic chromatin. H3 levels fall while the H3.3 increases relative to their NC10 amounts (dotted horizontal line) over the pre-ZGA cycles. Note that statistical comparisons between the two Dendra2 constructs have not been done, as they were expressed from different loci and imaged under different experimental settings.

with Alanine from H3 (H3.3$^{S31A}$) (ii) H3.3's chaperone binding domain replaced with H3's (H3.3$^{SVM}$), and (iii) all four H3.3-specific amino acids replaced with those of H3 (H3.3$^{ASVM}$), (Fig. 3A). In all cases, the gene structure, including the promoter, intron, and UTRs of H3.3, remained intact and no other codons were changed to maximize similarity to the endogenous H3.3A locus. These chimeras were all viable and fertile (Fig. EV2I).

To study how chromatin incorporation differed in these chimeras, we measured their total intensities on mitotic chromatin during each nuclear cycle. We observed that, though H3.3$^{S31A}$ chromatin incorporation was significantly reduced compared to H3.3 by NC13, its levels increased on chromatin over the nuclear cycles, resembling H3.3 more than H3 (Figs. 3B and EV2A).

Conversely, the total amount of H3.3$^{SVM}$ and H3.3$^{ASVM}$ on mitotic chromatin fell over the nuclear cycles, similar to H3 (Figs. 3B and EV2B,C). This suggests that chromatin incorporation is mainly determined by the chaperone binding site. These results are broadly consistent with the final interphase nuclear concentrations and import dynamics where H3.3$^{S31A}$ was intermediate between H3 and H3.3 while H3.3$^{SVM}$ and H3.3$^{ASVM}$ were more similar to H3 (Figs. 3C and EV2A–G). However, both nuclear H3.3$^{S31A}$ and H3.3$^{SVM}$ fell more quickly than H3.3 and H3, respectively, suggesting that chimeric histones may not be as stable and/or efficiently imported as their canonical counterparts. We speculate that chimeric histone proteins (H3.3$^{S31A}$ and H3.3$^{SVM}$) are not as efficiently handled by the chaperone machinery as species

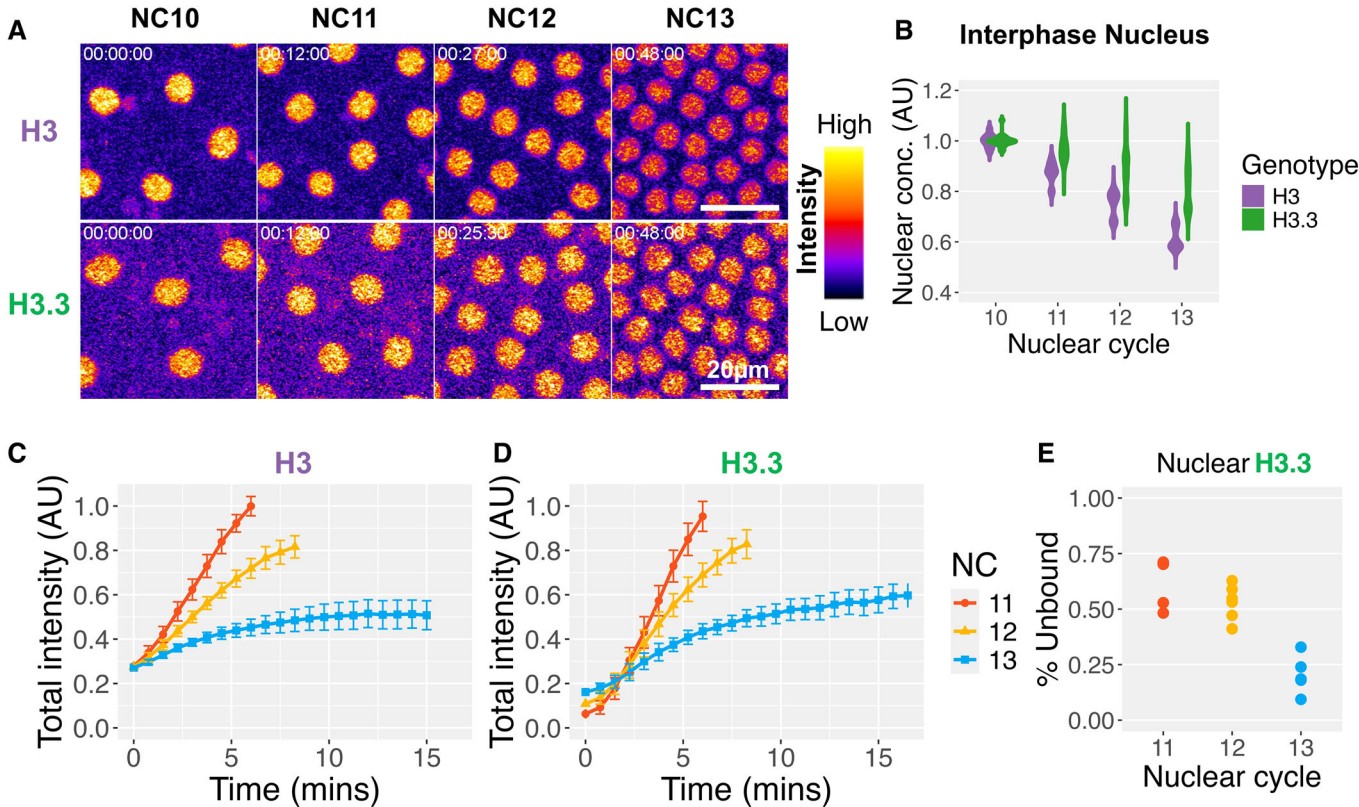

**Figure 2. Interphase nuclear availability of H3 decreases more rapidly than H3.3 over the pre-ZGA cycles.**

(A) Maximum intensity projections of H3-Dendra2 (top) and H3.3-Dendra2 (bottom) interphase nuclei 45 s before nuclear envelope breakdown (NEB) from NC10-13. Images are pseudo-colored with nonlinear look-up tables where purple indicates low and yellow indicates high intensities. Scale bar 20 μm. (B) Average interphase nuclear pixel intensities for H3-Dendra2 and H3.3-Dendra2 45 s before NEB in NC10-13, normalized to the average individual NC10 values. H3 and H3.3 concentrations decrease over time, but H3 loss is relatively more rapid. (C, D) Summed (total) pixel intensities for each nucleus over time for NC11-13, normalized to the maximum NC11 values for H3-Dendra2 (C) and H3.3-Dendra2 (D). Nuclear import plateaus after the first 5 min for H3, but merely slows and does not plateau for H3.3 in NC13. The solid line represents the mean, and the error bars represent the standard deviation. (E) The fraction of photoconverted unbound H3.3-Dendra2 after NEB in NC11-13 (see materials and methods for details). The "free" pool of H3.3 falls with each cycle. ($n = 3$ H3 and 5 H3.3 embryos in (B–D) and ≥5 embryos in (F); Statistical comparisons for (B) can be found in Appendix Table S2).

that are normally found in the organism, including H3.3[ASVM], which is protein-identical to H3. Together, these data indicate that the specific amino acid sequence of the chaperone binding site is the primary factor in differentiating the two histones for chromatin incorporation and nuclear import dynamics.

## H3 chaperone binding site conveys independence from Hira for chromatin incorporation

Since the chromatin incorporation of the H3/H3.3 chimeras appears to depend on their chaperone binding sites, we asked if impairing the canonical H3.3 chaperone, Hira, would affect the incorporation of H3.3[ASVM] expressed from the H3.3A locus. We generated embryos lacking functional maternal Hira using Hira[ssm-185b] (hereafter Hira[ssm]) homozygous mothers, which have a point mutation in the Hira locus (Loppin et al, 2000). This mutant Hira protein can bind but not incorporate H3.3 into chromatin (Figs. 3D,E and EV2H), resulting in sperm chromatin decondensation defects. These embryos develop as haploids and undergo one additional syncytial division before ZGA (NC14). Hira[ssm] embryos develop phenotypically normally through organogenesis and cuticle formation, but die before hatching

(Loppin et al, 2000). The fall in nuclear concentration of H3 is slightly more gradual in the haploid Hira[ssm] embryos than in wildtype, though H3 chromatin incorporation is not disrupted (Figs. 2B, 3F,G and EV2H). To test if H3-like chimeras expressed from the H3.3A locus use the canonical Hira pathway, we measured import and chromatin incorporation of H3.3[ASVM] in Hira[ssm]. We found that H3.3[ASVM] interphase nuclear concentration was more stable than H3 or H3.3 in Hira[ssm] embryos (Figs. 3H and EV2H). This stability is reflected in H3.3[ASVM] chromatin incorporation, where it only drops by ~20% between NC10 and NC14 compared to the observed ~40% drop in H3 (Fig. 3F–H). These data indicate that H3.3[ASVM] chromatin incorporation is Hira independent, even when expressed from the H3.3A locus.

## Local N/C ratios determine H3 and H3.3 chromatin incorporation

Since the N/C ratio controls many aspects of pre-ZGA development, we asked whether the local N/C ratio determines histone chromatin incorporation within a nuclear cycle. To test this, we employed mutants in the gene Shackleton (shkl) whose embryos

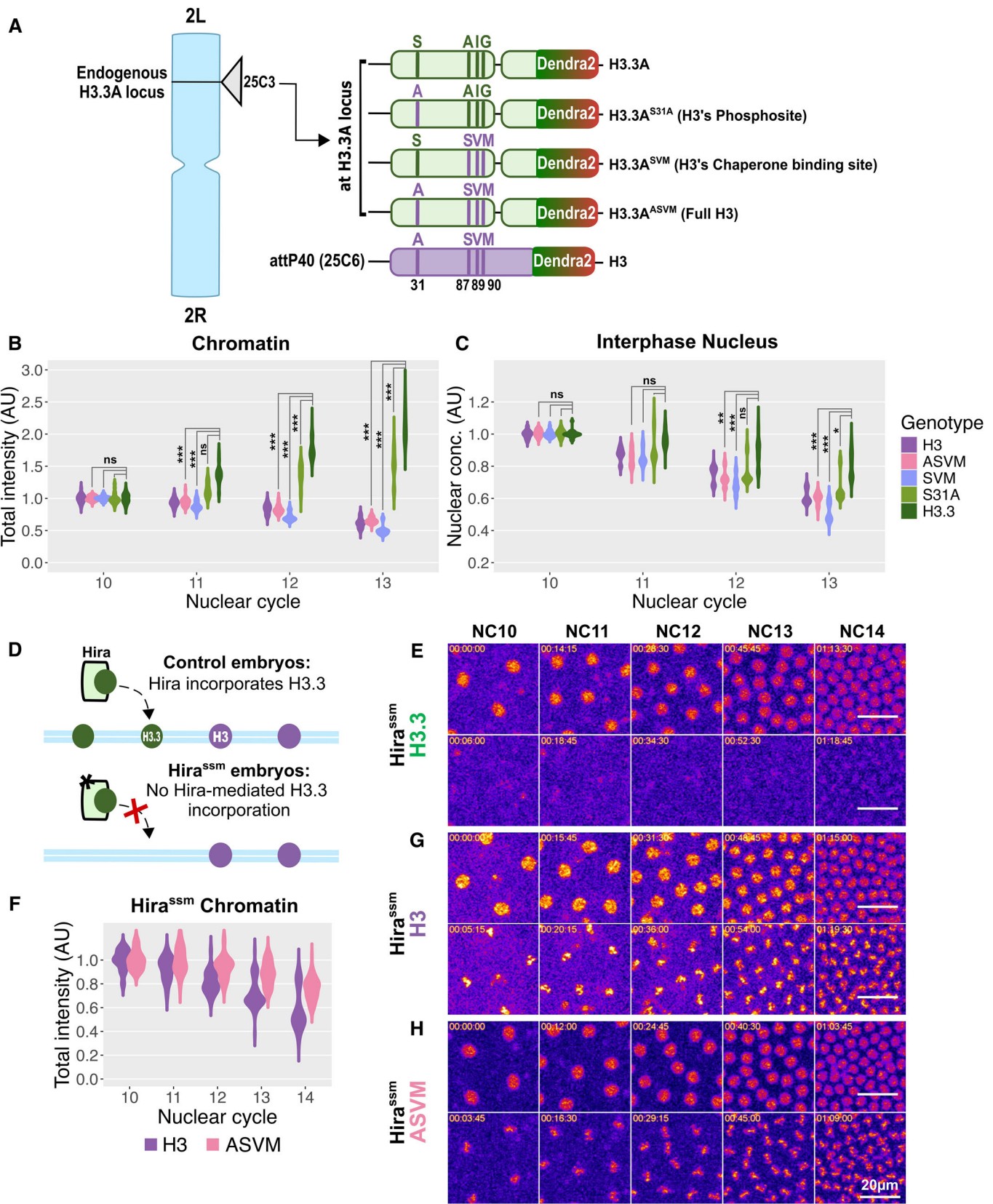

**Figure 3. The chaperone binding site determines H3 variant chromatin incorporation.**

(A) Schematic of the Dendra2-tagged H3/H3.3 replacement chimeras and H3.3 at the endogenous H3.3A locus. Pseudo-endogenous H3-Dendra2 is shown for comparison. S31A: H3.3 phosphosite (S) replaced with that of H3 (A), SVM: H3.3 chaperone binding site (AIG) replaced with that of H3 (SVM), and ASVM: all H3.3-specific amino acids replaced with those from H3. Refer to Fig. EV1A,B for genomic details. (B) Total intensities on mitotic chromatin of chimeras during NC10-13 were normalized to their NC10 values. The same data for H3-Dendra2 and H3.3-Dendra2 are shown in Fig. EV1C. H3.3$^{S31A}$ increases similarly to H3.3, while the constructs containing the H3 chaperone binding site decrease similarly to H3. (C) Interphase nuclear concentrations of chimeras 45 s before NEB during NC10-13, normalized to their NC10 values. H3-Dendra2 and H3.3-Dendra2 from Fig. 2B are included for reference. As seen for chromatin, nuclear accumulation generally follows the behavior of the chaperone binding site. Statistical significance was determined by two-way ANOVA, ns= $p > 0.05$, *$p < 0.05$, **$p < 0.01$, ***$p < 0.001$. Statistical comparisons for (B, C) can be found in the Appendix Tables S6, 7. (D) Schematic of H3.3 incorporation in control embryos and Hira$^{ssm}$ mutants. H3.3 is imported to the nucleus, but the mutant Hira chaperone fails to incorporate H3.3. Hira mutants develop as haploids and undergo one additional fast nuclear division. (E, G, H) Representative maximum intensity projections during interphase and mitosis over NC10-14: interphase nuclei (top) and mitotic chromatin (bottom) for H3.3-Dendra2 (E), H3-Dendra2 (G), and H3.3$^{ASVM}$-Dendra2 (H). Images are pseudo-colored with nonlinear look-up tables such that purple indicates low intensities and yellow indicates high intensities. Scale bar 20 μm. Hira$^{ssm}$ mutation nearly abolishes the observable H3.3 on mitotic chromatin (E). (F) H3 and H3.3$^{ASVM}$ continue to accumulate on chromatin in Hira$^{ssm}$ mutants. Total intensities of H3-Dendra2 (purple) and H3.3$^{ASVM}$-Dendra2 (pink) on mitotic chromatin in Hira$^{ssm}$ embryos between NC10-14, normalized to their average NC10 values. Though H3.3$^{ASVM}$ is successfully incorporated without active Hira, the chromatin amounts decrease more slowly than H3. ($n = 5$ all chimeras, 3 H3 ssm, 4 H3.3 ssm, and 5 H3.3$^{ASVM}$ ssm embryos.).

have non-uniform nuclear densities and therefore a gradient of nuclear sizes across the anterior/posterior axis (Fig. 4A,B; Movies EV1, 2) (Hayden et al, 2022). In these embryos, impaired cortical migration of early nuclei increases the N/C ratio in the center and decreases it in the posterior, which results in frequent partial extra divisions at the posterior pole (Fig. 4B) (Hayden et al, 2022). Although the nuclear densities within the embryos are altered, the shkl mutation does not affect the deposition of total H3 proteins or Hira mRNA (Fig. EV3D–F). For our analyses, we manually defined low and high nuclear density regions, with the low-density region always undergoing an extra division (Fig. 4B, see Methods). To control for potential positional effects, we measured chromatin incorporation at the middle and pole regions of control embryos for comparison (Fig. 4A). In control embryos, the drop in the total amount of H3 and rise in total H3.3 on chromatin are comparable between the middle and pole over the pre-ZGA cycles (Fig. 4C,D). In contrast, in shkl embryos, we observe decreased incorporation of H3 on chromatin at high nuclear densities compared to low nuclear densities (Figs. 4E and EV3A). This trend is reversed for H3.3, where chromatin from high-density regions has more total H3.3 than chromatin from low-density regions (Figs. 4F and EV3B).

This observation indicates that incorporation of H3 and H3.3 are reciprocal and depends on the local N/C ratio, leading to several possible models (Fig. 4G). First, the H3 pool available for chromatin incorporation may become limiting at high N/C ratios, leading to increased H3.3 incorporation. Second, since H3.3 is known to be associated with sites of active transcription (Ahmad and Henikoff, 2002; Chen et al, 2013; Chow et al, 2005; Jullien et al, 2012; Ng and Gurdon, 2008; Sakai et al, 2009; Sitbon et al, 2020), the increased H3.3 incorporation might be downstream of N/C ratio-dependent ZGA. Finally, since H3 is usually incorporated only during the S-phase, the changing H3 to H3.3 incorporation rates may be the result of N/C ratio-dependent cell cycle changes. Note that all these processes feedback onto one another, such as cell cycle slowing allowing time for ZGA (Strong et al, 2020; Syed et al, 2021).

## H3 nuclear availability depends on the local N/C ratio

To ask whether nuclear availability can explain the N/C ratio-dependent differences in H3 and H3.3 incorporation, we measured

their interphase accumulation in shkl embryos (Fig. 5A). Since H3 and H3.3 both have negligible nuclear export, their nuclear availabilities are determined by their import rates (Figs. 2B and EV1E) (Shindo and Amodeo, 2019). To assess the impact of the N/C ratio on nuclear import in individual nuclei, we calculated the number of neighbors within a 20 μm radius for each nucleus at its minimum volume (Figs. 4A,B and EV3C). We then binned the nuclei by their number of neighbors and determined their nuclear import curves for both H3 and H3.3. In control NC13 embryos, there is little variation in the number of neighbors, and all nuclei import H3 and H3.3 similarly (Fig. 5B–E). In NC13 shkl embryos, H3 import is anticorrelated with the local N/C ratio (Figs. 5F,H and EV4A). We observed slower H3 nuclear uptake at high N/C ratios, resulting in lower total interphase H3 accumulation (Fig. 5F). This was also reflected in the initial H3 import rates, where the nuclei with fewer neighbors had higher slopes (Fig. 5H). H3.3 uptake was less affected by the local N/C ratio (Fig. 5G,I and EV4B). A similar trend was also observed in NC12 for both histones, where more neighbors correspond to slower import. However, the range of behaviors was not as large as seen in NC13 (Fig. EV4C,D). These observations support a model where H3 pools are exhausted by the increasing N/C ratio, increasing the relative availability of H3.3 to H3 over the pre-ZGA cycles.

## H3.3 incorporation is not caused by exhaustion of H3 pools

Given that the available H3 seems to be depleted by the increasing N/C ratio, we sought to test if H3.3 chromatin incorporation depends on the size of the H3 pool (Fig. 6A). We hypothesized that as the embryo exhausted the supply of RD H3, it might increase the use of RI H3.3 to compensate. We knocked down Stem-loop binding protein (Slbp), which specifically binds and stabilizes the mRNAs of RD histones, including H3, but does not interact with H3.3 mRNAs (Lanzotti et al, 2002; Marzluff et al, 2008; Wagner et al, 2005). Slbp-RNAi dramatically decreases the size of the available H3 pool and results in frequent chromosomal segregation defects (Fig. EV5A,C)(Chari et al, 2019). For this reason, we only analyzed embryos that appeared reasonably healthy until the final cell cycle under consideration. All embryos that survived through at least NC12 had elongated cell cycles in NC12, and 60% arrested in

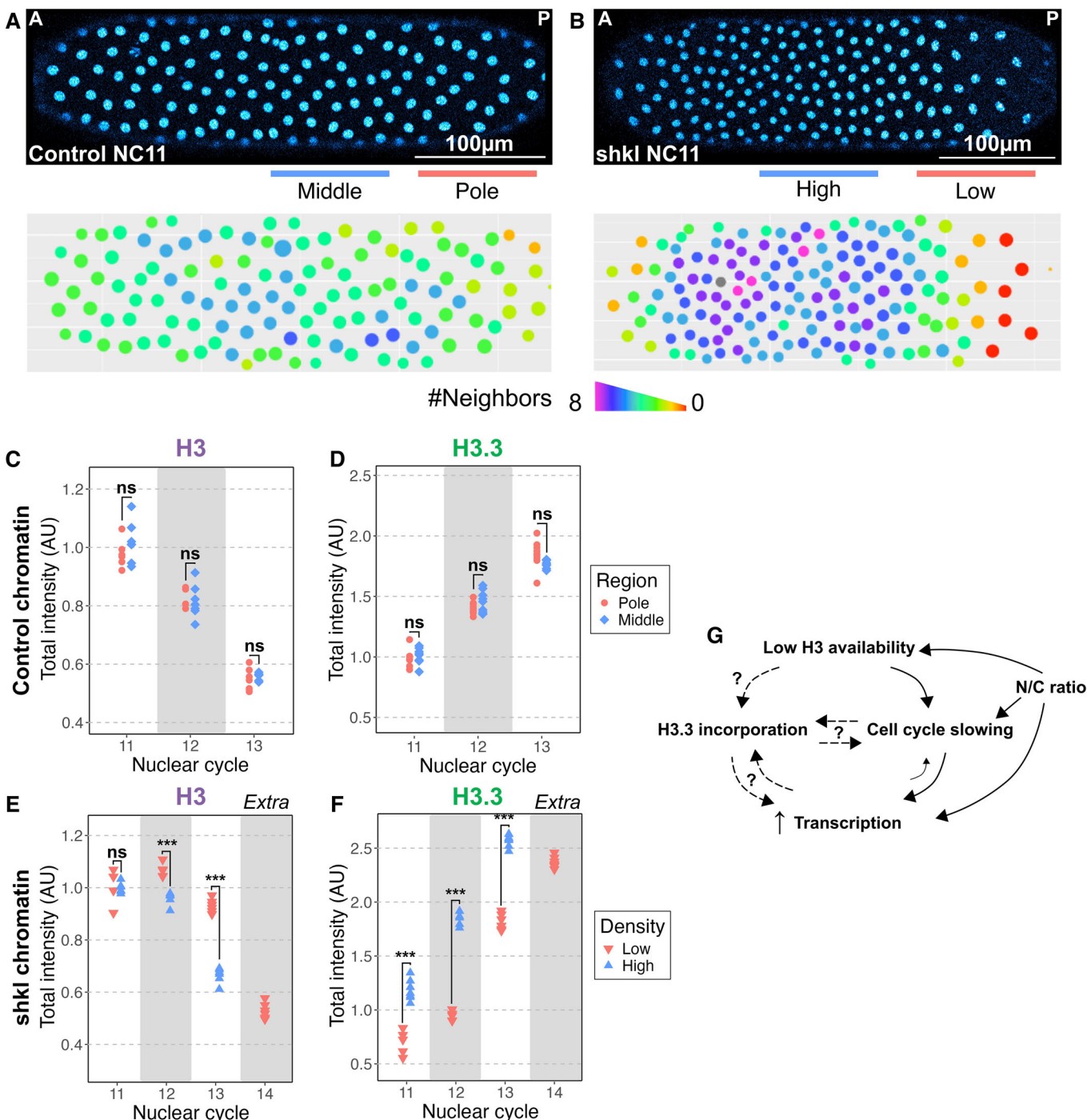

#Neighbors 8 ▶ 0

NC13 as reported previously, indicating the effectiveness of the knockdown (Fig. EV5E; Movies EV3–5) (Chari et al, 2019). In these embryos, H3.3 incorporation is largely unaffected by the reduction in RD H3 (Fig. 6B). To further validate that the lack of effect on H3.3 incorporation was not due to inefficient Slbp-knockdown, we also tested H3.3 incorporation in embryos that already display severe bridging in NC11. In these embryos, we detected no difference in the H3.3 incorporation in NC10 mitosis (Fig. EV5B). These results strongly indicate that simply running out of H3 is not the cause of the observed increase in H3.3 on chromatin.

## H3.3 incorporation does not depend on Zelda-dependent ZGA

Since H3.3 is associated with sites of active transcription in other systems (Ahmad and Henikoff, 2002; Chen et al, 2013; Chow et al, 2005; Jullien et al, 2012; Ng and Gurdon, 2008; Sakai et al, 2009; Sitbon et al, 2020), we next sought to test if H3.3 incorporation during ZGA depends on transcription. To do this, we knocked down the critical pioneer transcription factor Zelda (Figs. 6C and EV5D). Zelda controls the transcription of the majority of Pol II

**Figure 4. Local N/C ratios determine H3 and H3.3 chromatin incorporation.**

(A) Example NC11 control embryo with middle (blue) and pole (red) regions labeled as used in (C, D). The number of neighbors for each nucleus at its minimum volume within a 20 μm radius for the embryo is shown below. In controls, there is little variation in the number of neighbors. (B) Example NC11 shkl embryo with high (blue) and low (red) density regions labeled as used in (E, F). In the bottom panel representing the number of neighbors, note the increased range in the number of neighbors in shkl embryos. Scale bar 100 μm. (C, D) Total intensities on mitotic chromatin of H3-Dendra2 (C) and H3.3-Dendra2 (D) during NC11-13 in a representative control embryo, where each point indicates a single nucleus. Total H3-Dendra2 intensities fall and H3.3-Dendra2 intensities rise uniformly between the middle and pole regions within each cycle. (E, F) Total intensities on mitotic chromatin of H3-Dendra2 (E) and H3.3-Dendra2 (F) during NC11-13 in a representative shkl embryo, where each point indicates a single nucleus. NC14 represents a partial extra division in the low-density region. Chromatin in the low-density region retains more H3 ($p = 1.53\text{e-}04$ (NC12), $p < \text{e-}15$ (NC13)) and incorporates less H3.3 ($p = 7.21\text{e-}13$ (NC11), $p < \text{e-}15$ (NC12 and NC13)) within the same cell cycle compared to the high-density region. Similar results were observed in replicate embryos (Fig. EV3A,B). (G) Direct and indirect mechanisms of H3.3 incorporation in response to the N/C ratio. H3.3 incorporation could be a direct result of reduced nuclear H3 availability. Here, the increasing demand for nucleosomes with the increasing number of genomes would be met by H3.3. The N/C ratio also controls transcription and cell cycle duration. H3.3 incorporation could be downstream of either process. (Statistical significance was determined by two-way ANOVA, ns= $p > 0.05$, ***$p < 0.001$).

genes during ZGA, but disruption of Zelda does not change RD histone mRNA levels (Harrison et al, 2011; Huang et al, 2021; Liang et al, 2008; O'Haren et al, 2025). We found that H3.3 chromatin incorporation did not change in Zelda-RNAi embryos despite their inability to cellularize and longer NC13s (Figs. 6D and EV5E). This suggests that the large increase in H3.3 incorporation detected by microscopy in the final nuclear cycles does not depend on bulk ZGA.

## H3.3 incorporation depends on cell cycle state, but not cell cycle duration

Finally, to test the contribution of the cell cycle on the N/C ratio-dependent accumulation of H3.3 on chromatin, we used mutants in Chk1 (grapes in *Drosophila*) that are less efficient in cell cycle slowing (Figs. 6E and EV5E). Although these mutants have normal H3.3 mRNA deposition (Fig. EV5F–H), the lack of checkpoints causes an unusually rapid NC13. These embryos attempt to enter mitosis before their DNA is fully replicated, resulting in mitotic catastrophe (Fogarty et al, 1994). We found that H3.3 accumulation is disrupted as early as NC12 ($P$ value $= 10^{-8}$) in Chk1 mutants (Fig. 6F). The decreased H3.3 incorporation was likely not due to DNA underreplication since Hoechst staining shows no significant decrease in NC12 (Fig. EV5J,L). However, the total H3-type histone deposition on chromatin was much more variable in NC12 Chk1 mutant embryos (Fig. EV5K,M). Importantly, the Chk1 mutants have relatively normal NC12 durations (Blythe and Wieschaus, 2015b; Fogarty et al, 1994). In our experiments, Chk1 NC12 was only ~1 min faster than wildtype and Chk1 embryos with comparable cell cycle durations still displayed reduced H3.3 incorporation (Figs. 6F and EV5E). To further isolate the effect of cell cycle length on H3.3 incorporation, we used the natural variation in NC13 duration in control embryos. When we plotted H3.3 chromatin signal against the total NC13 duration for control embryos, we found no correlation (Fig. EV5IF). This result suggests that cell cycle duration as such does not directly regulate H3.3 chromatin incorporation. Instead, Chk1 appears to regulate H3.3 incorporation in a manner that is not mediated solely by lengthening the cell cycle.

## Discussion

We demonstrate that H3.3 replaces H3 on chromatin leading up to ZGA in *Drosophila*. This process depends on the specific H3.3 chaperone binding site and is controlled by the N/C ratio. We

tested which aspects of the N/C ratio control the dynamic incorporation of H3.3 and found that cell cycle state, but not H3 availability or bulk transcription, is the major regulator of H3.3 behavior. Given the fact that H3.3 pool size does not respond to H3 copy number in other *Drosophila* tissues (McPherson et al, 2023), our results suggest that H3.3 incorporation dynamics are likely independent of H3 availability. In the case of the chimeric histone proteins, the incorporation behavior was dependent on the chaperone binding site. For example, H3.3$^{\text{ASVM}}$ import and incorporation were similar to H3 in control embryos, and H3.3$^{\text{ASVM}}$ was still incorporated in Hira$^{\text{ssm}}$ mutants. This is consistent with the chaperone binding site determining the chromatin incorporation pathway and suggests that H3.3$^{\text{ASVM}}$ likely interacts with H3 chaperones such as Caf1.

Chk1 mutants decrease H3.3 incorporation even before the cell cycle is significantly slowed. Cell cycle slowing has been previously reported to regulate the incorporation of other histone variants in *Drosophila* (Johnson et al, 2018). However, our results indicate that cell cycle state and not duration per se, regulates H3.3 incorporation. In most cell types, the primary role of Chk1 is to stall the cell cycle to protect chromatin in response to DNA damage. Therefore, Chk1 activity directly or indirectly affects the chromatin state in a variety of ways. For example, Chk1 mutants lose the mitosis-specific phosphorylation of H3 earlier in anaphase than wild-type embryos (Su et al, 1999). We speculate that Chk1's role in regulating origin firing may be particularly important in this context (Feijoo et al, 2001; Moiseeva et al, 2019). Late-replicating regions and heterochromatin first emerge during NC13, and Chk1 mutants proceed into mitosis before the chromatin is fully replicated in NC13 (Atinbayeva et al, 2024; Fogarty et al, 1994; McKnight and Miller, 1977; Seller et al, 2019; Shermoen et al, 2010). Since H3.3 is often associated with late-replicating heterochromatic regions, the decreased H3.3 incorporation in Chk1 mutants may be an indirect result of increased origin firing (Feijoo et al, 2001; Moiseeva et al, 2019). However, it is unlikely that the effect of Chk1 on H3.3 incorporation is directly due to loss of heterochromatin since the H3K9me3 mark only becomes prominent at NC13 in wildtype embryos (Atinbayeva et al, 2024). Another possibility is that the additional Chk1 phosphosite that is found in H3.3-S31 may be important for promoting H3.3 incorporation during ZGA (Sitbon et al, 2020).

The interaction between H3-type histones and Chk1 has additional significance since H3 nuclear concentration has been proposed to directly regulate cell cycle length through H3

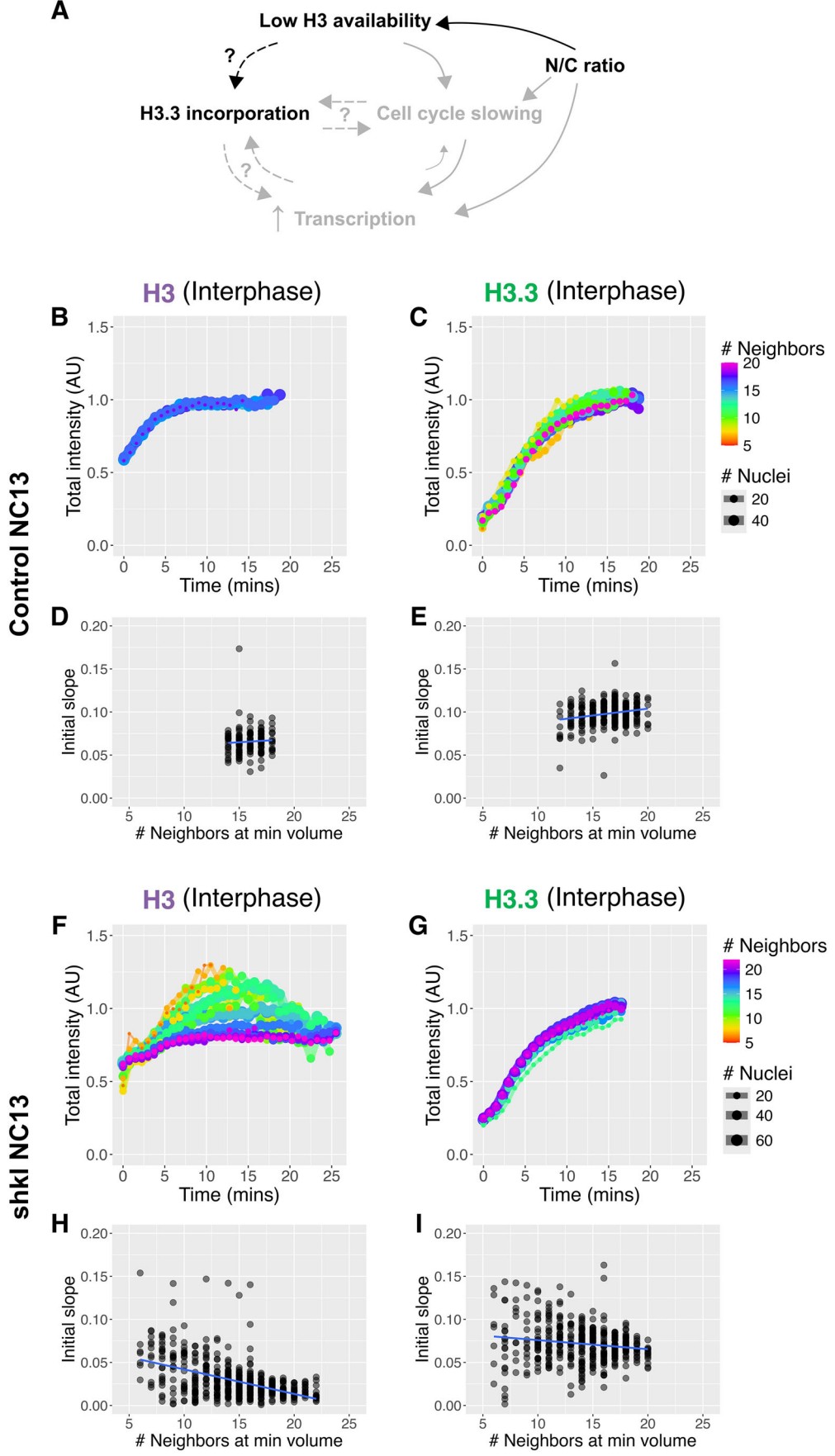

**Figure 5. Local N/C ratios differentially affect H3 and H3.3 nuclear availabilities.**

(A) Schematic of how the N/C ratio might affect H3 and H3.3 chromatin incorporation through loss of available H3. The bolded portion is the hypothesis under consideration. (B, C) Total intensities over time for nuclei in representative NC13 control embryos binned by the number of neighbors as in 4A for H3-Dendra2 (B) and H3.3-Dendra2 (C). Total intensity was normalized to the average maximum intensities achieved in NC13, and line color represents the number of neighbors. In controls, there is little variation in the number of neighbors or the import of H3 and H3.3 across the length of the embryo. (D, E) Initial slopes of nuclear import curves (change in total nuclear intensity over time for the first five timepoints) from representative NC13 control embryos from B and C for H3-Dendra2 (D) and H3.3-Dendra2 (E) plotted by the number of nuclear neighbors at their minimum nuclear volume. Note the uniformity in the number of neighbors and similarity in nuclear import behaviors in control embryos. (F, G) Total intensities over time for nuclei in representative NC13 shkl embryos binned by the number of neighbors as in 4B for H3-Dendra2 (F) and H3.3-Dendra2 (G). Nuclear import and accumulation of H3 inversely correlate with the number of neighbors, suggesting H3 nuclear import is N/C ratio sensitive. H3.3 nuclear import is less N/C ratio sensitive than H3. Similar results were observed in replicate embryos (Fig. EV4A–D). (H, I) Initial slopes of nuclear import curves (change in total nuclear intensity over time for the first five timepoints) from representative NC13 shkl embryos from F and G for H3-Dendra2 (H) and H3.3-Dendra2 (I) plotted by the number of nuclear neighbors. The slopes reflect a faster H3 uptake in nuclei with fewer neighbors and a slower H3 uptake in nuclei with more neighbors. Slopes in some nuclei with more neighbors are near zero, indicating that very little additional H3 is imported after nuclear envelope formation. Though the slopes reduce with the number of neighbors for H3.3, there is a non-negligible H3.3 import in the nuclei with the largest number of neighbors.

interactions with Chk1 (Shindo and Amodeo, 2021). In Hira[ssm] embryos that undergo one extra division before cell cycle slowing, the fall in nuclear H3 concentration between NC10 and the final fast cell cycle is strikingly similar to that seen in wildtype. Moreover, H3 nuclear concentrations appear to be strongly sensitive to the local N/C ratio in shkl embryos. This may be due to N/C ratio-dependent changes in nuclear import dynamics, which may also contribute to the observed changes in nuclear size across the shkl embryo (Nguyen et al, 2022). Together, these data are consistent with a model in which H3 nuclear concentrations regulate cell cycle slowing. However, H3.3 nuclear concentrations are less sensitive to the local N/C ratio than H3. Since H3.3 has an additional Chk1 phosphorylation site compared to H3, it may have different regulatory interactions with Chk1 (Chang et al, 2015; Sitbon et al, 2020). The relative contributions of both H3 and H3.3 nuclear availability to cell cycle slowing will require further exploration.

Finally, how the changing histone landscape contributes to ZGA remains an important open question. We have shown that bulk H3.3 incorporation does not depend on transcription from Zelda-dependent genes. However, the reciprocal relationship remains untested. H3.3 incorporation may increase transcription factor accessibility at specific genomic loci to mark them for activation. It is also possible that H3.3 incorporation occurs as a response to transcription initiated by other transcription factors, but does not specifically respond to the pioneer factor Zelda. We have shown that disruption of major ZGA does not impair bulk H3.3 incorporation, but the role of H3.3-containing nucleosomes in ZGA remains to be tested.

## Methods

### Reagents and tools table

| Reagent/resource | Reference or source | Identifier or catalog Number |
|---|---|---|
| **Experimental models** | | |
| y,w; 1xHisC.H3-Dendra2; | Amodeo lab | Shindo and Amodeo (2019) |
| y,w; H3.3A-Dendra2/CyO; | Amodeo lab | Shindo and Amodeo (2019) |

| Reagent/resource | Reference or source | Identifier or catalog Number |
|---|---|---|
| y,w; IX HisC.H3-Dendra2; shkl[GM163]/TM3 | IX HisC.H3-Dendra2 allele: Amodeo lab, shkl[GM163] allele: Di Talia lab | Shindo and Amodeo (2019) Yohn et al (2003) |
| y,w; IX HisC.H3-Dendra2; shkl[GM130]/TM3 | IX HisC.H3-Dendra2 allele: Amodeo lab, shkl[GM130] allele: Di Talia lab | Shindo and Amodeo (2019) Yohn et al (2003) |
| y,w; H3.3A-Dendra2/CyO; shkl[GM163]/TM6B | H3.3A-Dendra2 allele: Amodeo lab, shkl[GM163] allele: Di Talia lab | Shindo and Amodeo (2019) Yohn et al (2003) |
| y,w; H3.3A-Dendra2/CyO; shkl[GM130]/TM6B | H3.3A-Dendra2 allele: Amodeo lab, shkl[GM130] allele: Di Talia lab | Shindo and Amodeo (2019) Yohn et al (2003) |
| y,w; H3.3A-Dendra2[S31A]/CyO; | Amodeo lab | This paper |
| y,w; H3.3A-Dendra2[SVM]/CyO; | Amodeo lab | This paper |
| y,w; H3.3A-Dendra2[ASVM]/CyO; | Amodeo lab | This paper |
| ssm[185b],w/FM7c,w[a]; 1X HisC.H3-Dendra2; | Amodeo lab | Shindo and Amodeo (2019) |
| ssm[185b],w/FM7c,w[a]; H3.3A-Dendra2/CyO; | Amodeo lab | Shindo and Amodeo (2019) |
| ssm[185b],w/FM7c,w[a]; H3.3A-Dendra2[ASVM]/CyO; | Amodeo lab | This paper |
| y[1] v[1]; P{y[+t7.7] v[+t1.8] =TRiP.HMJ21114}attP40 (Slbp-RNAi) | Bloomington Drosophila Stock Center | BDSC: 51171 Perkins et al (2015) |
| ;;UAS-Zld-shRNA | Rushlow lab | |
| y,w; Mat-α-tub67-gal4, H3.3A-Dendra2 / CyO; Mat-α-tub15 | Weischaus lab, H3.3A-Dendra2 allele: Amodeo lab | Hunter and Wieschaus (2000), Shindo and Amodeo (2019) |
| y[1] sc[*] v[1] sev[21]; P{y[+t7.7] v[+t1.8] =TRiP.GL00094}attP2 (white-RNAi) | Bloomington Drosophila Stock Center | BDSC: 35573 Perkins et al (2015) |

| Reagent/resource | Reference or source | Identifier or catalog Number |
|---|---|---|
| y,w; H3.3A-Dendra2, grp[1]/CyO; | Weischaus lab, H3.3A-Dendra2 allele: Amodeo lab | |
| y,w;; | Weischaus lab | BDSC: 1495 |
| nos-PBac | Shvartsman Lab | |
| **Recombinant DNA** | | |
| pScarlessHD- H3.3 A[S31A]-Dendra2-DsRed | Amodeo lab | This paper |
| pScarlessHD- H3.3 A[SVM]-Dendra2-DsRed | Amodeo lab | This paper |
| pScarlessHD- H3.3 A[ASVM]-Dendra2-DsRed | Amodeo lab | This paper |
| pU6-BbsI-chiRNA | Harrison Lab, O'Connor-Giles Lab, Wildonger Lab | Addgene: 45946 |
| pU6-H3.3A-chiRNA | Amodeo lab | This paper |
| pU6-H3.3A-chiRNA_v2 | Amodeo lab | This paper |
| **Antibodies** | | |
| Anti-H3 | Abcam | Abcam: ab1791 |
| Anti-H3K9Me3 | Abcam | Abcam: ab8898 |
| Alexa Fluor 647 donkey anti-rabbit IgG | Invitrogen | Invitrogen: A31573 |
| **Oligonucleotides and other sequence-based reagents** | | |
| Guide-RNA for chimeras_1: GCGCGTCACCATTATGCCCA | Amodeo lab | This paper |
| Guide-RNA for chimeras_2: GCAAGGCGCCCCGCAAGCAGC | Amodeo lab | This paper |
| TaqMan gene expression array: Slbp (FAM) | Applied Biosystems | Applied Biosystems: Dm02135120_g1 |
| TaqMan gene expression array: zld (FAM) | Applied Biosystems | Applied Biosystems: Dm01845528_s1 |
| TaqMan gene expression array: His3.3A (FAM) | Applied Biosystems | Applied Biosystems: Dm02538716_s1 |
| TaqMan gene expression array: His3.3B (FAM) | Applied Biosystems | Applied Biosystems: Dm02330817_gH |
| TaqMan gene expression array: Hira (FAM) | Applied Biosystems | Applied Biosystems: Dm01833585_g1 |
| TaqMan gene expression array: Gapdh2 (VIC) | Applied Biosystems | Applied Biosystems: Dm01843776_s1 |
| **Chemicals, enzymes, and other reagents** | | |
| Protease inhibitor cocktail | Sigma | Sigma: P2714 |
| Hoechst 33342 | Thermo Scientific | Thermo Scientific: 62249 |
| EverBrite mounting medium | Biotium | Biotium: 23001 |
| Halocarbon oil | Sigma | Sigma: H8773 |
| TGX Stain-Free™ FastCast™ Acrylamide Kit, 12% | Bio-Rad | Bio-Rad: 1610185 |
| PicoPure™ RNA Isolation Kit | Applied Biosystems | Applied Biosystems: KIT0204 |
| ProtoScript First Strand cDNA Synthesis Kit | New England Biolabs | New England Biolabs: E6560L |

| Reagent/resource | Reference or source | Identifier or catalog Number |
|---|---|---|
| TaqMan GEX master mix | Applied Biosystems | Applied Biosystems: 4369016 |
| **Software** | | |
| Ilastik-1.4.0 | Open source | https://www.ilastik.org |
| Fiji (2.14.0/1.54 f) | Open source | https://fiji.sc/ |
| Zen 3.3 (Blue edition) | Zeiss | |
| R | Open source | https://www.r-project.org |
| Image lab | Bio-Rad | http://www.bio-rad.com/en-us/product/image-lab-software?ID=KRE6P5E8Z |
| **Other** | | |
| Bio-Rad ChemiDoc MP | Bio-Rad | 12003154 |
| Zeiss LSM980 confocal microscope with Airyscan-2 | Zeiss | |
| StepOnePlus RT-PCR machine | Applied Biosystems | 4376600 |

## Drosophila husbandry

All fly stocks were maintained at room temperature, on standard molasses media. The egg-laying cages were set up to collect embryos at 25 °C (except for the Slbp-RNAi flies). Slbp-RNAi egg lay cages and associated control w-RNAi cages were set up at 18 °C. Embryos from these cages were collected on apple juice agar plates with yeast paste, dechorionated with 50% bleach for up to 2 min, and washed twice with dH$_2$O. The $ssm^{185b}$ embryos were collected from $ssm^{185b}/ssm^{185b}$ homozygous females. For shkl embryos, the 2 shkl lines were crossed to obtain $shkl^{GM130e}/shkl^{GM163e}$ transheterozygous females, and their embryos were imaged. For all the RNAi crosses, males from the gal4-driver line, were crossed with virgins from UAS-RNAi lines to obtain progeny expressing both UAS and Gal4. Embryos from these progeny flies were used for imaging. Embryos from w-RNAi flies were used as controls for all RNAi experiments. For all the live imaging and RT-qPCR experiments, Chk1$^{-/-}$ embryos were collected from H3.3-Dendra2 grp[1] homozygous females. For all immunostaining experiments, Chk1$^{-/-}$ embryos were collected from grp[1] homozygous females heterozygous for H3.3-Dendra2.

## Plasmids and transgenesis

To generate stocks of Dendra2-tagged H3/H3.3 chimeras, CRISPR-Cas9 editing was performed at the endogenous H3.3 A locus. To this end, pScarlessHD-H3.3A-Dendra2-DsRed plasmid, reported in Shindo and Amodeo (2019), was modified through site-directed mutagenesis to express H3.3 with H3-specific amino acids, generating pScarlessHD-H3.3A[S31A]-Dendra2-DsRed (S31A mutation), pScarlessHD-H3.3A[SVM]-Dendra2-DsRed (A87S, I89V, G90M mutations), and pScarlessHD-H3.3A[ASVM]-Dendra2-DsRed (S31A, A87S, I89V, and G90M mutations) plasmids (Genscript).

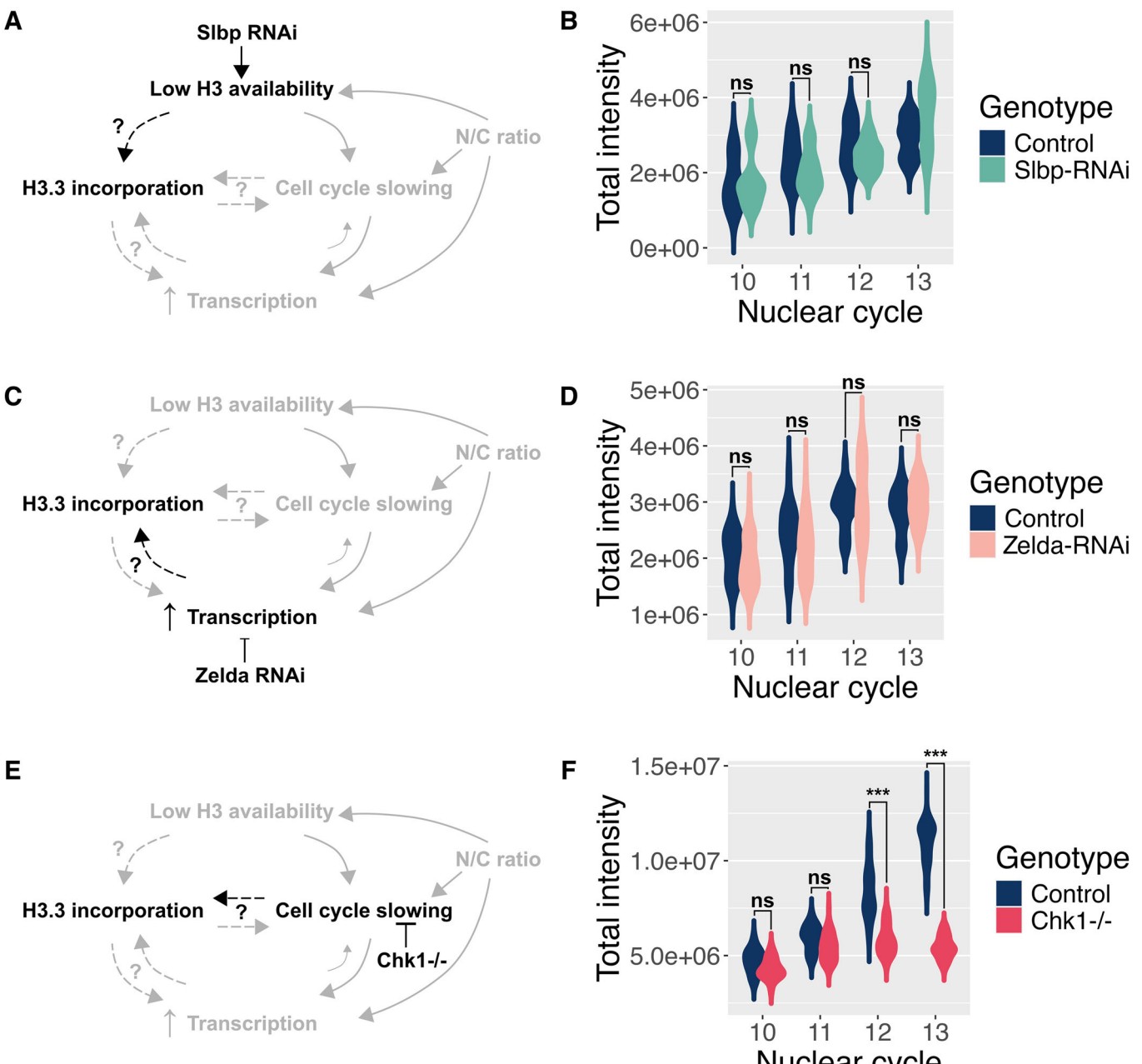

**Figure 6. H3.3 incorporation depends on cell cycle state but not H3 availability or Zelda-dependent transcription.**

(**A, C, E**) Schematics of different parameters that may regulate H3.3 chromatin incorporation. The bolded portion is the hypothesis under consideration in (**B, D, F**), respectively. (**A**) Slbp-RNAi decreases the size of the available H3 pool. (**B**) Total intensities of H3.3-Dendra2 on mitotic chromatin in white-RNAi (control) and Slbp-RNAi backgrounds during NC10-13. H3.3 incorporation does not increase upon lowering H3 availability. Note that most Slbp-RNAi embryos are arrested in NC13 without dividing and therefore do not contribute to the mitotic NC13 data. Therefore, $n = 2$ for NC13 and statistical comparison has been omitted for NC13 in Slbp-RNAi embryos ($n = 5$ for NC10-12) (**C**) Zelda-RNAi inhibits the majority of zygotic transcription, allowing us to test if H3.3 incorporation depends on transcription. (**D**) Total intensities of H3.3-Dendra2 on mitotic chromatin in white-RNAi (control) and Zelda-RNAi backgrounds during NC10-13. H3.3 incorporation does not change upon inhibiting Zelda-dependent transcription. (**E**) Chk1 (grp[1]) mutation prevents cell cycle slowing, allowing us to test if H3.3 incorporation is dependent on cell cycle state. (**F**) Total intensities of H3.3-Dendra2 on mitotic chromatin in control and embryos from chk1[−/−] mothers during NC10-13. H3.3 incorporation is reduced in both NC12 ($p = 1.01e{-}08$) and NC13 ($p < e{-}15$), indicating that cell cycle state, though not cell cycle duration, regulates H3.3 incorporation. Note that these embryos are homozygous for H3.3-Dendra2 and have double the fluorescent intensity compared to all previous embryos. ($n \geq 5$ embryos, Statistical significance was determined by two-way ANOVA, ns= $p > 0.05$, $*p < 0.05$, $**p < 0.01$, $***p < 0.001$).

Two CRISPR target sites were identified using Target Finder (Gratz et al, 2014), one near the stop codon and one near S31, and the corresponding gRNAs were cloned into pU6-BbsI-chiRNA vector (a gift from Melissa Harrison & Kate O'Connor-Giles & Jill Wildonger, Addgene plasmid #45946). Each mutant plasmid was co-injected with both the gRNA plasmids into nos-Cas9 embryos (TH00787.N) and DsRed+ progeny were selected (BestGene). These progeny were then crossed with nos-PBac flies (a generous gift from Robert Marmion and Stas Shvartsman) to remove the DsRed marker. DsRed negative single males were then crossed with y,w;Sp/CyO; to establish stocks. Insertion of Dendra2-tagged mutants was verified by PCR and Sanger sequencing.

## Microscopy

For live imaging, dechorionated embryos were mounted on glass-bottom MatTek dishes in deionized water and imaged with a 20x, 0.8 NA, objective of Zeiss LSM980 confocal microscope with Airyscan-2 at 45 s intervals for 2 h at room temperature (19–22 °C). All H3-Dendra2-tagged embryos were imaged using a 488 nm laser at 2% power, and all lines expressing Dendra2-tagged proteins from the endogenous H3.3 A locus (H3.3 and the chimeras) were imaged with a 488 nm with 0.5% power in Airyscan multiplex CO-8Y mode. All but shkl embryos and their controls were imaged at a $700 \times 700$ pixel resolution, with 1 µm Z-steps over a 15 µm range, with a frame time of 26.06 ms. All shkl embryos and their controls (Figs. 4, 5 and EV3, EV4) were imaged at a $2836 \times 2836$ pixel resolution, with 1.2 µm Z-steps over a 14.4 µm range, with a frame time of 328.29 ms. All images were acquired with a pixel size of $0.149 \mu m \times 0.149 \mu m$.

Immunostaining experiments were imaged with a 20x, 0.8 NA, objective in the Airyscan multiplex CO-8Y mode with 2 lasers: 405 nm with 0.5% power for Hoechst staining and 639 nm with either 0.2% power for Pan-H3 staining or 0.3% power for the H3K9Me3 staining. The samples were imaged at an $844 \times 844$ pixel resolution, with 1.2 µm Z-steps over a 14.4 µm range. The images were acquired with a pixel size of $0.124 \mu m \times 0.124 \mu m$.

## Nuclear export and unbound H3.3 measurement through Dendra2 photoconversion

For measuring the nuclear export and amount of free histone H3.3 (Figs. 2E and EV1E), we used the photoconvertible Dendra2 tag and the interactive bleaching panel in Zen software. We used a 4 µm diameter circular stencil to interactively photo-convert the nuclei. H3.3-Dendra2 within a single nucleus was photoconverted from green-to-red using a 405 nm laser at 3% power with 60 iterations of laser exposure at a speed of 1.37 µs/pixel. The nucleus was converted in the middle of each nuclear cycle for NC11-13 and then imaged with 561 nm at 1% laser power and 488 nm with 0.5% laser power at 15-s intervals until the end of the nuclear cycle. Images were captured at $576 \times 576$ pixels resolution with 1 µm Z-steps over a 15 µm range, with a frame time of 66.55 ms for each channel. The images were acquired with a 40x oil immersion objective, 1.3 NA, with a pixel size of $0.092 \mu m \times 0.092 \mu m$.

## Photobleaching corrections

To assess the potential effects of fluorophore photobleaching during our image capture, we performed parallel embryo experiments. In these experiments, we identified two embryos of the same age and imaged the interphase nucleus and the metaphase chromatin for both in NC10. Following this, we image only a sub-region of one of the two embryos continuously with our experimental settings described for H3-Dendra2 embryos above until NC13, while keeping the other embryo to develop parallelly without imaging. Once the imaged embryo reached NC13, both the imaged and unimaged parallel embryo were imaged again. We quantified the total nuclear signal from both embryos to evaluate the photobleaching effects. We then compared the continuously imaged section of the embryo, with the area outside the sub-region imaged, as well as the unimaged parallel embryo. Using these comparisons, we determined that the effect of photobleaching was minimal and therefore did not apply a numeric photobleaching correction to our data (Fig. EV1G,H; Appendix Tables S4, 5).

## Nuclear segmentation and intensity analysis

All raw CZI output files from ZEN 3.3 (blue edition) live imaging were first 3D Airyscan processed at a strength of 3.7 and then converted into individual TIFF files.

For mitotic chromatin quantification, the timepoints corresponding to metaphase chromatin from each nuclear cycle were extracted, and the z-stacks were sum-projected in FIJI (2.14.0/1.54 f). These files were segmented using the "pixel classification + object classification" applet in the ilastik-1.4.0 software (Berg et al, 2019) into chromatin and cytoplasm. The individually segmented mitotic chromatin objects were then exported as a single CSV file containing object properties such as total intensity, mean intensity, and size. The total intensity within each chromatin mass was calculated and normalized to the average NC10 chromatin values (or NC11 for shkl embryos and their controls) for that genotype.

In shkl embryos and their controls (Figs. 4C–F and EV3A,B), chromatin was segmented from different regions within an embryo (middle and pole regions for control, and from low and high-density regions for shkl). In control embryos, the middle regions were defined by outlining a box ($250 \times 250$ pixels) in NC10 around the line separating the embryo into two halves. A similar-sized box was outlined with one edge at the tip of the embryo to define the pole region. In shkl embryos, the regions with the highest apparent nuclear density within the center was defined as the high-density region, and the region that underwent the partial extra division in NC14 was defined as the low-density region. To account for the asynchronous nature of the divisions in the shkl embryo, within each region, five to six nuclei that divided synchronously along the mitotic wave were quantified. For both control and shkl embryos, at least five nuclei per embryo were quantified in each cycle for each region.

For analyzing the interphase nuclear concentrations, nuclei from 45 s before the nuclear envelope breakdown were segmented in 3D using the "pixel classification + object classification" applet on ilastik software. The CSV file with the mean intensities of each nucleus was exported and normalized to the average NC10 nuclear concentration values for each genotype.

For obtaining the nuclear import curves, individual nuclear cycles were run through the pixel classification + object classification applet in the ilastik software. The results were exported as CSV files and processed with a custom R script. For shkl embryos (Fig. 5B,C,F,G), the pixel prediction maps were used with the "tracking with learning" applet (ilastik) to segment the nuclei as well as track them over time. The tracking result with object properties was exported as a CSV file and processed with a custom R script. Intensities were normalized by the average total intensity of the nuclei at their maximum size in each cycle. For each case, the volume was calculated by multiplying the voxel size by the "size in pixels" of an object. Import rates were calculated by using a linear regression for the total nuclear intensity over time for the first five timepoints in the nuclear import curves.

## Neighborhood analysis

Nuclei within each embryo were tracked over a single nuclear cycle using the "tracking + learning" applet on ilastik. The tracking result with the coordinates of each nucleus over time was obtained as a CSV file, along with other parameters, including the total intensity, mean intensity, and nuclear size. The CSV file was analyzed to calculate the number of nuclear neighbors for each nucleus within a 20 μm radius using a custom R script. The script calculates the number of neighbors each nucleus has at its minimum volume since the maximum nuclear import occurs at this time point. To overcome the noise from the incomplete edge nuclei, which are centered lower in the embryo, we utilized the differences in their Z-coordinates to filter them out, after using them for the number of neighbor calculations. For shkl embryos, as the nuclear cycles are asynchronous, nuclear divisions start at different timepoints within the same cell cycle, and the nuclear density changes as the neighboring nuclei divide. Therefore, the total intensity traces were aligned to match their minimum volumes (as shown in Fig. 1B) to T0. Nuclei with the same number of neighbors were binned together and weighted to reflect the number of nuclei being averaged. The total intensity curves were then normalized such that the average total intensity of the nuclei at their maximum size was equal to 1.

## Immunostaining

Embryos were collected from y,w;; and Chk1 mutant mothers after 2.5 and 4 h of egg laying, respectively. The embryos were dechorionated with 50% bleach for 2 min, followed by two washes with deionized water. Embryos were fixed in 4% paraformaldehyde/heptane for 15 min and devitellinized in 1:1 methanol/heptane followed by 2×washes in methanol. The fixed embryos were rehydrated in PBST and blocked with 3% BSA for 1 h at room temperature. Embryos were incubated in either rabbit anti-H3 antibody (1:500, Abcam: ab1791) or rabbit anti-H3K9Me3 antibody (1:500, Abcam: ab8898) overnight at 4 °C. They were then washed and incubated for 2 h in Alexa Fluor 647-conjugated donkey anti-rabbit IgG antibody (1:1000, Invitrogen: A31573) at room temperature. This was followed by washes and Hoechst staining (1:1000, Thermo Scientific #62249) for 30 min at room temperature. The embryos were mounted in EverBrite mounting medium (Biotium #23001) and imaged as described in the microscopy section.

## Cell cycle time measurements

Cell cycle durations were measured from metaphase to metaphase. To account for day-to-day temperature variability, we normalized the mean NC11 durations in control embryos to 10 min and scaled for other cell cycles in all embryos acquired on the same day, as done previously (Blythe and Wieschaus, 2015b).

## Western blot analysis

For shkl embryos and their controls, embryos were staged under halocarbon oil (Sigma, H8773), and NC14 embryos were collected. Embryos from Slbp-RNAi flies and their controls (w-RNAi) were collected after 1 h of egg laying. The embryos were dechorionated with 50% bleach for 2 min followed by two washes with deionized water. They were then collected in a microcentrifuge tube and lysed with forceps in ice-cold embryo lysis buffer (50 mM Tris pH 8.0, 150 mM NaCl, 0.5% Triton-X, 1 mM $MgCl_2$, 0.1 mM EDTA, and 1X protease inhibitor cocktail (Sigma: P2714)). Twenty-five embryos were collected per genotype to quantify pan-H3 levels. Lamelli buffer was added in a 1:1 volume, and the samples were boiled at 95 °C for 5 min. The protein lysates were run on a TGX Stain-Free 12% acrylamide gel (Bio-Rad Laboratories), stain-free activated for 45 s under UV, and transferred onto an LF-PVDF membrane. Membranes were incubated in rabbit anti-H3 antibody (1:1000, Abcam: ab1791) overnight at 4 °C. They were then washed and incubated for 2 h in Alexa Fluor 647-conjugated donkey anti-rabbit IgG antibody (1:2000, Invitrogen: A31573). The membranes were then imaged to detect fluorescence using a gel imager (Bio-Rad ChemiDoc MP).

## RNA isolation

Input RNA for qPCRs were isolated from dechorionated single embryos. For Slbp mRNA analysis, a 2 h collection was done to obtain embryos from Slbp-RNAi and w-RNAi mothers grown at 18 °C. For Zelda mRNA analysis, a 1 h collection was done to obtain embryos from Zelda-RNAi and w-RNAi mothers grown at 25 °C. For H3.3A, H3.3B, and Hira mRNA analysis in Chk1 mutant embryos, embryos were staged using the H3.3-Dendra2 fluorophore, and NC12 embryos were collected from grp[1], H3.3-Dendra2, and H3.3-Dendra2; mothers were grown at 25 °C. For Hira mRNA analysis in shkl mutant embryos, embryos were staged in halocarbon oil (Sigma, H8773) and NC14 embryos were collected from H3-Dendra2; shkl[GM130e]/shkl[GM163e] transheterozygous; and H3-Dendra2; mothers grown at 25 °C. Individual embryos were placed into LoBind RNAse-free tubes (Eppendorf 022431021) and lysed in 20 μl of lysis buffer (Applied Biosystems, KIT0204). RNA isolation was performed following the manufacturer's protocol.

## RT-qPCRs

cDNA was made from the RNA isolated from single embryos using random primer mix from ProtoScript First Strand cDNA Synthesis Kit following the manufacturer's protocol (New England Biolabs, E6560L). RT-qPCRs were performed on a StepOnePlus RT-PCR machine (4376600) using the TaqMan GEX master mix (Applied Biosystems, 4369016) and following gene expression arrays: Slbp (Dm02135120_g1, FAM), zld (Dm01845528_s1, FAM), His3.3 A (Dm02538716_s1, FAM),

His3.3B (Dm02330817_gH, FAM) and Hira (Dm01833585_g1, FAM) normalized to Gapdh2(Dm01843776_s1, VIC). RNA was analyzed from at least three individual embryos from each genotype.

## Hatch rate assays

About 10–15 virgin females were collected from all the genotypes (y,w; H3.3-Dendra2/+;, y,w ;S31A-Dendra2/+;, y,w; SVM-Dendra2/+;, y,w; ASVM-Dendra2/+; and y,w;;) and allowed to mate with y,w;; males for one day on standard molasses media fly food vials and then moved to the egg lay cages with apple juice agar plates and fresh yeast paste on day 2 (day 1 of quantification). For each genotype, up to 100 embryos from an overnight collection were moved to a fresh apple juice plate and assessed for hatching after 24 h. Hatch rates were calculated for 5 consecutive days of egg laying for each set. At least four independent egg-laying cages were measured for each genotype.

## Statistical analysis

Two-way ANOVA tests were conducted to assess the statistical significance between the dataset means of different genotypes over the nuclear cycles. All studies were performed with nuclei from at least three embryos. For shkl embryos, a two-way ANOVA test was used to determine the statistical significance of nuclei within different regions of the same embryo over the different nuclear cycles, with each nucleus as a replicate. For all other embryos, the average chromatin/nuclear values for each NC from each embryo were considered as a replicate. One-way ANOVA tests were conducted to assess the significance in RT-qPCR, western blot, and immunostaining experiments. For immunostaining experiments, one-way ANOVA tests were performed on normalized day averages of embryos to account for day-to-day variability. Results from these tests are reported in the Appendix Tables S1–10.

## Data availability

All the source data and code used to generate the main figures of this manuscript can be found at the following sources. Source data: Dryad (https://doi.org/10.5061/dryad.6m905qgcj). Code: Github (https://github.com/anushadbhatt/AM_paper).

The source data of this paper are collected in the following database record: biostudies:S-SCDT-10_1038-S44319-025-00596-1.

## Peer review information

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

## Acknowledgements

We thank Shruthi Balachandra, Eric Alpert, Grace Carey, and Kiera Schwarz for constructive discussion of this manuscript. We further thank the members of Bickel, Kasper, Lacefield, Moseley, and Landino labs at Dartmouth for their helpful suggestions. We thank Patrick Robison from the Dartmouth bioMT Core, Ann Lavanway, and Britton Johnson for technical support. We thank Stefano Di Talia for shkl stock flies, Eric Weischaus for ssm, grp, and gal4-driver stock flies; Robert Marmion for the nos-PBac transposase flies, Christine Rushlow for the Zelda-RNAi flies, and Robert Duronio for the 1xHisC plasmid. We thank FlyBase, funded by NHGRI and NIGMS, and Bloomington Drosophila Stock Center (NIHP40OD018537) for providing essential resources. This work was funded by NIH/NIGMS (P20-GM113132 and R35GM150853 to AAA). ADB is supported by the Sondra and Charles Gilman Graduate Research Fellowship.

## Author contributions

**Anusha D Bhatt**: Conceptualization; Data curation; Validation; Investigation; Visualization; Methodology; Writing—original draft; Writing—review and editing. **Madeleine G Brown**: Validation; Investigation; Writing—original draft. **Aurora B Wackford**: Investigation. **Yuki Shindo**: Conceptualization; Resources; Methodology. **Amanda A Amodeo**: Conceptualization; Resources; Supervision; Funding acquisition; Writing—original draft; Project administration; Writing—review and editing.

Source data underlying figure panels in this paper may have individual authorship assigned. Where available, figure panel/source data authorship is listed in the following database record: biostudies:S-SCDT-10_1038-S44319-025-00596-1.

## Disclosure and competing interests statement

The authors declare no competing interests.

# Expanded View Figures

**Figure EV1. Tools and controls underlying image quantification of mitotic and interphase Dendra2 and H3.3 export.**

(A) A 5 kb region containing a single histone gene cluster with one copy of each of the 5 replication-coupled histones, including their promoters, and UTRs as in the endogenous locus in which H3 was N-terminally tagged with the green-to-red photo-switchable fluorophore Dendra2 was inserted at the attp40 site on chromosome 2L (from Shindo and Amodeo, 2019). (B) The endogenous H3.3A gene locus was edited to express an N-terminally tagged H3.3-Dendra2 using CRISPR/Cas9 (from Shindo and Amodeo, 2019). (C) Total pixel intensities (corresponding to total amounts) on mitotic chromatin for H3-Dendra2 (purple) and H3.3-Dendra2 (green) between NC10-13, normalized to the average individual NC10 values. Chromatin-bound H3 decreases over NC10-13, whereas H3.3 increases. (D) Maximum intensity projections of H3-Dendra2 (top) and H3.3-Dendra2 (bottom) on mitotic chromatin from NC10-13. Images are pseudo-colored with nonlinear look-up tables such that purple indicates low intensities and yellow indicates high intensities. H3 intensities fall, and H3.3 intensities rise over the cycles. Scale bar 20 μm. (E) The total pixel intensity of individual photoconverted H3.3-Dendra2 nuclei in NC11-13 ($n = 5$). Time is shown relative to nuclear envelope breakdown (NEB). H3.3-Dendra2 intensity remains constant before NEB, indicating that the nuclear export is negligible. The loss of red signal at NEB represents the pool of unbound H3.3 in the nucleus. Each trace represents a single nucleus, each from different embryos. These data were used to plot the unbound H3.3 fraction in Fig. 2E. (F) Initial slopes of the nuclear import curves (change in total nuclear intensity over time for the first five timepoints) shown in (2C, D) for NC11-13. All slopes are normalized to NC11 values. (G, H) Total pixel intensity of H3-Dendra2 on chromatin (G) and interphase nuclei (H) with and without continued laser exposure. Two parallel embryos were used to obtain NC10 and NC13 mitotic chromatin with different levels of laser exposure for photobleaching correction (see Methods for details). The data were divided into three regions: "Image area" corresponds to nuclei imaged throughout NC10-13; "outside" corresponds to nuclei outside the image area, but within the imaged embryo; and "parallel embryo" corresponds to nuclei that were only imaged once in NC10 and once in NC13 without continued exposure. Photobleaching was observed to be negligible in experiments quantifying both the mitotic chromatin and interphase nuclear concentration. (Statistical comparisons for (C, E, G, H) can be found in Appendix Tables S1, 3-5).

▶

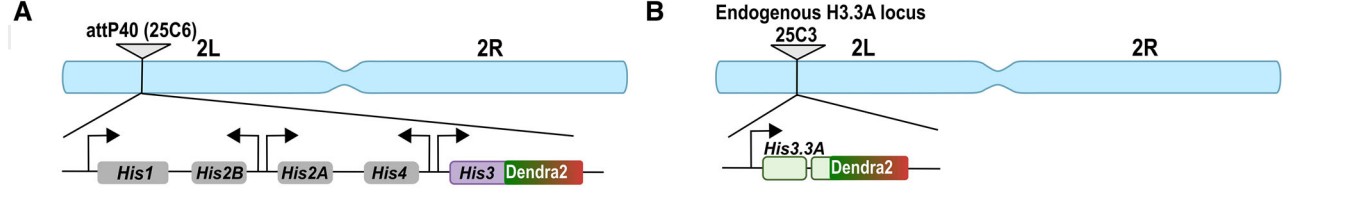

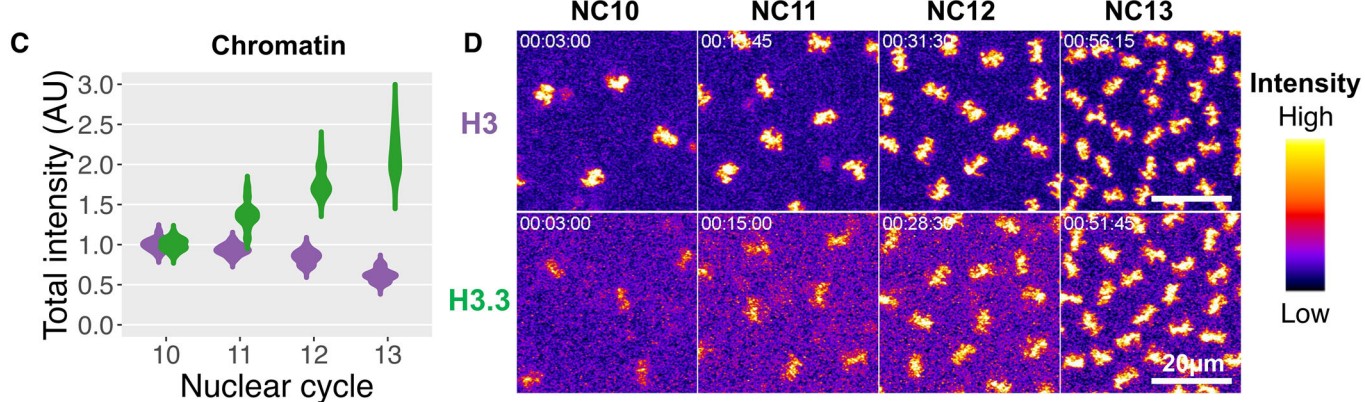

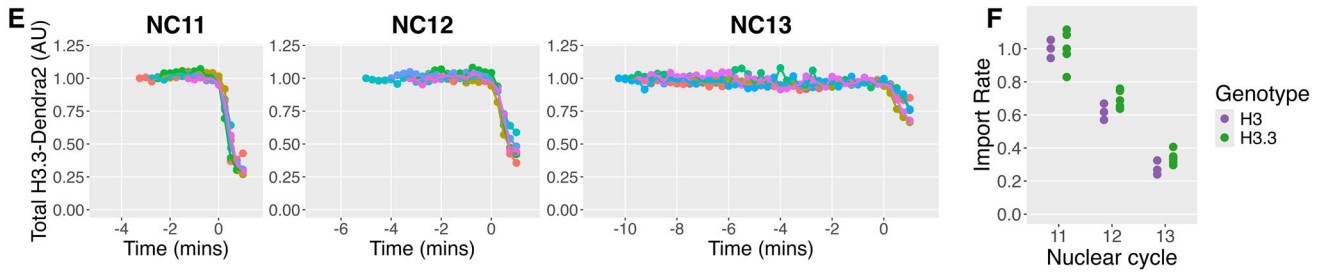

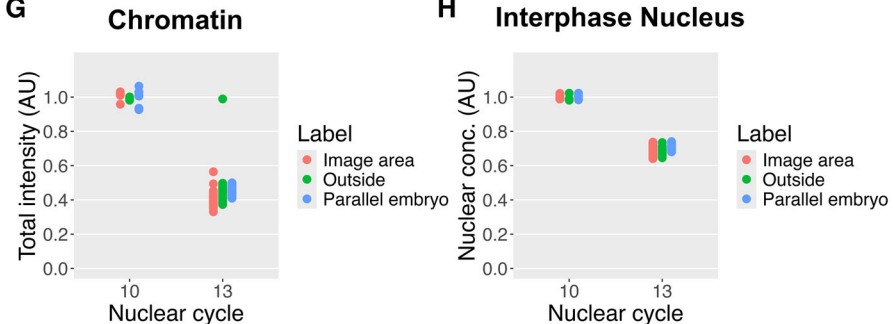

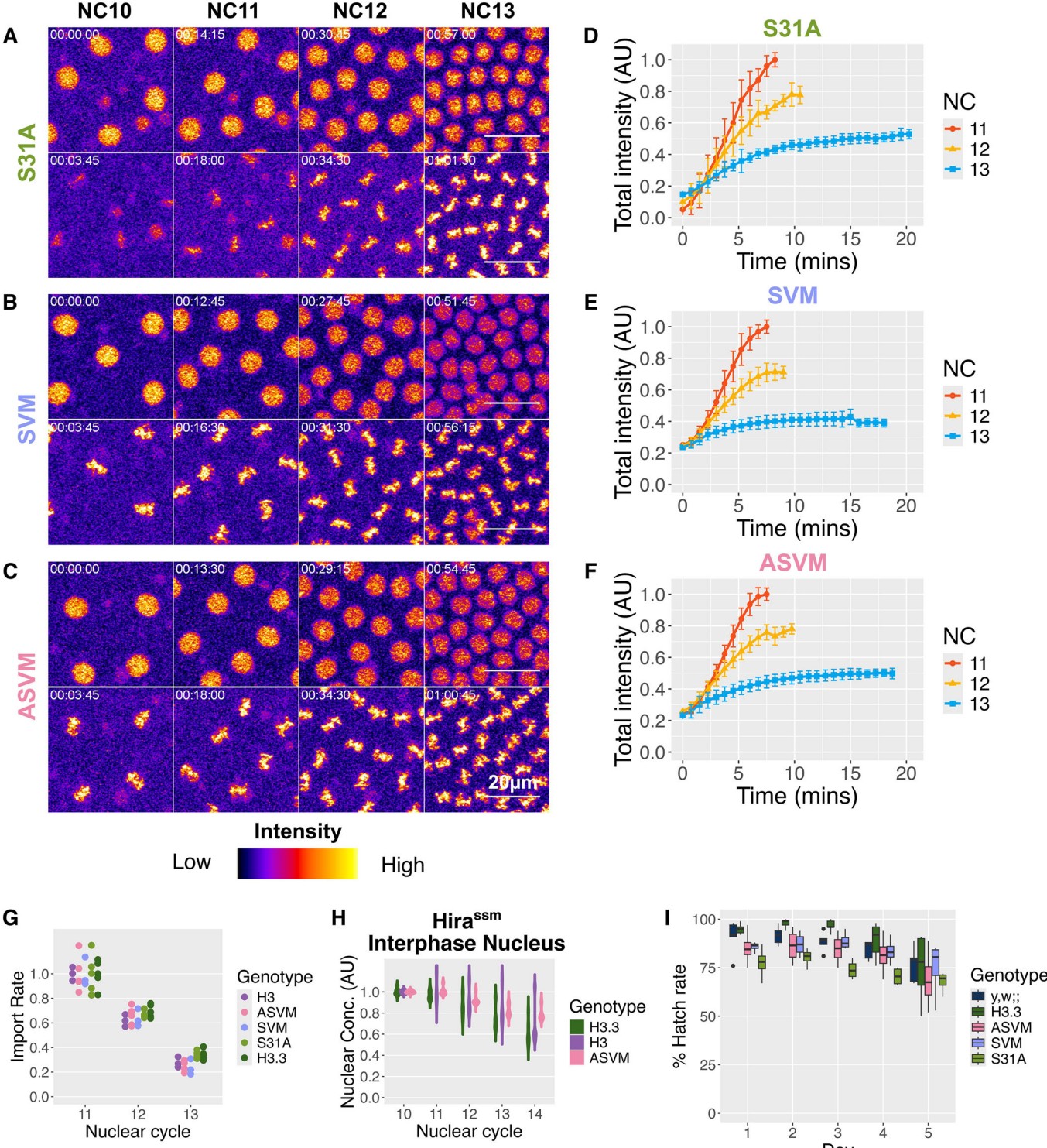

◀

**Figure EV2. Representative images of H3/H3.3 chimeras, import curves.**

(A) Representative maximum intensity projections of H3.3$^{S31A}$-Dendra2 during interphase and mitosis over NC10-13: interphase nuclei (top) and mitotic chromatin (bottom). Images are pseudo-colored with nonlinear look-up tables such that purple indicates low intensities and yellow indicates high intensities. H3.3$^{S31A}$ behaves similarly to H3.3. (B) Representative maximum intensity projections of H3.3$^{SVM}$-Dendra2 during interphase and mitosis over NC10-13: interphase nuclei (top) and mitotic chromatin (bottom). H3.3$^{SVM}$ behaves similarly to H3. (C) Representative maximum intensity projections of H3.3$^{ASVM}$-Dendra2 during interphase and mitosis over NC10-13: interphase nuclei (top) and mitotic chromatin (bottom). H3.3$^{ASVM}$ behaves similarly to H3. Data from embryos in A-C are quantified in Fig. 3. Scale bar 20 μm. (D–F) Total pixel intensities over time for NC11-13 normalized to the maximum NC11 values for H3.3$^{S31A}$-Dendra2 (D), H3.3$^{SVM}$-Dendra2 (E), and H3.3$^{ASVM}$-Dendra2 (F). H3.3$^{S31A}$-Dendra2 import is similar to H3.3-Dendra2, and only slows after 5 min without plateauing. H3.3$^{SVM}$-Dendra2 and H3.3$^{ASVM}$-Dendra2 import in a similar manner to H3-Dendra2 and plateau after 5 min. The solid line represents the mean, and the error bars represent the standard deviation. (G) The initial slopes of nuclear import curves of chimeras are shown in (D–F) for NC11-13. H3-Dendra2 and H3.3-Dendra2 slopes from S1E are included for reference. All slopes are normalized to NC11 values. (H) Average interphase nuclear intensities of H3.3-Dendra2 (green), H3-Dendra2 (purple), and H3.3$^{ASVM}$-Dendra2 (pink) in Hira$^{ssm}$ embryos 45 s before the NEB in NC10-14, normalized to their average intensities in NC10. Though H3.3 is not incorporated, it is imported into the nucleus, and its concentrations reduce with each cycle. H3 nuclear concentrations also drop with each cycle. However, H3.3$^{ASVM}$ concentrations are relatively more stable over the cycles. ($n = 5$ all chimeras, 3 H3 ssm, 4 H3.3 ssm, and 5 H3.3$^{ASVM}$ ssm embryos.) (I) Five-day hatch rates of the H3/H3.3 replacement chimeras compared to control (y,w;;) and H3.3-Dendra2. ($n = 4$ sets of egg-laying cages for each genotype. Statistical comparisons for (G, I) can be found in Appendix Tables S8, 9).

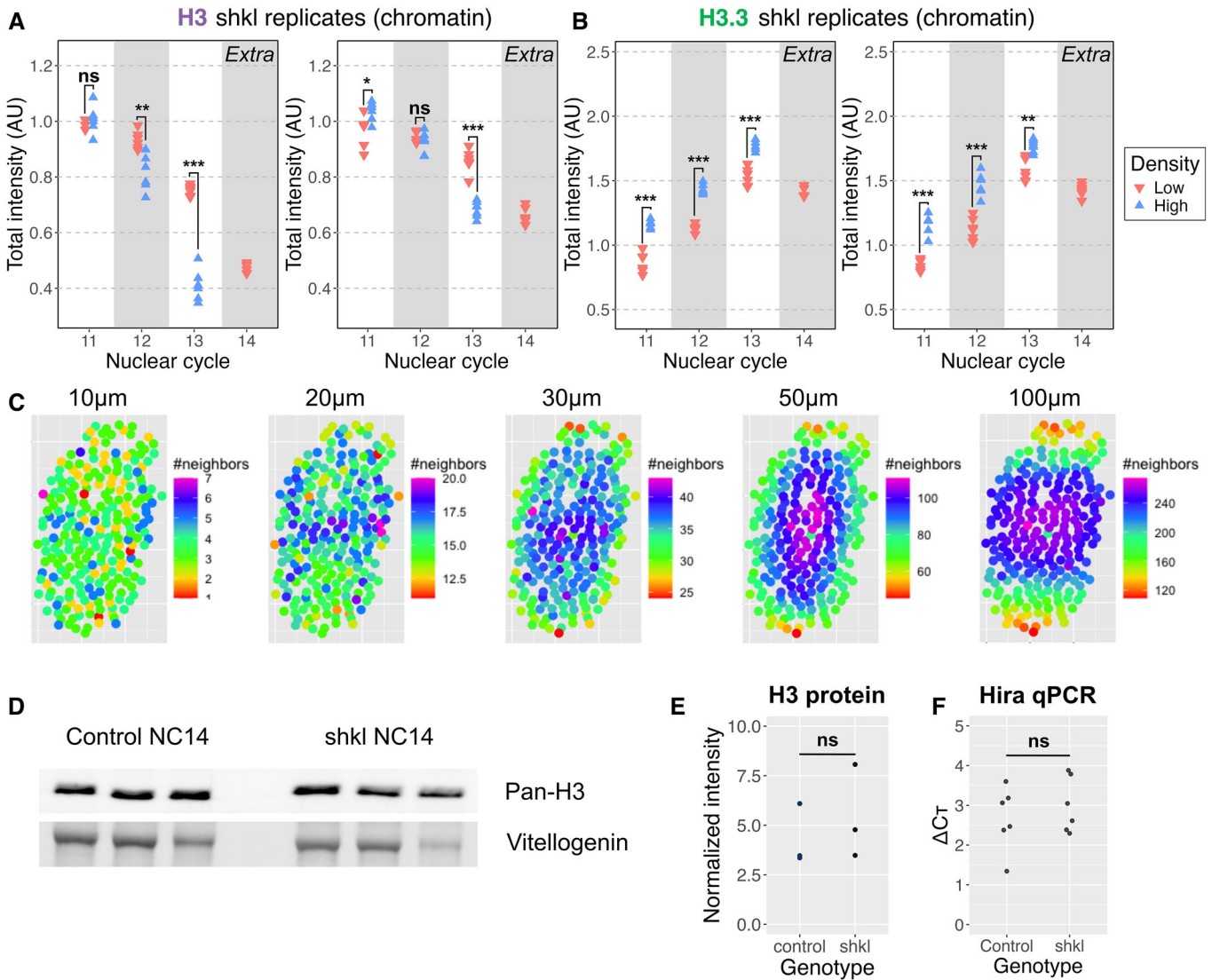

**Figure EV3. shkl embryos consistently respond to the N/C ratio without changes in total H3 loading.**

(A) Replicate shkl embryos of the same genotype as shown in Fig. 4E demonstrate that H3-Dendra2 is consistently retained on mitotic chromatin by nuclei at low-density regions compared to high-density regions within the same cell cycle over NC11-14. (Embryo 1: $p = 0.99$ (NC11), 1.63e-03 (NC12), <e-15 (NC13), Embryo 2: $p = 4.26e-02$ (NC11), 0.99 (NC12), 2.69e-09 (NC13)). (B) Replicate shkl embryos of the same genotype as shown in Fig. 4F demonstrate that H3.3-Dendra2 incorporation is reduced on mitotic chromatin by nuclei at low-density regions compared to high-density regions within the same cell cycle over NC11-14. (Embryo 1: $p = 2.01e-09$ (NC11), 3.67e-10 (NC12), 5.19e-07 (NC13), Embryo 2: $p = 1.21e-06$ (NC11), 6.48e-10 (NC12), 1.53e-03 (NC13)) (C) Example control embryo in which the radius used to determine the number of neighbors was varied from 10 to 100 μm, as shown. A 20 μm radius was deemed optimal for neighborhood analysis as it enabled us to accurately capture the gradient observed in the shkl embryos while excluding edge effects due to embryo curvature. (D) Western blot against a Pan-H3 antibody in control and shkl NC14 embryos. Stainfree signal for Vitellogenin (~45 kDa) is used as a loading control. (E) Quantification of (D). Pan-H3 levels are not significantly different between control and shkl embryos. (F) RT-qPCR results for Hira mRNA in control and shkl NC14 single embryos. Differences in $\Delta C_T$ values are not significant. (Statistical significance was determined by one-way/two-way ANOVA, ns= $p > 0.05$, *$p < 0.05$, **$p < 0.01$, ***$p < 0.001$).

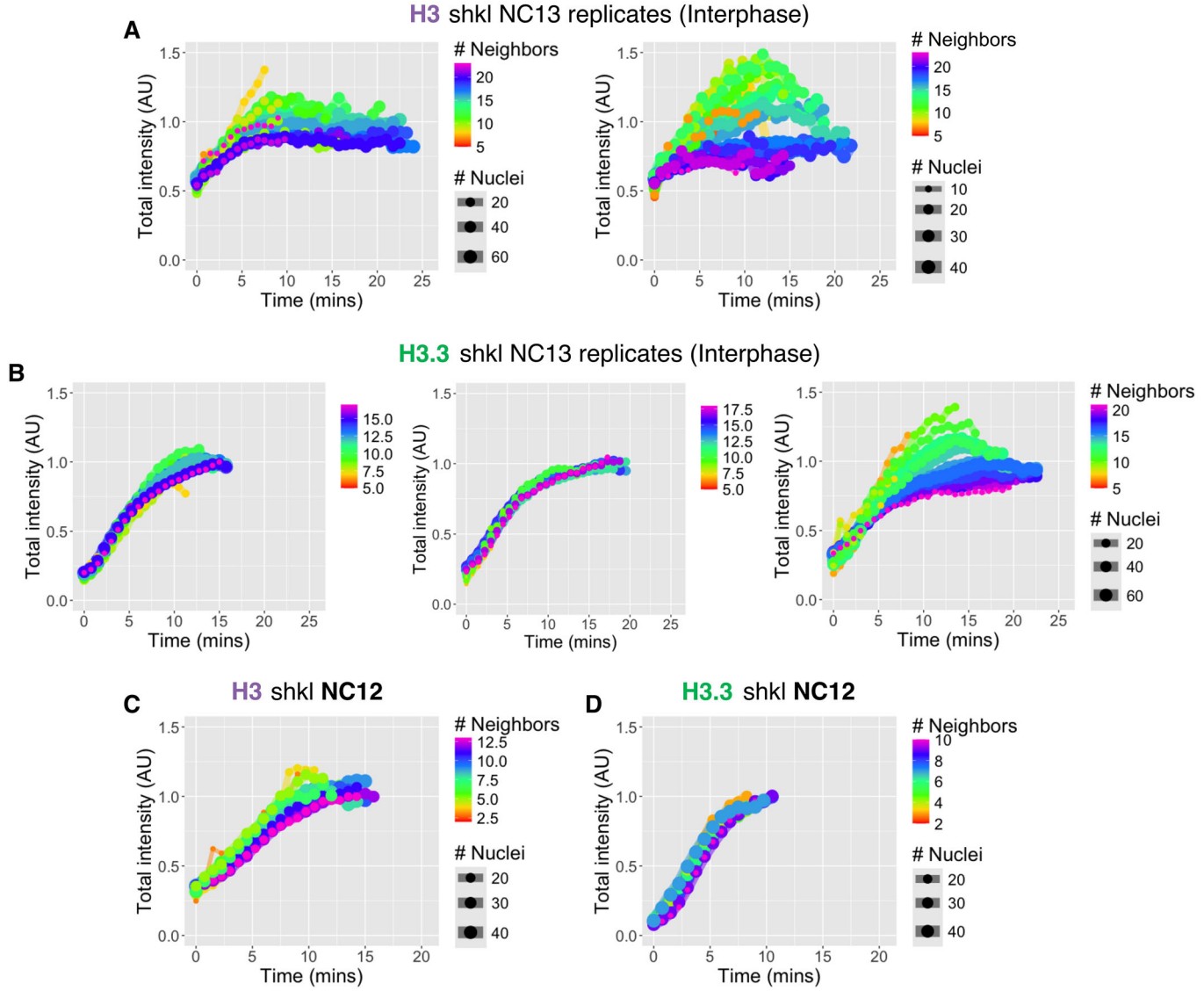

**Figure EV4. Replicate shkl embryos demonstrate consistent effects.**

(A) Replicate shkl embryos of the same genotype as shown in Fig. 5F demonstrate that nuclear import and accumulation of H3 inversely correlate with the number of neighbors surrounding a given nucleus, suggesting H3 nuclear import is N/C ratio sensitive. (B) Replicate shkl embryos of the same genotype as shown in Fig. 5G demonstrate that nuclear import and accumulation of H3.3 is less N/C ratio sensitive than H3 in most cases. (C, D) Total intensities over time for H3-Dendra2 (C) and H3.3-Dendra2 (D) in NC12 shkl embryos indicate that the trends observed in NC13 begin in NC12, though to a lesser extent. These data were taken from the same embryo that was used in Fig. 5F,G, respectively.

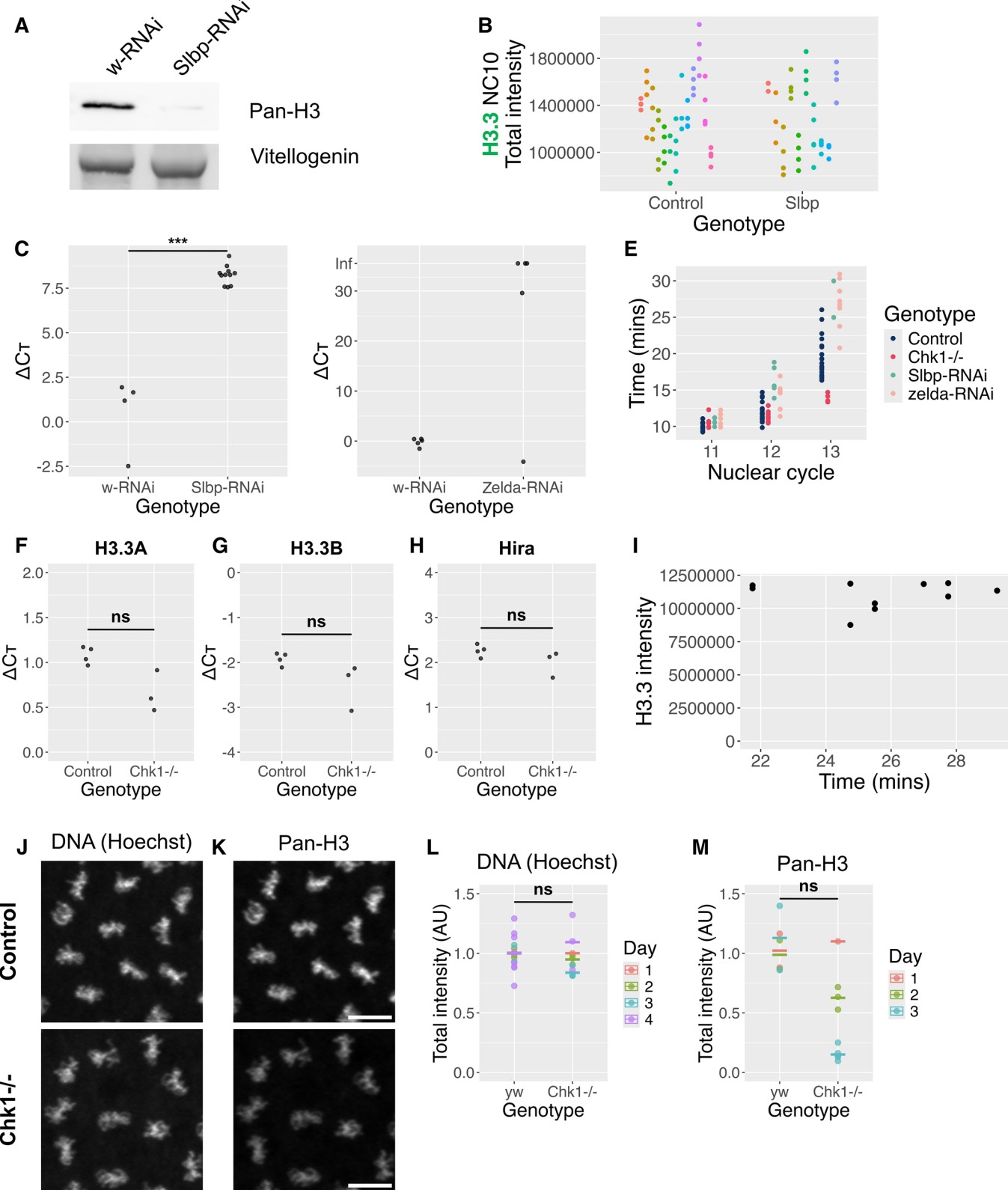

**Figure EV5. Quantification of histone composition and cell cycle times in RNAi and mutant embryos.**

(A) Western blot against a Pan-H3 antibody in w-RNAi (control) and Slbp-RNAi embryos after a 1 h collection. Stainfree signal for Vitellogenin (~45 kDa) is used as a loading control. Slbp-RNAi causes H3 knockdown. (B) H3.3 incorporation in NC10 mitotic chromatin in w-RNAi (control) and severely affected Slbp embryos, which do not survive past NC11 due to mitotic defects. H3.3 levels are comparable between w-RNAi (control) and Slbp embryos. Different colors represent nuclei from different embryos. (C) RT-qPCR results for Slbp mRNA in w-RNAi (control) and Slbp-RNAi single embryos from a 2 h collection. Slbp mRNA is significantly reduced (p = 2.04e-08) in RNAi embryos. (D) RT-qPCR results for Zelda mRNA in w-RNAi (control) and Zelda-RNAi single embryos from a 1 h collection. Zelda mRNA is dramatically reduced in RNAi embryos, often undetectable. (E) Cell cycle durations of control (a mix of y,w and w-RNAi) and RNAi/mutant embryos used in Fig. 6. Note that cell cycles are shortened in Chk1$^{-/-}$ (*H3.3-Dendra2, grp$^1$*) embryos and lengthened in Slbp-RNAi and Zelda-RNAi embryos. (F–H) RT-qPCR results for H3.3 A (F), H3.3B (G), and Hira (H) mRNA in control (H3.3-Dendra2) and Chk1$^{-/-}$ (H3.3-Dendra2, grp$^1$) mutant NC12 single embryos. Differences in $\Delta C_T$ values are not significant. (I) H3.3 chromatin incorporation versus NC13 duration in control (H3.3-Dendra2) embryos. No correlation is observed between NC13 cell cycle duration and H3.3 incorporation. (J, K) DNA (J) and Pan-H3 (K) staining in control (y,w) and Chk1$^{-/-}$ mutant NC12 embryos. Scale bar 10 µm. (L) Quantification of J. DNA amounts are unchanged in the Chk1$^{-/-}$ mutant. (M) Quantification of K. Pan-H3 stain is more variable in the Chk1$^{-/-}$ mutant. Each point indicates the average amount in a single embryo, and solid lines indicate average amounts from all the embryos of a single day. Day averages were used to perform one-way ANOVA significance tests. (n ≥ 3 embryos. Statistical significance was determined by one-way/two-way ANOVA, ns= p > .05, *p < .05, **p < 0.01, ***p < 0.001. Statistical comparisons for (E) can be found in the Appendix Table S10).

