## [Peer Review File · EMBO Reports]

Local nuclear to cytoplasmic ratio regulates H3.3 incorporation via cell cycle state during zygotic genome activation

Anusha Bhatt, Madeleine Brown, Aurora Wackford, Yuki Shindo, and Amanda Amodeo

Corresponding author(s): Amanda Amodeo (amanda.a.amodeo@dartmouth.edu)

Review Timeline:

Transfer Date:	11th Mar 25
Editorial Decision:	20th Mar 25
Revision Received:	26th Jul 25
Editorial Decision:	10th Sep 25
Revision Received:	14th Sep 25
Accepted:	26th Sep 25

Editor: Esther Schnapp / Martina Rembold

**Transaction Report: This manuscript was transferred to
EMBO reports following peer review at Review Commons.**

**Review
COMMONS**

Review #1

1. Evidence, reproducibility and clarity:

Evidence, reproducibility and clarity (Required)

****Summary:****

Bhatt et al. seek to define factors that influence H3.3 incorporation in the embryo. They test various hypotheses, pinpointing the nuclear/cytoplasmic ratio and Chk1, which affects cell cycle state, as influencers. The authors use a variety of clever *Drosophila* genetic manipulations in this comprehensive study. The data are presented well and conclusions reasonably drawn and not overblown. I have only minor comments to improve readability and clarity. I suggest two OPTIONAL experiments below.

****Major comments:****

We found this manuscript well written and experimentally thorough, and the data are meticulously presented. We have one modification that we feel is essential to reader understanding and one experimental concern:

The authors provide the photobleaching details in the methodology, but given how integral this measurement is to the conclusions of the paper, we feel that this should be addressed in clear prose in the body of the text. The authors explain briefly how nuclear export is assayed, but not import (line 99). Would help tremendously to clarify the methods here. This is especially important as import is again measured in Fig 4. This should also be clarified (also in the main body and not solely in the methods).

If the embryos appeared "reasonably healthy" (line 113) after slbp RNAi, how do the authors know that the RNAi was effective, especially in THESE embryos, given siblings had clear and drastic phenotype? This is especially critical given that the authors find no effect on H3.3 incorporation after slbp RNAi (and presumably H3 reduction), but this result would also be observed if the slbp RNAi was just not effective in these embryos.

****Minor comments:****

Introduction:

Consider using "replication dependent" (RD) rather than "replication coupled." Both are used in the field, but RD parallels RI ("replication independent").

Would help for clarity if the authors noted that H3 is equivalent to H3.2 in *Drosophila*. Also it is relevant that there are two H3.3 loci as the authors knock mutations into the H3.3A locus, but leave the H3.3B locus intact. The authors should clarify that there are two H3.3 genes in the *Drosophila* genome.

Please add information and citation (line 58): H3.3 is required to complete development when H3.2 copy number is reduced (PMID: 37279945, McPherson et al. 2023)

Results:

Embryo genotype is unclear (line 147): Hira[*ssm*] haploid embryos inherit the Hira mutation maternally? Are Hira homozygous mothers crossed to homozygous fathers to generate these embryos, or are mothers heterozygous? This detail should be in the main text for clarity.

Line 161: Shkl affects nuclear density, but it also appears from Fig 3 to affect nuclear size? The authors do not address this, but it should at least be mentioned.

The authors often describe nuclear H3/H3.3 as chromatin incorporated, but these image-based methods do not distinguish between chromatin-incorporated and nuclear protein. I very much appreciate how the authors laid out their model in Fig 3 and then used the same figure to explain which part of the model they are testing in Figs 4 and 5. This is not a critique- we can complement too!

OPTIONAL experimental suggestion: The experiments in Figure 4 and 5 are clever. One would expect that H3 levels might exhaust faster in embryos lacking all H3.2 histone genes (Gunesdogan, 2010, PMID: 20814422), allowing a comparison testing the H3 availability > H3.3 incorporation portion of the hypothesis without manipulating the N/C ratio. This might also result in a more consistent system than *slbp* RNAi (below).

O'Haren 2024 (PMID: 39661467) did not find increased Pol II at the HLB after *zelda* RNAi (line 227). Might also want to mention here that *zelda* RNAi does not result in changes to H3 at the mRNA level (O'Haren 2024), as that would confound the model.

Discussion:

Should discuss results in context of McPherson et al. 2023 (PMID: 37279945), who showed that decreasing H3.2 gene numbers does not increase H3.3 production at the mRNA or protein levels.

The Shackleton mutation is a clever way to alter N/C ratio, but the authors should point out that it is difficult (impossible?) to directly and cleanly manipulate the N/C ratio. For example, *Shkl* mutants seem to also have various nuclear sizes.

How is H3.3 expression controlled? Is it possible that H3.3 biosynthesis is affected in *Chk1*

mutants?

Figures:

While I appreciate the statistical summaries in tables, it is still helpful to display standard significance on the figures themselves.

Fig 1:

A: Is it possible to label panels with the nuclear cycle?

B: Statistics required - caption suggests statistics are in Table S2, but why not put on graph?

C/D: Would be helpful if authors could plot H3/H3.3 on same graph because what we really need to compare is NC13 between H3/H3.3 (and statistics between these curves)

E: The comparison in the text is between H3.3 and H3, but only H3.3 data is shown. I realize that it is published prior, but the comparison in figure would be helpful.

Fig 2:

A: A very helpful figure. Slightly unclear that the H3 that is not Dendra tagged is at the H3.3 locus. Also unclear that the H3.3A-Dendra2 line exists and used as control, as is not shown in figure.

Should show H3 and H3.3 controls (Figure S2)

F/H- As the comparison is between H3 and ASVM, it would help to combine these data onto the same graph. As the color is currently used unnecessarily to represent nuclear cycle, the authors could use their purple/pink color coding to represent H3/ASVM.

In the legend of Fig 2 the authors write "in the absence of Hira." Technically, there is only a point mutation in Hira. It is not absent.

Fig 3:

G: Please show WT for comparison. Can use data in Fig 3A.

Model in H is very helpful (complement)!

Fig 4:

B/C/F/G: The authors use a point size scale to represent the number of nuclei, but the graphs are so overlaid that it is not particularly useful. Is there a better way to display this dimension?

D/E/H/I: What does "min volume" mean on the X axis?

Fig 5:

F: OPTIONAL Experimental request: Here I would like to see H3 as a control.

2. Significance:

Significance (Required)

General assessment: Many long-standing mysteries surround zygotic genome activation, and here the authors tackle one: what are the signals to remodel the zygotic chromatin around ZGA? This is a tricky question to answer, as basically all manipulations done to the embryo have widespread effects on gene expression in general, confounding any conclusions. The authors use clever novel techniques to address the question. Using photoconvertible H3 and H3.3, they can compare the nuclear dynamics of these proteins after embryo manipulation. Their model is thorough and they address most aspects of it. The hurdle this study struggles to overcome is the same that all ZGA studies have, which is that manipulation of the embryo causes cascading disasters (for example, one cannot manipulate the nuclear:cytoplasmic ratio without also altering cell cycle timing), so it's challenging to attribute molecular phenotypes to a single cause. This doesn't diminish the utility of the study.

Advance: The conceptual advance of this study is that it implicates the nuclear:cytoplasmic ratio and Chk1 in H3.3 incorporation. The authors suggest these factors influence cell cycle closing, which then affects H3.3 incorporation, although directly testing the granularity of this model is beyond the scope of the study. The authors also provide technical advancement in their use of measuring histone dynamics and using changes in the dynamics upon treatment as a useful readout. I envision this strategy (and the dendra transgenes) to be broadly useful in the cell cycle and developmental fields.

Audience: The basic research presented in this study will likely attract colleagues from the cell cycle and embryogenesis fields. It has broader implications beyond Drosophila and even zygotic genome activation.

This reviewer's expertise: Chromatin, Drosophila, Gene Regulation

3. How much time do you estimate the authors will need to complete the suggested revisions:

Estimated time to Complete Revisions (Required)

(Decision Recommendation)

Between 1 and 3 months

4. Review Commons values the work of reviewers and encourages them to get credit for their work. Select 'Yes' below to register your reviewing activity at Web of Science Reviewer Recognition Service (formerly Publons); note that the content of your review will not be visible on Web of Science.

Yes

Review #2

1. Evidence, reproducibility and clarity:

Evidence, reproducibility and clarity (Required)

This manuscript investigates the regulation of H3.3 incorporation during zygotic genome activation (ZGA) in *Drosophila*, proposing that the nuclear-to-cytoplasmic (N/C) ratio plays a central role in this process. While the study is conceptually interesting, several concerns arise regarding the lack of proper control experiments and the clarity of the writing. The manuscript is difficult to follow due to vague descriptions, insufficient distinctions between established knowledge and novel findings, and a lack of rigorous statistical analyses. These issues need to be addressed before the study can be considered for publication.

****Major Concerns****

The manuscript would benefit from a clearer introduction that explicitly distinguishes between previously known mechanisms of histone regulation during ZGA and the novel contributions of this study. Currently, the introduction lacks sufficient background on early embryonic chromatin regulation, making it difficult for readers unfamiliar with the field to grasp the significance of the findings. The authors should also be more precise when discussing the timing of ZGA. While they state that ZGA occurs after 13 nuclear divisions, it is well established that a minor wave of ZGA begins at nuclear cycle 7-8, whereas the major wave occurs after cycle 13. Clarifying this distinction will improve the manuscript's accessibility to a broader audience.

One of the primary weaknesses of this study is the lack of adequate control experiments. In Figure 1, the authors suggest that the levels of H3 and H3.3 are influenced by the N/C ratio, but it is unclear whether transcription itself plays a role in these dynamics. To properly test this, RNA-seq or Western blot analyses should be performed at nuclear cycles 10 and 13-14 to compare the levels of newly transcribed H3 or H3.3 against maternally supplied histones. Without such data, the authors cannot rule out transcriptional regulation as a contributing factor.

In Figure 2, the manuscript introduces chimeric embryos expressing modified histone variants, but their developmental viability is not addressed. It is essential to determine whether these embryos survive and whether they exhibit any phenotypic consequences such as altered hatching rates, defects in nuclear division, or developmental arrest. Moreover, given that H3.3 is associated with actively transcribed genes, an RNA-seq analysis of chimeric embryos should be included to assess transcriptional changes linked to H3.3 incorporation.

Figures 3 and 4 raise additional concerns about whether histone cluster transcription is altered in *shkl* mutant embryos. The authors propose that the *shkl* mutation affects the N/C ratio, yet it remains unclear whether this leads to changes in the transcription of histone clusters. Furthermore, since HIRA is a key chaperone for H3.3, it would be important to assess whether its levels or function are compromised in *shkl* mutants. To address these gaps, RT-qPCR or RNA-seq should be performed to quantify histone cluster transcription, and Western blot analysis should be used to determine if HIRA protein levels are affected. A similar issue arises in Figure 5, where the authors claim that H3.3 incorporation is dependent on cell cycle state but do not sufficiently test whether this is linked to changes in HIRA levels. Given the importance of HIRA in H3.3 deposition, its levels should be examined in *Slbp*, *Zelda*, and *Chk1* RNAi embryos to verify whether changes in H3.3 incorporation correlate with HIRA function. Without this, it is difficult to conclude that the observed effects are strictly due to cell cycle regulation rather than histone chaperone dynamics.

Several figures require additional statistical analyses to support the claims made. In Figure 1B, statistical testing should be included to validate the reported differences. Figure 1C-D states that "H3.3 accumulation reduces more slowly than H3," yet there is no quantitative comparison to substantiate this claim. Similarly, Figure 1E presents the conclusion that "These changes in nuclear import and incorporation result in a less dramatic loss of the free nuclear H3.3 pool than previously seen for H3," despite the fact that H3 data are not included in this figure. The conclusions drawn from these data need to be supported with appropriate statistical comparisons and more precise descriptions of what is being measured.

Figure 2 presents additional concerns regarding data interpretation. The comparisons between H3.3 and H3.3S31A to H3 and H3.3SVM/ASVM lack statistical analysis, making it difficult to determine the significance of the observed differences. The disappearance of H3.3 from mitotic chromosomes in Figure 2E is also not explained. If this phenomenon is functionally relevant, the authors should provide a mechanistic interpretation, or at the very least, discuss potential explanations in the text. In Figures 2F-H, the reasoning behind comparing the nuclear intensity of H3.3 to H3 in Hira mutants is unclear. To properly assess the role of HIRA in H3.3 chromatin accumulation, a more appropriate comparison would be between wild-type H3.3 and H3.3 levels in Hira knockdown embryos.

A broader concern is that the authors only test HIRA as a histone chaperone but do not consider alternative chaperones that could influence H3.3 deposition. Since multiple chaperone systems regulate histone incorporation, it would strengthen the conclusions if additional chaperones were tested.

Additionally, the manuscript does not include any validation of the RNAi knockdown efficiencies used throughout the study. This raises concerns about whether the observed phenotypes are truly due to target gene depletion or off-target effects. RT-qPCR or Western blot analyses should be performed to confirm knockdown efficiency.

Finally, the section discussing "H3.3 incorporation depends on cell cycle state, but not cell cycle duration" is unclear. The term "cell cycle state" is vague and should be explicitly defined. Does this refer to a specific phase of the cell cycle, changes in chromatin accessibility, or another regulatory mechanism?

2. Significance:

Significance (Required)

This manuscript investigates the regulation of H3.3 incorporation during zygotic genome activation (ZGA) in *Drosophila*, proposing that the nuclear-to-cytoplasmic (N/C) ratio plays a central role in this process. While the study is conceptually interesting, several concerns arise regarding the lack of proper control experiments and the clarity of the writing. The manuscript is difficult to follow due to vague descriptions, insufficient distinctions between established knowledge and novel findings, and a lack of rigorous statistical analyses. These issues need to be addressed before the study can be considered for publication.

3. How much time do you estimate the authors will need to complete the suggested revisions:

Estimated time to Complete Revisions (Required)

(Decision Recommendation)

More than 6 months

4. Review Commons values the work of reviewers and encourages them to get credit for their work. Select 'Yes' below to register your reviewing activity at Web of Science Reviewer Recognition Service (formerly Publons); note that the content of your review will not be visible on Web of Science.

Yes

Review #3

1. Evidence, reproducibility and clarity:

Evidence, reproducibility and clarity (Required)

****Summary:****

Based on previous findings of the changing ratios of histone H3 to its variant H3.3, the authors test how H3.3 incorporation into chromatin is regulated for ZGA. They demonstrate here that H3 nuclear availability drops and replacement by H3.3 relies on chaperone binding, though not on its typical chaperone Hira. Furthermore, they show that nuclear-cytoplasmic (N/C) ratios can influence this histone exchange likely by influencing cell cycle state.

****Major comments:****

1. The claims are largely supported by the data but I think a couple more experiments could help bolster the claims about cell cycle and chk1 regulation.

a. Creating a phosphomimetic of the chk1 phosphorylation site on H3.3 to see if it can overcome the defects seen in chk1 mutants

b. Assessing heterochromatin of embryos without chk1 (or ASVM mutants) for example, by looking at H3K9me3 levels

The first experiments could take several months if the flies haven't already been generated by the authors but the second should be quicker.

2. It would also be interesting to see what the health of the flies with some mutations in this paper are beyond the embryo stage if they are viable (e.g., development to adulthood, fertility etc.)

a. the SVM, ASVM mutations

b. the hira + ASVM mutations

The authors might already have this data but if not they have the flies and it shouldn't take long to get these data.

3. In the discussion section, can the authors speculate on how they think H3.3 ASVM is getting incorporated if not through Hira. Are there other known H3 variant chaperones, or can the core histone chaperone substitute?

****Minor comments:****

While the paper is well written, I found the figures very confusing and difficult to interpret. Comments here are meant to make it easier to interpret.

1. Fig 1 and most of the paper would benefit from a schematic of early embryo transitions labelled with time and stages of cell cycle to make interpreting data easier

2. Fig 1- same green color is used for nuclear cycle 12 and for H3.3 making it confusing when reading graphs. Please check other figures where there is a similar use of color for two different things

3. Fig 1C,D might benefit more from being split up into 3 graphs by cell cycle with H3 and H3.3 plotted on the same graphs rather than the way it is now

4. Line 130-133: can they also comment on the difference between SVM and ASVM. It seems like SVM might be even worse than ASVM (Fig 2C). Is this related to chk1 phosphorylation?

5. Fig 2F-G: It is very difficult to compare between histones when they are on different graphs, please consider putting H3, H3.3 and H3.3ASVM in a hirasm background on the same graph.

6. Fig 3- move G to become A and then have A and B.

7. The initial slope graphs in 4D, E, H and I are not easy to understand and would benefit from an explanation in the legend.

2. Significance:

Significance (Required)

This paper addresses an important and understudied question- how do histones and their variants mediate chromatin regulation in the early embryo before zygotic genome activation? The authors follow up on some previous findings and provide new insights using clever genetics and cell biology in *Drosophila melanogaster*. However, the authors do not directly look at chromatin structural changes using existing genomic tools. This may be beyond the scope of this work but would make for a nice addition to strengthen their claims if they can implement these chromatin accessibility techniques in the early embryo.

Histones affect a majority of biological processes and understanding their role in the early embryo is key to understanding development. I believe this study applies to a broad audience interested in basic science. However, I do think the authors might benefit from a more broad discussion of their results to attract a broad readership.

3. How much time do you estimate the authors will need to complete the suggested revisions:

Estimated time to Complete Revisions (Required)

(Decision Recommendation)

Between 1 and 3 months

Yes

Revision Plan

Manuscript number: RC-2024-02832

Corresponding author(s): Amanda Amodeo

[The “revision plan” should delineate the revisions that authors intend to carry out in response to the points raised by the referees. It also provides the authors with the opportunity to explain their view of the paper and of the referee reports.]

The document is important for the editors of affiliate journals when they make a first decision on the transferred manuscript. It will also be useful to readers of the reprint and help them to obtain a balanced view of the paper.

*If you wish to submit a full revision, please use our "Full Revision" template. **It is important to use the appropriate template to clearly inform the editors of your intentions.**]*

1. General Statements [optional]

This section is optional. Insert here any general statements you wish to make about the goal of the study or about the reviews.

*We thank the reviewers for the time and care that went into their comments. We have provided our response in two forms. First, we have placed each of our responses to individual comments into the categories (planned revisions, completed revisions, not planned) as per the Review Commons template. Second, we have also provided the same comments in their original order at the end of the document so that the reviewers can find all of their own comments in one place. Throughout the response we use **black for reviewer** comments, **blue for our responses**, and **red for new text added to the manuscript**. We have also highlighted the changes that we have already included in the revised manuscript. We hope that this makes re-assessing the work as efficient as possible.*

2. Description of the planned revisions

Insert here a point-by-point reply that explains what revisions, additional experimentations and analyses are planned to address the points raised by the referees.

Reviewer #1

If the embryos appeared "reasonably healthy" (line 113) after slbp RNAi, how do the authors know that the RNAi was effective, especially in THESE embryos, given siblings had clear and drastic phenotype? This is especially critical given that the authors find no effect on H3.3 incorporation after slbp RNAi (and presumably H3 reduction), but this result would also be observed if the slbp RNAi was just not effective in these embryos.

Revision Plan

We apologize for the confusion caused by our word choice. The “healthy” slbp-RNAi embryos had measurable phenotypes consistent with histone depletion that we have reported previously (Chari et al, 2019) including cell cycle elongation and early cell cycle arrest (Figure S4D). However, they did not have the catastrophic mitosis observed in more severely affected embryos. We agree with the reviewer that a concern of this experiment is that the less severely affected embryos likely have more remaining RD histones including H3. To address this we also tested H3.3 incorporation in the embryos that fail to progress to later cell cycles in the cycles that we could measure. Even in these more severely affected embryos we were not able to detect a change in H3.3 incorporation relative to controls (lines 240-243 and Fig S4B). Unfortunately, it is impossible to conduct the ideal experiment, which would be a complete removal of H3 since this is incompatible with oogenesis and embryo survival.

To address this confusion we have added supplemental videos of control, moderately affected and severely affected SLBP-RNAi embryos as movies 3-5 and modified the text to read:

“All embryos that survive through at least NC12, had elongated cell cycles in NC12 and 60% arrested in NC13 as reported previously indicating the effectiveness of the knockdown (Figure S4C, Movie 3-5)³⁹. In these embryos, H3.3 incorporation is largely unaffected by the reduction in RD H3 (Figure 5B).” lines 236-240

Finally, to characterize the range of SLBP knockdown in the RNAi embryos we propose to do single embryo RT-qPCRs for SLBP mRNA for multiple individual embryos. This will provide a measure of the range knockdown that we observed in our H3.3 movies.

How is H3.3 expression controlled? Is it possible that H3.3 biosynthesis is affected in Chk1 mutants?

To address this question we propose to perform RT-qPCR for H3.3A and H3.3B as well as Hira in the Chk1 mutant. Unfortunately, we do not have antibodies that reliably distinguish between H3 and H3.3 in our hands (despite literature reports), but we will also perform a pan-H3 immunostaining in the Chk1 embryos to measure how the total H3-type histone pool changes as a result of the loss of Chk1.

Fig 5:

F: OPTIONAL Experimental request: Here I would like to see H3 as a control.

This is a very good suggestion, and we are currently imaging H3-Dendra2 in the Chk1 background. However, our preliminary results suggest that there may be some synthetic early lethality between the tagged H3-Dendra2 and Chk1 since these embryos are much less healthy than H3.3-Dendra2 Chk1 embryos or Chk1 with other reporters. In addition, we have observed a much higher level of background fluorescence in this cross than in the H3-Dendra2 control. We are uncertain if we will be able to obtain usable data from this experiment, but will continue to try to find conditions that allow us to analyze this data.

Revision Plan

As an orthogonal approach to answer the question, we will perform immunostaining with a pan-H3 antibody in Chk1 mutant embryos to measure total H3 levels under these conditions. Since the majority of H3-type histone is H3.2 and we know how H3.3 changes, this staining will give us insight into the dynamics of H3 in Chk1 mutant embryos.

Reviewer #2

In Figure 2, the manuscript introduces chimeric embryos expressing modified histone variants, but their developmental viability is not addressed. It is essential to determine whether these embryos survive and whether they exhibit any phenotypic consequences such as altered hatching rates, defects in nuclear division, or developmental arrest.

Tagging histones is often deleterious to organismal health. In Drosophila there are two H3.3 loci (H3.3A and H3.3B). In all of our chimera experiments we have left the H3.3B and one copy of the H3.3A locus unperturbed to provide a supply of untagged H3.3. This allows us to study H3.3 and chimera dynamics without compromising organism health. All of our chimeras are viable and fertile with no obvious morphological defects. We have added the following sentences to the text to clarify this point:

“There are ~100 copies of H3 in the Drosophila genome, but only 2 of H3.3 (H3.3A and H3.3B)²⁶. To determine which factor controls nuclear availability and chromatin incorporation, we genetically engineered flies to express Dendra2-tagged H3/H3.3 chimeras at the endogenous H3.3A locus, keeping the H3.3B locus intact....These chimeras were all viable and fertile.” lines 127-131, 136

In addition we propose performing hatch rate assays for embryos from the chimeric embryos of S31A, SVM and ASVM to assess if there is any decrease in fecundity due to the presence of the chimeras.

Figures 3 and 4 raise additional concerns about whether histone cluster transcription is altered in shkl mutant embryos. The authors propose that the shkl mutation affects the N/C ratio, yet it remains unclear whether this leads to changes in the transcription of histone clusters. Furthermore, since HIRA is a key chaperone for H3.3, it would be important to assess whether its levels or function are compromised in shkl mutants. To address these gaps, RT-qPCR or RNA-seq should be performed to quantify histone cluster transcription, and Western blot analysis should be used to determine if HIRA protein levels are affected.

The changes in the N/C ratio that are observed in the shkl mutant are within SINGLE embryo (differences in nuclear spacing). In these experiments we are comparing nuclei within a common cytoplasm that have different local nuclear densities (N/C ratios). Therefore, if Shkl were somehow affecting the transcription of histones or their chaperones we would expect all of the nuclei within the same mutant embryo to be equally affected since they are genetically

Revision Plan

identical and share a common cytoplasm. We do not directly compare the behavior of shkl embryos to wildtype except to demonstrate that there is no positional effect on the import of H3 and H3.3 across the length of the embryo in wildtype. To clarify our experimental system for these experiments we have added additional panels to Figure 4A and B that depict the number of neighbors for both control and Shkl embryos.

*Nonetheless, to address the reviewer's concern that shkl may change the amount of H3 present in the embryo, **we propose to conduct a western blot comparison of wildtype and shkl embryos using a pan-H3 antibody.** There are no tools (antibodies or fluorescently tagged proteins) to assess HIRA protein levels in Drosophila. We therefore **propose to perform RT-qPCR for Hira in wildtype and shkl embryos.***

A similar issue arises in Figure 5, where the authors claim that H3.3 incorporation is dependent on cell cycle state but do not sufficiently test whether this is linked to changes in HIRA levels. Given the importance of HIRA in H3.3 deposition, its levels should be examined in Slbp, Zelda, and Chk1 RNAi embryos to verify whether changes in H3.3 incorporation correlate with HIRA function. Without this, it is difficult to conclude that the observed effects are strictly due to cell cycle regulation rather than histone chaperone dynamics.

*Since H3.3 incorporation is unaffected in the Slbp and Zelda-RNAi lines there is no reason to suspect a change in Hira function. There are no available tools (antibodies or fluorescently tagged proteins) to directly measure Hira protein in Drosophila. To test if changes in HIRA loading might contribute to the decreased H3.3 incorporation in the Chk1 mutant **we propose to perform RT-qPCR for Hira in wildtype and Chk1 embryos.***

Additionally, the manuscript does not include any validation of the RNAi knockdown efficiencies used throughout the study. This raises concerns about whether the observed phenotypes are truly due to target gene depletion or off-target effects. RT-qPCR or Western blot analyses should be performed to confirm knockdown efficiency.

Both the Zelda and slbp-RNAi lines used for knockdowns have been used and validated in the early fly embryo in previously published works ((Yamada et al., 2019), (Duan et al., 2021), (O'Haren et al., 2025), (Chari et al, 2019)) and the phenotypes that we observe in our embryos are consistent with the published data including altered cell cycle durations (Figure S4C) and lack of cellularization/gastrulation. We note that the zelda RNAi phenotypes are also highly consistent with the effects of Zelda germline clones. To validate that slbp-RNAi knocks down histones we included a western blot for Pan-H3 in slbp-RNAi embryos that demonstrates a large effect on total H3 levels (Figure S4A). To further demonstrate the phenotypic effects of the slbp-RNAi we have added supplemental movies (Videos 4 and 5).

*To fully characterize the RNAi efficiency under our conditions **we propose to perform RT-qPCR for slbp in slbp-RNAi and Zelda in Zelda-RNAi compared to control (w) RNAi embryos.***

Revision Plan

Reviewer #3

Major comments:

1. The claims are largely supported by the data but I think a couple more experiments could help bolster the claims about cell cycle and chk1 regulation.

a. Creating a phosphomimetic of the chk1 phosphorylation site on H3.3 to see if it can overcome the defects seen in chk1 mutants

b. Assessing heterochromatin of embryos without chk1 (or ASVM mutants) for example, by looking at H3K9me3 levels

The first experiments could take several months if the flies haven't already been generated by the authors but the second should be quicker.

a. This is an excellent experimental suggestion which is bolstered by the fact that in frogs H3.3 S31A cannot rescue H3.3 morpholino during gastrulation, but H3.3S31D can (Sitbon et al, 2020). However, to correctly conduct this experiment would require generating and validating multiple additional endogenous H3.3 replacement lines, likely without a fluorescent tag as they can interfere with histone rescue constructs in most species. As the reviewer notes, this would take several months of work (we have not generated the critical flies yet) and may not yield a satisfying answer since there are reports that H3.3 may be dispensable in flies aside from as a source of H3-type histone outside of S-phase (Hödl and Bassler, 2012). While we hope to continue experiments along these lines in the future we feel that this is beyond the scope of the current manuscript.

b. To address this we propose to stain for H3K9me3 in wildtype and Chk1[±] embryos. Since the ASVM line is not a full replacement of all H3.3 we think that staining for H3K9me3 in this line is unlikely to yield a detectable difference.

2. It would also be interesting to see what the health of the flies with some mutations in this paper are beyond the embryo stage if they are viable (e.g., development to adulthood, fertility etc.)

a. the SVM, ASVM mutations

b. the hira + ASVM mutations

The authors might already have this data but if not they have the flies and it shouldn't take long to get these data.

a. To address this concern we propose to conduct hatch rate assays for embryos from the Dendra tagged H3.3, S31A, SVM, ASVM flies. However, we do note that in our experiments only one copy of the H3.3A locus was mutated and tagged with Dendra2 leaving one copy of H3.3A and both copies of H3.3B untouched to ensure normal development as tagging all copies of histone genes can lead to lethality.

Revision Plan

b. All Hira mutants develop as haploids due to the inability to decondense the sperm chromatin (which is dependent on Hira). This leads to one extra division to restore the N/C ratio prior to cell cycle slowing and ZGA. These embryos go on to gastrulate and die late in development after cuticle formation (presumably due to their decreased ploidy) (Loppin et al., 2000). The addition of ASVM into the Hira background does not appear to rescue the ploidy defect as these embryos also undergo the extra division (Figure 3H). We are therefore confident that these embryos will not hatch. We have added the information about the development of Hira mutant to the text as follow:

“These embryos develop as haploids and undergo one additional syncytial division before ZGA (NC14). Hira^{ssm}embryos develop otherwise phenotypically normally through organogenesis and cuticle formation, but die before hatching⁵⁷.” lines 164-167

3. Description of the revisions that have already been incorporated in the transferred manuscript

Please insert a point-by-point reply describing the revisions that were already carried out and included in the transferred manuscript. If no revisions have been carried out yet, please leave this section empty.

Reviewer #1

Evidence, reproducibility and clarity

Summary:

Bhatt et al. seek to define factors that influence H3.3 incorporation in the embryo. They test various hypotheses, pinpointing the nuclear/cytoplasmic ratio and Chk1, which affects cell cycle state, as influencers. The authors use a variety of clever Drosophila genetic manipulations in this comprehensive study. The data are presented well and conclusions reasonably drawn and not overblown. I have only minor comments to improve readability and clarity. I suggest two OPTIONAL experiments below.

We thank the reviewer for their positive and helpful comments.

Major comments:

We found this manuscript well written and experimentally thorough, and the data are meticulously presented. We have one modification that we feel is essential to reader understanding and one experimental concern:

The authors provide the photobleaching details in the methodology, but given how integral this measurement is to the conclusions of the paper, we feel that this should be addressed in clear prose in the body of the text. The authors explain briefly how nuclear export is assayed, but not import (line 99). Would help tremendously to clarify the methods here. This is especially important as import is again measured in Fig 4. This should also be clarified (also in the main body and not solely in the methods).

Revision Plan

We have added the following sentences to the main body of the text to clarify how photobleaching and import were assayed.

“We note that these differences are not due to photobleaching as our measurements on imaged and unimaged embryos indicate that photobleaching is negligible under our experimental conditions (see methods, Figure S1G-H)” lines 98-101

and

“Since nuclear export is effectively zero, we attribute the increase in total H3.3 over time solely to import and therefore the slope of total H3.3 over time corresponds to the import rate.” lines 111-113

In addition we have clarified how import was calculated to figure legends in Figure 5D (formerly 4D) and S1F which now read:

“Initial slopes of nuclear import curves (change in total nuclear intensity over time for the first 5 timepoints) ...”

We also added the following explanation of how nuclear import rates were calculated to the methods section:

“Import rates were calculated by using a linear regression for the total nuclear intensity over time for the first 5 timepoints in the nuclear import curves.” lines 471-473, methods

If the embryos appeared "reasonably healthy" (line 113) after slbp RNAi, how do the authors know that the RNAi was effective, especially in THESE embryos, given siblings had clear and drastic phenotype? This is especially critical given that the authors find no effect on H3.3 incorporation after slbp RNAi (and presumably H3 reduction), but this result would also be observed if the slbp RNAi was just not effective in these embryos.

We apologize for the confusion caused by our word choice. The “healthy” slbp-RNAi embryos had measurable phenotypes consistent with histone depletion that we have reported previously (Chari et al, 2019) including cell cycle elongation and early cell cycle arrest (Figure S4D). However, they did not have the catastrophic mitosis observed in more severely affected embryos. We agree with the reviewer that a concern of this experiment is that the less severely affected embryos likely have more remaining RD histones including H3. To address this we also tested H3.3 incorporation in the embryos that fail to progress to later cell cycles in the cycles that we could measure. Even in these more severely affected embryos we were not able to detect a change in H3.3 incorporation relative to controls (lines 240-243 and Fig S4B). Unfortunately, it is impossible to conduct the ideal experiment, which would be a complete removal of H3 since this is incompatible with oogenesis and embryo survival.

Revision Plan

To address this confusion we have added supplemental videos of control, moderately affected and severely affected SLBP-RNAi embryos as movies 3-5 and modified the text to read:

“All embryos that survive through at least NC12, had elongated cell cycles in NC12 and 60% arrested in NC13 as reported previously indicating the effectiveness of the knockdown (Figure S4C, Movie 3-5)³⁹. In these embryos, H3.3 incorporation is largely unaffected by the reduction in RD H3 (Figure 5B).” lines 236-240

Minor comments:

Introduction:

Consider using "replication dependent" (RD) rather than "replication coupled." Both are used in the field, but RD parallels RI ("replication independent").

We thank the reviewer for this suggestion. We have made the text edits to change "replication coupled" (RC) to "replication dependent" (RD) throughout the manuscript.

Would help for clarity if the authors noted that H3 is equivalent to H3.2 in Drosophila. Also it is relevant that there are two H3.3 loci as the authors knock mutations into the H3.3A locus, but leave the H3.3B locus intact. The authors should clarify that there are two H3.3 genes in the Drosophila genome.

We have changed the text as follows to increase clarity as suggested:

“Similarly, we have previously shown that RD H3.2 (hereafter referred to as H3) is replaced by RI H3.3 during these same cycles, though the cause remains unclear²⁹” lines 52-54

“There are ~100 copies of H3 in the Drosophila genome, but only 2 of H3.3 (H3.3A and H3.3B)²⁶. To determine which factor controls nuclear availability and chromatin incorporation, we genetically engineered flies to express Dendra2-tagged H3/H3.3 chimeras at the endogenous H3.3A locus, keeping the H3.3B locus intact.” lines 127-131

Please add information and citation (line 58): H3.3 is required to complete development when H3.2 copy number is reduced (PMID: 37279945, McPherson et al. 2023)

We have added the suggested information. The text now reads

“Nonetheless, H3.3 is required to complete development when H3.2 copy number is reduced⁵⁴.” lines 61-62

Results:

Embryo genotype is unclear (line 147): Hira[ssm] haploid embryos inherit the Hira mutation maternally? Are Hira homozygous mothers crossed to homozygous fathers to generate these embryos, or are mothers heterozygous? This detail should be in the main text for clarity.

Revision Plan

The Hira mutants are maternal effect. We crossed homozygous Hira^{ssm} females to their hemizygous Hira^{ssm} or FM7C brothers. However, the genotype of the male is irrelevant since the Hira phenotype prevents sperm pronuclear fusion and therefore there is no paternal contribution to the embryonic genotype. We have clarified this point in the text:

“We generated embryos lacking functional maternal Hira using Hira^{ssm-185b} (hereafter Hira^{ssm}) homozygous mothers which have a point mutation in the Hira locus⁵⁷.” lines 160-162

Line 161: Shkl affects nuclear density, but it also appears from Fig 3 to affect nuclear size? The authors do not address this, but it should at least be mentioned.

We thank the reviewer for the astute observation. More dense regions of the Shkl embryos do in fact have smaller nuclei. We believe that this is a direct result of the increased N/C ratio since nuclear size also falls during normal development as the N/C ratio increases. We have added a new figure 1 in which we more carefully describe the events of early embryogenesis in flies including a quantification of nuclear size and number in the pre-ZGA cell cycles (Figure 1C). We also note the correlation of nuclear size with nuclear density in the text:

“During the pre-ZGA cycles (NC10-13), the maximum volume that each nucleus attains decreases in response to the doubling number of nuclei with each division (Figure 1C).” lines 86-87

“To test this, we employed mutants in the gene Shackleton (shkl) whose embryos have non-uniform nuclear densities and therefore a gradient of nuclear sizes across the anterior/posterior axis (Figure 3A-B, Movie 1-2)⁵⁸.” lines 180-183

The authors often describe nuclear H3/H3.3 as chromatin incorporated, but these image-based methods do not distinguish between chromatin-incorporated and nuclear protein.

To distinguish between chromatin incorporated and nuclear free histone we have exploited the fact that histones that are not incorporated into DNA freely diffuse away from the chromatin mass during mitosis while those that are bound into nucleosomes remain on chromatin during this time. In our previous study we showed that H3-Dendra2 that is photoconverted during mitosis remains stably associated with the mitotic chromatin through multiple cell cycles (Shindo and Amodeo, 2019) strengthening our use of this metric. To help clarify this point as well as other methodological details we have added a new Figure 1B which documents the time points at which we make various measurements within the lifecycle of the nucleus. We also edited the text to read:

“We have previously shown that with each NC, the pool of free H3 in the nucleus is depleted and its levels on chromatin during mitosis decrease (Figure 1D, S1C-D)²⁹. In

Revision Plan

contrast, H3.3 mitotic chromatin levels increase during the same cycles (Figure 1D, S1C-D)²⁹.” lines 89-92

I very much appreciate how the authors laid out their model in Fig 3 and then used the same figure to explain which part of the model they are testing in Figs 4 and 5. This is not a critique—we can complement too!

Thank you!

O'Haren 2024 (PMID: 39661467) did not find increased Pol II at the HLB after zelda RNAi (line 227). Might also want to mention here that zelda RNAi does not result in changes to H3 at the mRNA level (O'Haren 2024), as that would confound the model.

We thank the reviewer for the suggestion. We have removed the discussion of Pol II localization and replaced it with the information about histone mRNA :

“zelda controls the transcription of the majority of Pol II genes during ZGA but disruption of zelda does not change RD histone mRNA levels⁶⁷⁻⁷⁰.” lines 249-251

Discussion:

Should discuss results in context of McPherson et al. 2023 (PMID: 37279945), who showed that decreasing H3.2 gene numbers does not increase H3.3 production at the mRNA or protein levels.

We expanded our discussion to include the following:

“Given the fact that H3.3 pool size does not respond to H3 copy number in other Drosophila tissues,⁵⁴ our results suggest that H3.3 incorporation dynamics are likely independent of H3 availability.” lines 278-280

The Shackleton mutation is a clever way to alter N/C ratio, but the authors should point out that it is difficult (impossible?) to directly and cleanly manipulate the N/C ratio. For example, Shkl mutants seem to also have various nuclear sizes.

As discussed above, we think that nuclear size is a direct response to the N/C ratio. We have added the following sentence to the discussion as well as a citation to a paper which discusses how the N/C ratio might contribute to nuclear import in early embryos to the discussion:

“This may be due to N/C ratio-dependent changes in nuclear import dynamics which may also contribute to the observed changes in nuclear size across the shkl embryo⁷⁵.” lines 307-309

Figures:

While I appreciate the statistical summaries in tables, it is still helpful to display standard significance on the figures themselves.

Revision Plan

We have added statistical comparisons in Figure 3 (formerly Figure 2). We do not feel that it is appropriate to directly compare the intensities of the H3-Dendra2 construct expressed from the pseudo-endogenous locus to the H3.3 and chimeric proteins expressed from the H3.3A locus as they were imaged using different settings. Although we plot H3 on the same graph as the other proteins to allow for ease of comparison of their trends over time it is not appropriate to directly compare their normalized intensities which including statistical tests would encourage. We have added a note to the legend of Figure 1 explaining this which reads:

“Note that statistical comparisons between the two Dendra2 constructs have not been done as they were expressed from different loci and imaged under different experimental settings.”

Fig 1:

A: Is it possible to label panels with the nuclear cycle? *We have done this.*

E: The comparison in the text is between H3.3 and H3, but only H3.3 data is shown. I realize that it is published prior, but the comparison in figure would be helpful.

We have added the previously published values to the text.

“These changes in nuclear import and incorporation result in a less complete loss of the free nuclear H3.3 pool (~70% free in NC11 to ~30% in NC13) than previously seen for H3 (~55% free in NC11 to ~20% in NC13)” lines 116-119

Fig 2:

A: A very helpful figure. Slightly unclear that the H3 that is not Dendra tagged is at the H3.3 locus. Also unclear that the H3.3A-Dendra2 line exists and used as control, as is not shown in figure.

Should show H3 and H3.3 controls (Figure S2)

We have edited the figure to add Dendra2 to all of the constructs and made clear the location of each construct including adding the landing site for H3-Dendra2. We have also cited Figure S1 in the legend which contains a more detailed diagram of the integration strategy.

F/H- As the comparison is between H3 and ASVM, it would help to combine these data onto the same graph. As the color is currently used unnecessarily to represent nuclear cycle, the authors could use their purple/pink color coding to represent H3/ASVM. *We have combined these data onto a single graph as requested and changed the colors appropriately. We have not added statistical comparisons to this graph as we again believe that they would be inappropriate.*

In the legend of Fig 2 the authors write "in the absence of Hira." Technically, there is only a point mutation in Hira. It is not absent.

Good catch! We have changed this to “in Hira^{ssm} mutants”.

Revision Plan

Fig 3:

G: Please show WT for comparison. Can use data in Fig 3A. *We have added the color-coded number of neighbor embryo representations for WT and Shkl embryos underneath the example embryo images in 4A-B (formerly 3A-B,G).*

Model in H is very helpful (complement)! *Thank you.*

Fig 4:

D/E/H/I: What does "min volume" mean on the X axis?

Since the uneven N/C ratio in the shkl embryos results in a wavy cell cycle pattern there is no single time point where we can calculate the number of neighbors for the whole embryo (since not all nuclei are in the same cell cycle at a given point). Therefore, we had to choose a criterion for when we would calculate the number of neighbors for each nucleus. We chose nuclear size as a proxy for nuclear age since nuclear size increases throughout interphase (see new figure 1B). So, the minimum volume is the newly formed nucleus in a given cell cycle. We also tested other timepoints for the number of neighbors (maximum nuclear volume, just before nuclear envelope breakdown and midway between these two points) and found similar results. We chose to use minimum volume in this paper because this is the time point when the nucleus is growing most quickly and nuclear import is at its highest. We have added the following explanation to the methods:

"For shkl embryos, as the nuclear cycles are asynchronous, nuclear divisions start at different timepoints within the same cell cycle and the nuclear density changes as the neighboring nuclei divide. Therefore, the total intensity traces were aligned to match their minimum volumes (as shown in Figure 1B) to T0." lines 485-488, methods

And the following detail to the figure legend:

*"...plotted by the number of nuclear neighbors at their minimum nuclear volume..."
Figure 5 legend*

We also added a depiction of the lifecycle of the nucleus in which we marked the minimum volume as the new Figure 1B.

Reviewer #2 (Evidence, reproducibility and clarity (Required)):

This manuscript investigates the regulation of H3.3 incorporation during zygotic genome activation (ZGA) in *Drosophila*, proposing that the nuclear-to-cytoplasmic (N/C) ratio plays a central role in this process. While the study is conceptually interesting, several concerns arise regarding the lack of proper control experiments and the clarity of the writing. The manuscript is difficult to follow due to vague descriptions, insufficient distinctions between established knowledge and novel findings, and a lack of rigorous statistical analyses. These issues need to be addressed before the study can be considered for publication.

Revision Plan

We thank the reviewers for their careful reading of this manuscript. We have sought to clarify the concerns regarding clarity through numerous text edits detailed below. We did include ANOVA analysis for all of the relevant statistical comparisons in the supplemental table. However, to increase clarity we have also added some statistical comparisons in the main figures. We note that we do not feel that it is appropriate to directly compare the intensities of the H3-Dendra2 construct expressed from the pseudo-endogenous locus to the H3.3 and chimeric proteins expressed from the H3.3A locus as they were imaged using different settings. Although we plot H3 on the same graph as the other proteins to allow for ease of comparison of their trends over time it is not appropriate to directly compare their normalized intensities which including statistical tests would encourage. We have added a note to the legend of the new Figure 1 explaining this which reads:

“Note that statistical comparisons between the two Dendra2 constructs have not been done as they were expressed from different loci and imaged under different experimental settings.”

Major Concerns

The manuscript would benefit from a clearer introduction that explicitly distinguishes between previously known mechanisms of histone regulation during ZGA and the novel contributions of this study. Currently, the introduction lacks sufficient background on early embryonic chromatin regulation, making it difficult for readers unfamiliar with the field to grasp the significance of the findings. The authors should also be more precise when discussing the timing of ZGA. While they state that ZGA occurs after 13 nuclear divisions, it is well established that a minor wave of ZGA begins at nuclear cycle 7-8, whereas the major wave occurs after cycle 13. Clarifying this distinction will improve the manuscript's accessibility to a broader audience.

We have added a new figure 1 to make the timing and nuclear behaviors of the embryo during ZGA in Drosophila more clear. We have also added information about how the chromatin changes during Drosophila ZGA in the following sentence:

“ In Drosophila, these changes include refinement of nucleosomal positioning, partitioning of euchromatin and heterochromatin and formation of topologically associated domains^{20-22,24}.” lines 39-41

We have clarified the major and minor waves of ZGA in the introduction and results by adding the following sentences to the introduction and results respectively:

“In most organisms ZGA happens in multiple waves but the chromatin undergoes extensive remodeling to facilitate bulk transcription during the major wave of ZGA (hereafter referred to as ZGA)^{18-20,22-25}..” lines 36-39

Revision Plan

“In Drosophila, ZGA occurs in 2 waves. The minor wave starts as early as the 7th cycle, while major ZGA occurs after 13 rapid syncytial nuclear cycles (NCs) and is accompanied by cell cycle slowing and cellularization (Figure 1A-B).” lines 83-85

We hope that these changes help to reduce confusion and make the paper more accessible. However, we are happy to add additional information if the reviewer can provide specific points which require further attention.

One of the primary weaknesses of this study is the lack of adequate control experiments. In Figure 1, the authors suggest that the levels of H3 and H3.3 are influenced by the N/C ratio, but it is unclear whether transcription itself plays a role in these dynamics. To properly test this, RNA-seq or Western blot analyses should be performed at nuclear cycles 10 and 13-14 to compare the levels of newly transcribed H3 or H3.3 against maternally supplied histones. Without such data, the authors cannot rule out transcriptional regulation as a contributing factor.

In the pre-ZGA cell cycles the vast majority of protein including histones is maternally loaded. Gunesdogan et al. (2010) showed that the zygotic RD histone cluster nulls survive past NC14 (well past ZGA) with no discernible defects indicating that maternal RD histone supplies are sufficient for normal development during the cell cycles under consideration. Therefore, new transcription of replication coupled histones is not needed for apparently normal development during this period. Moreover, we have done the western blot analysis using a Pan-H3 antibody as suggested by the reviewer in our previously published paper (Shindo and Amodeo, 2019 supplemental figure S3A-B) and found that total H3-type histone proteins only increase moderately during this period of development, nowhere near the rate of the nuclear doublings. We have added the following sentence to clarify this point.

“These divisions are driven by maternally provided components and the total amount of H3 type histones do not keep up with the pace of new DNA produced²⁹.” lines 88-89

We have also previously done RNA-seq on wild-type embryos (and those with altered maternal histone levels) (Chari et al 2019). In this RNA-seq (like most RNA-seq in flies) we used poly-A selection and therefore cannot detect the RD histone mRNAs (which have a stem-loop instead of a poly-A tail). We have plotted the mRNA concentrations for both H3.3 variants from that dataset below for the reviewers reference (we have not included this in the revised manuscript). The total H3.3 mRNA levels are nearly constant from egg laying (NC0- these are from unfertilized embryos) until after ZGA (NC14). These data combined with the westerns discussed above give us confidence that what we are observing is the partitioning of large pools of maternally provided histones with only a relatively small contribution of new histone synthesis.

Revision Plan

Several figures require additional statistical analyses to support the claims made. In Figure 1B, statistical testing should be included to validate the reported differences. Figure 1C-D states that "H3.3 accumulation reduces more slowly than H3," yet there is no quantitative comparison to substantiate this claim. Similarly, Figure 1E presents the conclusion that "These changes in nuclear import and incorporation result in a less dramatic loss of the free nuclear H3.3 pool than previously seen for H3," despite the fact that H3 data are not included in this figure. The conclusions drawn from these data need to be supported with appropriate statistical comparisons and more precise descriptions of what is being measured.

For Figure 1B (now 2B) we do not feel that it is appropriate to directly compare the intensities of the H3-Dendra2 construct expressed from the pseudo-endogenous locus to the H3.3 and chimeric proteins expressed from the H3.3A locus as they were imaged using different settings and therefore we do not feel that direct statistical tests are appropriate. Rather, we plot the two histones on the same graph normalized to their own NC10 values so that the trend in their decrease over time may be compared. The statistical tests for H3.3 compared to the chimeras which were originally in the supplemental table have been added to Figure 3 (formerly figure 2). It is important to note that in this directly comparable situation the ASVM mutant (whose trends closely mirror H3) is highly statistically distinct from H3.3.

We have added a note to the legend of the new Figure 1 explaining this which reads:

"Note that statistical comparisons between the two Dendra2 constructs have not been done as they were expressed from different loci and imaged under different experimental settings."

For Figure 1C-D (now 2C-D) we have removed this claim from the text. We were referring to the plateau in nuclear import for H3 that is less dramatic in H3.3, but this is more carefully discussed in the next paragraph and its addition at that point generated confusion. The text now reads:

Revision Plan

“To further assess how nuclear uptake dynamics changed during these cycles, we tracked total nuclear H3 and H3.3 in each cycle (Figure 2C-D). Since nuclear export is effectively zero, we attribute the increase in total H3.3 over time solely to import and therefore the slope of total H3.3 over time corresponds to the import rate. Though the change in initial import rates between NC10 and NC13 are similar between the two histones (Figure S1F), we observed a notable difference in their behavior in NC13. H3 nuclear accumulation plateaus ~5 minutes into NC13, whereas H3.3 nuclear accumulation merely slows (Figure 2C-D).” lines 109-116

For Figure 1E (now 2E), to address the difference between H3 and H3.3 free pools we have added the previously published values to the text and changed the phrasing from “less dramatic” to “less complete”. The sentence now reads:

“These changes in nuclear import and incorporation result in a less complete loss of the free nuclear H3.3 pool (~70% free in NC11 to ~30% in NC13) than previously seen for H3 (~55% free in NC11 to ~20% in NC13)” lines 116-119

The disappearance of H3.3 from mitotic chromosomes in Figure 2E is also not explained. If this phenomenon is functionally relevant, the authors should provide a mechanistic interpretation, or at the very least, discuss potential explanations in the text. In Figures 2F-H, the reasoning behind comparing the nuclear intensity of H3.3 to H3 in Hira mutants is unclear. To properly assess the role of HIRA in H3.3 chromatin accumulation, a more appropriate comparison would be between wild-type H3.3 and H3.3 levels in Hira knockdown embryos.

As explained in the text and depicted in Figure 3D (formerly 2D), the HIRA^{ssm} mutant is a point mutation that prevents observable H3.3 chromatin incorporation, but not nuclear import. This is what is depicted in Figure 3E (formerly 2E). The loss of H3.3 from mitotic chromatin is due to the inability to incorporate H3.3 into chromatin as expected for a HIRA mutant. We have edited the figure 3 legend to make this more clear. It now reads:

“Hira^{ssm} mutation nearly abolishes the observable H3.3 on mitotic chromatin (E).”

In Figure 3F (formerly 2F-H) we ask what happens to H3 chromatin incorporation when there is almost no incorporation of H3.3 due to the HIRA mutation. In this mutant there is so little H3.3 incorporation that we cannot quantify H3.3 levels on mitotic chromatin (see the new Figure 1B for the stage where chromatin levels are quantified). This experiment was done to test if H3.3^{ASVM} (expressed at the H3.3A locus) is incorporated into chromatin in embryos lacking the function of H3.3’s canonical chaperone. We have edited the text to make this more clear:

“Since the chromatin incorporation of the H3/H3.3 chimeras appears to depend on their chaperone binding sites, we asked if impairing the canonical H3.3 chaperone, Hira, would affect the incorporation of H3.3^{ASVM} expressed from the H3.3A locus.” lines 158-160

Revision Plan

Finally, the section discussing "H3.3 incorporation depends on cell cycle state, but not cell cycle duration" is unclear. The term "cell cycle state" is vague and should be explicitly defined. Does this refer to a specific phase of the cell cycle, changes in chromatin accessibility, or another regulatory mechanism?

The term cell cycle state is deliberately vague. We know that Chk1 regulates many aspects of cell cycle progression and cannot determine from our data which aspect(s) of cell cycle regulation by Chk1 are important for H3.3 incorporation. Our data indicate that it is not simply interphase duration as we originally hypothesized. We have expanded our discussion section to underscore some aspects of Chk1 regulation that we speculate may be responsible for the change in H3.3 behavior.

"Chk1 mutants decrease H3.3 incorporation even before the cell cycle is significantly slowed. Cell cycle slowing has been previously reported to regulate the incorporation of other histone variants in Drosophila¹⁵. However, our results indicate that cell cycle state and not duration per se, regulates H3.3 incorporation. In most cell types, the primary role of Chk1 is to stall the cell cycle to protect chromatin in response to DNA damage. Therefore, Chk1 activity directly or indirectly affects the chromatin state in a variety of ways. We speculate that Chk1's role in regulating origin firing may be particularly important in this context^{73,74}. Late replicating regions and heterochromatin first emerge during ZGA, and Chk1 mutants proceed into mitosis before the chromatin is fully replicated^{22,23,25,71}. Since H3.3 is often associated with heterochromatin, the decreased H3.3 incorporation in Chk1 mutants may be an indirect result of increased origin firing and decreased heterochromatin formation^{73,74}." lines 287-298

Reviewer #3 (Evidence, reproducibility and clarity (Required)):

Summary:

Based on previous findings of the changing ratios of histone H3 to its variant H3.3, the authors test how H3.3 incorporation into chromatin is regulated for ZGA. They demonstrate here that H3 nuclear availability drops and replacement by H3.3 relies on chaperone binding, though not on its typical chaperone Hira. Furthermore, they show that nuclear-cytoplasmic (N/C) ratios can influence this histone exchange likely by influencing cell cycle state.

We thank the reviewer for their thoughtful comments. We note that our data ARE consistent with H3.3 incorporation depending on Hira through its chaperone binding site.

3. In the discussion section, can the authors speculate on how they think H3.3 ASVM is getting incorporated if not through Hira. Are there other known H3 variant chaperones, or can the core histone chaperone substitute?

Revision Plan

We have expanded our discussion to include the the following:

“In the case of the chimeric histone proteins the incorporation behavior was dependent on the chaperone binding site. For example, H3.3^{ASVM} import and incorporation was similar to H3 in control embryos and H3.3^{ASVM} was still incorporated in Hira^{ssm} mutants. This is consistent with the chaperone binding site determining the chromatin incorporation pathway and suggests that H3.3^{ASVM} likely interacts with H3 chaperones such as Caf1.” lines 280-285

Minor comments:

While the paper is well written, I found the figures very confusing and difficult to interpret. Comments here are meant to make it easier to interpret.

1. Fig 1 and most of the paper would benefit from a schematic of early embryo transitions labelled with time and stages of cell cycle to make interpreting data easier

This is an excellent suggestion! We have added a new figure (Figure 1) to explain both the biological system and the way that we measured many properties in this paper.

2. Fig 1- same green color is used for nuclear cycle 12 and for H3.3 making it confusing when reading graphs. Please check other figures where there is a similar use of color for two different things

We have changed the colors so that they are more distinct.

4. Line 130-133: can they also comment on the difference between SVM and ASVM. It seems like SVM might be even worse than ASVM (Fig 2C). Is this related to chk1 phosphorylation?

We think that this is a property of the mixed chimeras since S31A is also imported less efficiently than H3.3 (though we cannot be sure without further experiments). We have added this explanation to the text:

“We speculate that chimeric histone proteins (H3.3^{S31A} and H3.3^{SVM}) are not as efficiently handled by the chaperone machinery as species that are normally found in the organism including H3.3^{ASVM} which is protein-identical to H3.” lines 150-152

5. Fig 2F-G: It is very difficult to compare between histones when they are on different graphs, please consider putting H3, H3.3 and H3.3ASVM in a hirassm background on the same graph.

We have done this in the new Figure 3F.

6. Fig 3- move G to become A and then have A and B.

Revision Plan

We have restructured this figure to include the nuclear density map of control in response to a comment from Reviewer 1. Although not exactly what the reviewer has envisioned, we hope that this adds clarity to the figure.

7. The initial slope graphs in 4D, E, H and I are not easy to understand and would benefit from an explanation in the legend.

We have edited the legend of Figure 5D (formerly 4D) and S1F which now read:

“Initial slopes of nuclear import curves (change in total nuclear intensity over time for the first 5 timepoints) ...”

In addition we have updated the methods to include:

“Import rates were calculated by using a linear regression for the total nuclear intensity over time for the first 5 timepoints in the nuclear import curves.” lines 471-473, methods

4. Description of analyses that authors prefer not to carry out

Please include a point-by-point response explaining why some of the requested data or additional analyses might not be necessary or cannot be provided within the scope of a revision. This can be due to time or resource limitations or in case of disagreement about the necessity of such additional data given the scope of the study. Please leave empty if not applicable.

Reviewer #1

OPTIONAL experimental suggestion: The experiments in Figure 4 and 5 are clever. One would expect that H3 levels might exhaust faster in embryos lacking all H3.2 histone genes (Gunesdogan, 2010, PMID: 20814422), allowing a comparison testing the H3 availability > H3.3 incorporation portion of the hypothesis without manipulating the N/C ratio. This might also result in a more consistent system than slbp RNAi (below).

We thank the reviewer for the experimental suggestion. We also considered this experimental manipulation to decrease RD histone H3.2. We chose not to do this experiment because in the Gunesdogan paper they show that the zygotic HisC nulls have normal development until after NC14 (unlike the maternal SLBP-RNAi that we used) suggesting that maternal H3.2 supplies do not become limiting until after the stages under consideration in our paper. Maternal HisC-nulls are, of course, impossible to generate since histones are essential.

Figures:

Revision Plan

While I appreciate the statistical summaries in tables, it is still helpful to display standard significance on the figures themselves.

We have added statistical comparisons in Figure 3 (formerly Figure 2). We do not feel that it is appropriate to directly compare the intensities of the H3-Dendra2 construct expressed from the pseudo-endogenous locus to the H3.3 and chimeric proteins expressed from the H3.3A locus as they were imaged using different settings. Although we plot H3 on the same graph as the other proteins to allow for ease of comparison of their trends over time it is not appropriate to directly compare their normalized intensities which including statistical tests would encourage. We have added a note to the legend of Figure 1 explaining this which reads:

“Note that statistical comparisons between the two Dendra2 constructs have not been done as they were expressed from different loci and imaged under different experimental settings.”

Fig 1:

B: Statistics required - caption suggests statistics are in Table S2, but why not put on graph? *Please see the explanation above for why we do not feel that it is appropriate to perform this comparison.*

C/D: Would be helpful if authors could plot H3/H3.3 on same graph because what we really need to compare is NC13 between H3/H3.3 (and statistics between these curves) *Please see the explanation above for why we do not feel that it is appropriate to perform this comparison. These curves can be directly compared within a construct and we can evaluate their trends over time, but the normalized values should not be directly compared in the way that would be encouraged by plotting the data as suggested.*

Fig 4:

B/C/F/G: The authors use a point size scale to represent the number of nuclei, but the graphs are so overlaid that it is not particularly useful. Is there a better way to display this dimension? *We chose to represent the data in this way so that the visual impact of each line is representative of the amount of data (number of nuclei in each bin) that underlies it. This helps to prevent sparsely populated outlier bins at the edges of the distribution from dominating the interpretation of the data. If the reviewer has a suggestion for a better way to visualize this information we would welcome their suggestion, but we cannot think of a better way at this time.*

Reviewer #2

Moreover, given that H3.3 is associated with actively transcribed genes, an RNA-seq analysis of chimeric embryos should be included to assess transcriptional changes linked to H3.3 incorporation.

Revision Plan

This is an excellent suggestion and will definitely be a future project for the lab. However, to do this experiment correctly we will need to generate untagged chimeric lines that will (hopefully) allow for the full replacement of H3.3 with the chimeric histones instead of a single copy among 4. This is beyond the scope of this paper.

Figure 2 presents additional concerns regarding data interpretation. The comparisons between H3.3 and H3.3S31A to H3 and H3.3SVM/ASVM lack statistical analysis, making it difficult to determine the significance of the observed differences.

As discussed above, it is not appropriate to directly compare H3 to H3.3 and the chimeras at the H3.3A locus since they are expressed from different promoters and imaged with different settings. The ANOVA comparisons between all of the constructs in the H3.3A locus can be found in the supplemental table. We have also added the statistical significance between each chimera and H3.3 within a cell cycle to the figure. Including the full set of comparisons for all genotypes and timepoints makes the figure nearly impossible to interpret, but they remain available in the supplemental table.

A broader concern is that the authors only test HIRA as a histone chaperone but do not consider alternative chaperones that could influence H3.3 deposition. Since multiple chaperone systems regulate histone incorporation, it would strengthen the conclusions if additional chaperones were tested.

Since HIRA^{ssm} reduced H3.3-Dendra2 incorporation to nearly undetectable levels (Figure 3E) we believe that it is the primary H3.3 incorporation pathway during this period of development. Therefore, we believe that removing HIRA function is a sufficient test of the dependence of H3.3^{ASVM} on the major H3.3 chaperone at this time. Although it would be interesting to fully map how all H3 and H3.3 chimera constructs respond to all histone chaperone pathways, we believe that this is beyond the scope of this manuscript.

Reviewer #3

3. Fig 1C,D might benefit more from being split up into 3 graphs by cell cycle with H3 and H3.3 plotted on the same graphs rather than the way it is now

We do not feel that it is appropriate to directly compare the intensities of the H3-Dendra2 construct expressed from the pseudo-endogenous locus to the H3.3 and chimeric proteins expressed from the H3.3A locus as they were imaged using different settings. These curves can be directly compared within a construct and we can evaluate their trends over time, but the normalized values should not be directly compared in the way that would be encouraged by plotting the data as suggested.

Revision Plan

FULL REVIEWER COMMENTS IN ORDER;

We have duplicated the reviewer comments and our responses in the order they originally appeared so that the reviewers can more easily find their own comments and our responses in their original context. All of the below comments are also included in one of the three response categories above.

Reviewer #1

Evidence, reproducibility and clarity

Summary:

Bhatt et al. seek to define factors that influence H3.3 incorporation in the embryo. They test various hypotheses, pinpointing the nuclear/cytoplasmic ratio and Chk1, which affects cell cycle state, as influencers. The authors use a variety of clever *Drosophila* genetic manipulations in this comprehensive study. The data are presented well and conclusions reasonably drawn and not overblown. I have only minor comments to improve readability and clarity. I suggest two OPTIONAL experiments below.

We thank the reviewer for their positive and helpful comments.

Major comments:

We found this manuscript well written and experimentally thorough, and the data are meticulously presented. We have one modification that we feel is essential to reader understanding and one experimental concern:

The authors provide the photobleaching details in the methodology, but given how integral this measurement is to the conclusions of the paper, we feel that this should be addressed in clear prose in the body of the text. The authors explain briefly how nuclear export is assayed, but not import (line 99). Would help tremendously to clarify the methods here. This is especially important as import is again measured in Fig 4. This should also be clarified (also in the main body and not solely in the methods).

We have added the following sentences to the main body of the text to clarify how photobleaching and import were assayed.

“We note that these differences are not due to photobleaching as our measurements on imaged and unimaged embryos indicate that photobleaching is negligible under our experimental conditions (see methods, Figure S1G-H)” lines 98-101

and

“Since nuclear export is effectively zero, we attribute the increase in total H3.3 over time solely to import and therefore the slope of total H3.3 over time corresponds to the import rate.” lines 111-113

Revision Plan

In addition we have clarified how import was calculated to figure legends in Figure 5D (formerly 4D) and S1F which now read:

“Initial slopes of nuclear import curves (change in total nuclear intensity over time for the first 5 timepoints) ...”

We also added the following explanation of how nuclear import rates were calculated to the methods section:

“Import rates were calculated by using a linear regression for the total nuclear intensity over time for the first 5 timepoints in the nuclear import curves.” lines 471-473, methods

If the embryos appeared "reasonably healthy" (line 113) after slbp RNAi, how do the authors know that the RNAi was effective, especially in THESE embryos, given siblings had clear and drastic phenotype? This is especially critical given that the authors find no effect on H3.3 incorporation after slbp RNAi (and presumably H3 reduction), but this result would also be observed if the slbp RNAi was just not effective in these embryos.

We apologize for the confusion caused by our word choice. The “healthy” slbp-RNAi embryos had measurable phenotypes consistent with histone depletion that we have reported previously (Chari et al, 2019) including cell cycle elongation and early cell cycle arrest (Figure S4D). However, they did not have the catastrophic mitosis observed in more severely affected embryos. We agree with the reviewer that a concern of this experiment is that the less severely affected embryos likely have more remaining RD histones including H3. To address this we also tested H3.3 incorporation in the embryos that fail to progress to later cell cycles in the cycles that we could measure. Even in these more severely affected embryos we were not able to detect a change in H3.3 incorporation relative to controls (lines 240-243 and Fig S4B). Unfortunately, it is impossible to conduct the ideal experiment, which would be a complete removal of H3 since this is incompatible with oogenesis and embryo survival.

To address this confusion we have added supplemental videos of control, moderately affected and severely affected SLBP-RNAi embryos as movies 3-5 and modified the text to read:

“All embryos that survive through at least NC12, had elongated cell cycles in NC12 and 60% arrested in NC13 as reported previously indicating the effectiveness of the knockdown (Figure S4C, Movie 3-5)³⁹. In these embryos, H3.3 incorporation is largely unaffected by the reduction in RD H3 (Figure 6B).” lines 236-240

Finally, to characterize the range of SLBP knockdown in the RNAi embryos we propose to do single embryo RT-qPCRs for SLBP mRNA for multiple individual embryos. This will provide a measure of the range knockdown that we observed in our H3.3 movies.

Minor comments:

Introduction:

Revision Plan

Consider using "replication dependent" (RD) rather than "replication coupled." Both are used in the field, but RD parallels RI ("replication independent").

We thank the reviewer for this suggestion. We have made the text edits to change "replication coupled" (RC) to "replication dependent" (RD) throughout the manuscript.

Would help for clarity if the authors noted that H3 is equivalent to H3.2 in Drosophila. Also it is relevant that there are two H3.3 loci as the authors knock mutations into the H3.3A locus, but leave the H3.3B locus intact. The authors should clarify that there are two H3.3 genes in the Drosophila genome.

We have changed the text as follows to increase clarity as suggested:

"Similarly, we have previously shown that RD H3.2 (hereafter referred to as H3) is replaced by RI H3.3 during these same cycles, though the cause remains unclear²⁹" lines 52-54

"There are ~100 copies of H3 in the Drosophila genome, but only 2 of H3.3 (H3.3A and H3.3B)²⁶. To determine which factor controls nuclear availability and chromatin incorporation, we genetically engineered flies to express Dendra2-tagged H3/H3.3 chimeras at the endogenous H3.3A locus, keeping the H3.3B locus intact." lines 127-131

Please add information and citation (line 58): H3.3 is required to complete development when H3.2 copy number is reduced (PMID: 37279945, McPherson et al. 2023)

We have added the suggested information. The text now reads

"Nonetheless, H3.3 is required to complete development when H3.2 copy number is reduced⁵⁴." lines 61-62

Results:

Embryo genotype is unclear (line 147): Hira^[ssm] haploid embryos inherit the Hira mutation maternally? Are Hira homozygous mothers crossed to homozygous fathers to generate these embryos, or are mothers heterozygous? This detail should be in the main text for clarity.

The Hira mutants are maternal effect. We crossed homozygous Hira^{ssm} females to their hemizygous Hira^{ssm} or FM7C brothers. However, the genotype of the male is irrelevant since the Hira phenotype prevents sperm pronuclear fusion and therefore there is no paternal contribution to the embryonic genotype. We have clarified this point in the text:

"We generated embryos lacking functional maternal Hira using Hira^{ssm-185b} (hereafter Hira^{ssm}) homozygous mothers which have a point mutation in the Hira locus⁵⁷." lines 160-162

Revision Plan

Line 161: Shkl affects nuclear density, but it also appears from Fig 3 to affect nuclear size? The authors do not address this, but it should at least be mentioned.

We thank the reviewer for the astute observation. More dense regions of the Shkl embryos do in fact have smaller nuclei. We believe that this is a direct result of the increased N/C ratio since nuclear size also falls during normal development as the N/C ratio increases. We have added a new figure 1 in which we more carefully describe the events of early embryogenesis in flies including a quantification of nuclear size and number in the pre-ZGA cell cycles (Figure 1C). We also note the correlation of nuclear size with nuclear density in the text:

“During the pre-ZGA cycles (NC10-13), the maximum volume that each nucleus attains decreases in response to the doubling number of nuclei with each division (Figure 1C).” lines 86-87

“To test this, we employed mutants in the gene Shackleton (shkl) whose embryos have non-uniform nuclear densities and therefore a gradient of nuclear sizes across the anterior/posterior axis (Figure 3A-B, Movie 1-2)⁵⁸.” lines 180-183

The authors often describe nuclear H3/H3.3 as chromatin incorporated, but these image-based methods do not distinguish between chromatin-incorporated and nuclear protein.

To distinguish between chromatin incorporated and nuclear free histone we have exploited the fact that histones that are not incorporated into DNA freely diffuse away from the chromatin mass during mitosis while those that are bound into nucleosomes remain on chromatin during this time. In our previous study we showed that H3-Dendra2 that is photoconverted during mitosis remains stably associated with the mitotic chromatin through multiple cell cycles (Shindo and Amodeo, 2019) strengthening our use of this metric. To help clarify this point as well as other methodological details we have added a new Figure 1B which documents the time points at which we make various measurements within the lifecycle of the nucleus. We also edited the text to read:

“We have previously shown that with each NC, the pool of free H3 in the nucleus is depleted and its levels on chromatin during mitosis decrease (Figure 1D, S1C-D)²⁹. In contrast, H3.3 mitotic chromatin levels increase during the same cycles (Figure 1D, S1C-D)²⁹.” lines 89-92

I very much appreciate how the authors laid out their model in Fig 3 and then used the same figure to explain which part of the model they are testing in Figs 4 and 5. This is not a critique- we can complement too!

Thank you!

Revision Plan

OPTIONAL experimental suggestion: The experiments in Figure 4 and 5 are clever. One would expect that H3 levels might exhaust faster in embryos lacking all H3.2 histone genes (Gunesdogan, 2010, PMID: 20814422), allowing a comparison testing the H3 availability > H3.3 incorporation portion of the hypothesis without manipulating the N/C ratio. This might also result in a more consistent system than slbp RNAi (below).

We thank the reviewer for the experimental suggestion. We also considered this experimental manipulation to decrease RD histone H3.2. We chose not to do this experiment because in the Gunesdogan paper they show that the zygotic HisC nulls have normal development until after NC14 (unlike the maternal SLBP-RNAi that we used) suggesting that maternal H3.2 supplies do not become limiting until after the stages under consideration in our paper. Maternal HisC-nulls are, of course, impossible to generate since histones are essential.

O'Haren 2024 (PMID: 39661467) did not find increased Pol II at the HLB after zelda RNAi (line 227). Might also want to mention here that zelda RNAi does not result in changes to H3 at the mRNA level (O'Haren 2024), as that would confound the model.

We thank the reviewer for the suggestion. We have removed the discussion of Pol II localization and replaced it with the information about histone mRNA :

“zelda controls the transcription of the majority of Pol II genes during ZGA but disruption of zelda does not change RD histone mRNA levels⁶⁷⁻⁷⁰.” lines 249-251

Discussion:

Should discuss results in context of McPherson et al. 2023 (PMID: 37279945), who showed that decreasing H3.2 gene numbers does not increase H3.3 production at the mRNA or protein levels.

We expanded our discussion to include the following:

“Given the fact that H3.3 pool size does not respond to H3 copy number in other Drosophila tissues,⁵⁴ our results suggest that H3.3 incorporation dynamics are likely independent of H3 availability.” lines 278-280

The Shackleton mutation is a clever way to alter N/C ratio, but the authors should point out that it is difficult (impossible?) to directly and cleanly manipulate the N/C ratio. For example, Shkl mutants seem to also have various nuclear sizes.

As discussed above, we think that nuclear size is a direct response to the N/C ratio. We have added the following sentence to the discussion as well as a citation to a paper which discusses how the N/C ratio might contribute to nuclear import in early embryos to the discussion:

“This may be due to N/C ratio-dependent changes in nuclear import dynamics which may also contribute to the observed changes in nuclear size across the shkl embryo⁷⁵.” lines 307-309

Revision Plan

How is H3.3 expression controlled? Is it possible that H3.3 biosynthesis is affected in Chk1 mutants?

To address this question we propose to perform RT-qPCR for H3.3A and H3.3B as well as Hira in the Chk1 mutant. Unfortunately, we do not have antibodies that reliably distinguish between H3 and H3.3 in our hands (despite literature reports), but we will also perform a pan-H3 immunostaining in the Chk1 embryos to measure how the total H3-type histone pool changes as a result of the loss of Chk1.

Figures:

While I appreciate the statistical summaries in tables, it is still helpful to display standard significance on the figures themselves.

We have added statistical comparisons in Figure 3 (formerly Figure 2). We do not feel that it is appropriate to directly compare the intensities of the H3-Dendra2 construct expressed from the pseudo-endogenous locus to the H3.3 and chimeric proteins expressed from the H3.3A locus as they were imaged using different settings. Although we plot H3 on the same graph as the other proteins to allow for ease of comparison of their trends over time it is not appropriate to directly compare their normalized intensities which including statistical tests would encourage. We have added a note to the legend of Figure 1 explaining this which reads:

“Note that statistical comparisons between the two Dendra2 constructs have not been done as they were expressed from different loci and imaged under different experimental settings.”

Fig 1:

A: Is it possible to label panels with the nuclear cycle? *We have done this.*

B: Statistics required - caption suggests statistics are in Table S2, but why not put on graph? *Please see the explanation above for why we do not feel that it is appropriate to perform this comparison.*

C/D: Would be helpful if authors could plot H3/H3.3 on same graph because what we really need to compare is NC13 between H3/H3.3 (and statistics between these curves) *Please see the explanation above for why we do not feel that it is appropriate to perform this comparison. These curves can be directly compared within a construct and we can evaluate their trends over time, but the normalized values should not be directly compared in the way that would be encouraged by plotting the data as suggested.*

E: The comparison in the text is between H3.3 and H3, but only H3.3 data is shown. I realize that it is published prior, but the comparison in figure would be helpful.

We have added the previously published values to the text.

Revision Plan

“These changes in nuclear import and incorporation result in a less complete loss of the free nuclear H3.3 pool (~70% free in NC11 to ~30% in NC13) than previously seen for H3 (~55% free in NC11 to ~20% in NC13)” lines 116-119

Fig 2:

A: A very helpful figure. Slightly unclear that the H3 that is not Dendra tagged is at the H3.3 locus. Also unclear that the H3.3A-Dendra2 line exists and used as control, as is not shown in figure.

Should show H3 and H3.3 controls (Figure S2)

We have edited the figure to add Dendra2 to all of the constructs and made clear the location of each construct including adding the landing site for H3-Dendra2. We have also cited Figure S1 in the legend which contains a more detailed diagram of the integration strategy.

F/H- As the comparison is between H3 and ASVM, it would help to combine these data onto the same graph. As the color is currently used unnecessarily to represent nuclear cycle, the authors could use their purple/pink color coding to represent H3/ASVM. *We have combined these data onto a single graph as requested and changed the colors appropriately. We have not added statistical comparisons to this graph as we again believe that they would be inappropriate.*

In the legend of Fig 2 the authors write "in the absence of Hira." Technically, there is only a point mutation in Hira. It is not absent.

Good catch! We have changed this to “in Hira^{ssm} mutants”.

Fig 3:

G: Please show WT for comparison. Can use data in Fig 3A. *We have added the color-coded number of neighbor embryo representations for WT and Shkl embryos underneath the example embryo images in 4A-B (formerly 3A-B,G).*

Model in H is very helpful (complement)! *Thank you.*

Fig 4:

B/C/F/G: The authors use a point size scale to represent the number of nuclei, but the graphs are so overlaid that it is not particularly useful. Is there a better way to display this dimension? *We chose to represent the data in this way so that the visual impact of each line is representative of the amount of data (number of nuclei in each bin) that underlies it. This helps to prevent sparsely populated outlier bins at the edges of the distribution from dominating the interpretation of the data. If the reviewer has a suggestion for a better way to visualize this information we would welcome their suggestion, but we cannot think of a better way at this time.*

D/E/H/I: What does "min volume" mean on the X axis?

Since the uneven N/C ratio in the shkl embryos results in a wavy cell cycle pattern there is no single time point where we can calculate the number of neighbors for the whole embryo (since

Revision Plan

not all nuclei are in the same cell cycle at a given point). Therefore, we had to choose a criterion for when we would calculate the number of neighbors for each nucleus. We chose nuclear size as a proxy for nuclear age since nuclear size increases throughout interphase (see new figure 1B). So, the minimum volume is the newly formed nucleus in a given cell cycle. We also tested other timepoints for the number of neighbors (maximum nuclear volume, just before nuclear envelope breakdown and midway between these two points) and found similar results. We chose to use minimum volume in this paper because this is the time point when the nucleus is growing most quickly and nuclear import is at its highest. We have added the following explanation to the methods:

“For shk1 embryos, as the nuclear cycles are asynchronous, nuclear divisions start at different timepoints within the same cell cycle and the nuclear density changes as the neighboring nuclei divide. Therefore, the total intensity traces were aligned to match their minimum volumes (as shown in Figure 1B) to T0.” lines 485-488, methods

And the following detail to the figure legend:

*“...plotted by the number of nuclear neighbors at their minimum nuclear volume...”
Figure 5 legend*

We also added a depiction of the lifecycle of the nucleus in which we marked the minimum volume as the new Figure 1B.

Fig 5:

F: OPTIONAL Experimental request: Here I would like to see H3 as a control.

This is a very good suggestion, and we are currently imaging H3-Dendra2 in the Chk1 background. However, our preliminary results suggest that there may be some synthetic early lethality between the tagged H3-Dendra2 and Chk1 since these embryos are much less healthy than H3.3-Dendra2 Chk1 embryos or Chk1 with other reporters. In addition, we have observed a much higher level of background fluorescence in this cross than in the H3-Dendra2 control. We are uncertain if we will be able to obtain usable data from this experiment, but will continue to try to find conditions that allow us to analyze this data.

As an orthogonal approach to answer the question, we will perform immunostaining with a pan-H3 antibody in Chk1 mutant embryos to measure total H3 levels under these conditions. Since the majority of H3-type histone is H3.2 and we know how H3.3 changes, this staining will give us insight into the dynamics of H3 in Chk1 mutant embryos.

Significance

General assessment: Many long-standing mysteries surround zygotic genome activation, and here the authors tackle one: what are the signals to remodel the zygotic chromatin around ZGA? This is a tricky question to answer, as basically all manipulations done to the embryo

Revision Plan

have widespread effects on gene expression in general, confounding any conclusions. The authors use clever novel techniques to address the question. Using photoconvertible H3 and H3.3, they can compare the nuclear dynamics of these proteins after embryo manipulation. Their model is thorough and they address most aspects of it. The hurdle this study struggles to overcome is the same that all ZGA studies have, which is that manipulation of the embryo causes cascading disasters (for example, one cannot manipulate the nuclear:cytoplasmic ratio without also altering cell cycle timing), so it's challenging to attribute molecular phenotypes to a single cause. This doesn't diminish the utility of the study.

Advance: The conceptual advance of this study is that it implicates the nuclear:cytoplasmic ratio and Chk1 in H3.3 incorporation. The authors suggest these factors influence cell cycle closing, which then affects H3.3 incorporation, although directly testing the granularity of this model is beyond the scope of the study. The authors also provide technical advancement in their use of measuring histone dynamics and using changes in the dynamics upon treatment as a useful readout. I envision this strategy (and the dendra transgenes) to be broadly useful in the cell cycle and developmental fields.

Audience: The basic research presented in this study will likely attract colleagues from the cell cycle and embryogenesis fields. It has broader implications beyond *Drosophila* and even zygotic genome activation.

This reviewer's expertise: Chromatin, *Drosophila*, Gene Regulation

Reviewer #2 (Evidence, reproducibility and clarity (Required)):

This manuscript investigates the regulation of H3.3 incorporation during zygotic genome activation (ZGA) in *Drosophila*, proposing that the nuclear-to-cytoplasmic (N/C) ratio plays a central role in this process. While the study is conceptually interesting, several concerns arise regarding the lack of proper control experiments and the clarity of the writing. The manuscript is difficult to follow due to vague descriptions, insufficient distinctions between established knowledge and novel findings, and a lack of rigorous statistical analyses. These issues need to be addressed before the study can be considered for publication.

We thank the reviewers for their careful reading of this manuscript. We have sought to clarify the concerns regarding clarity through numerous text edits detailed below. We did include ANOVA analysis for all of the relevant statistical comparisons in the supplemental table. However, to increase clarity we have also added some statistical comparisons in the main figures. We note that we do not feel that it is appropriate to directly compare the intensities of the H3-Dendra2 construct expressed from the pseudo-endogenous locus to the H3.3 and chimeric proteins expressed from the H3.3A locus as they were imaged using different settings. Although we plot H3 on the same graph as the other proteins to allow for ease of comparison of their trends over time it is not appropriate to directly compare their normalized intensities which including statistical tests would encourage. We have added a note to the legend of the new Figure 1

Revision Plan

explaining this which reads:

“Note that statistical comparisons between the two Dendra2 constructs have not been done as they were expressed from different loci and imaged under different experimental settings.”

Major Concerns

The manuscript would benefit from a clearer introduction that explicitly distinguishes between previously known mechanisms of histone regulation during ZGA and the novel contributions of this study. Currently, the introduction lacks sufficient background on early embryonic chromatin regulation, making it difficult for readers unfamiliar with the field to grasp the significance of the findings. The authors should also be more precise when discussing the timing of ZGA. While they state that ZGA occurs after 13 nuclear divisions, it is well established that a minor wave of ZGA begins at nuclear cycle 7-8, whereas the major wave occurs after cycle 13. Clarifying this distinction will improve the manuscript's accessibility to a broader audience.

We have added a new figure 1 to make the timing and nuclear behaviors of the embryo during ZGA in Drosophila more clear. We have also added information about how the chromatin changes during Drosophila ZGA in the following sentence:

“ In Drosophila, these changes include refinement of nucleosomal positioning, partitioning of euchromatin and heterochromatin and formation of topologically associated domains^{20-22,24}.” lines 39-41

We have clarified the major and minor waves of ZGA in the introduction and results by adding the following sentences to the introduction and results respectively:

“In most organisms ZGA happens in multiple waves but the chromatin undergoes extensive remodeling to facilitate bulk transcription during the major wave of ZGA (hereafter referred to as ZGA)^{18-20,22-25}..” lines 36-39

“In Drosophila, ZGA occurs in 2 waves. The minor wave starts as early as the 7th cycle, while major ZGA occurs after 13 rapid syncytial nuclear cycles (NCs) and is accompanied by cell cycle slowing and cellularization (Figure 1A-B).” lines 83-85

We hope that these changes help to reduce confusion and make the paper more accessible. However, we are happy to add additional information if the reviewer can provide specific points which require further attention.

One of the primary weaknesses of this study is the lack of adequate control experiments. In Figure 1, the authors suggest that the levels of H3 and H3.3 are influenced by the N/C ratio, but

Revision Plan

it is unclear whether transcription itself plays a role in these dynamics. To properly test this, RNA-seq or Western blot analyses should be performed at nuclear cycles 10 and 13-14 to compare the levels of newly transcribed H3 or H3.3 against maternally supplied histones. Without such data, the authors cannot rule out transcriptional regulation as a contributing factor.

In the pre-ZGA cell cycles the vast majority of protein including histones is maternally loaded. Gunesdogan et al. (2010) showed that the zygotic RD histone cluster nulls survive past NC14 (well past ZGA) with no discernible defects indicating that maternal RD histone supplies are sufficient for normal development during the cell cycles under consideration. Therefore, new transcription of replication coupled histones is not needed for apparently normal development during this period. Moreover, we have done the western blot analysis using a Pan-H3 antibody as suggested by the reviewer in our previously published paper (Shindo and Amodeo, 2019 supplemental figure S3A-B) and found that total H3-type histone proteins only increase moderately during this period of development, nowhere near the rate of the nuclear doublings. We have added the following sentence to clarify this point.

“These divisions are driven by maternally provided components and the total amount of H3 type histones do not keep up with the pace of new DNA produced²⁹.” lines 88-89

We have also previously done RNA-seq on wild-type embryos (and those with altered maternal histone levels) (Chari et al 2019). In this RNA-seq (like most RNA-seq in flies) we used poly-A selection and therefore cannot detect the RD histone mRNAs (which have a stem-loop instead of a poly-A tail). We have plotted the mRNA concentrations for both H3.3 variants from that dataset below for the reviewers reference (we have not included this in the revised manuscript). The total H3.3 mRNA levels are nearly constant from egg laying (NC0- these are from unfertilized embryos) until after ZGA (NC14). These data combined with the westerns discussed above give us confidence that what we are observing is the partitioning of large pools of maternally provided histones with only a relatively small contribution of new histone synthesis.

Revision Plan

In Figure 2, the manuscript introduces chimeric embryos expressing modified histone variants, but their developmental viability is not addressed. It is essential to determine whether these embryos survive and whether they exhibit any phenotypic consequences such as altered hatching rates, defects in nuclear division, or developmental arrest.

Tagging histones is often deleterious to organismal health. In Drosophila there are two H3.3 loci (H3.3A and H3.3B). In all of our chimera experiments we have left the H3.3B and one copy of the H3.3A locus unperturbed to provide a supply of untagged H3.3. This allows us to study H3.3 and chimera dynamics without compromising organism health. All of our chimeras are viable and fertile with no obvious morphological defects. We have added the following sentences to the text to clarify this point:

“There are ~100 copies of H3 in the Drosophila genome, but only 2 of H3.3 (H3.3A and H3.3B)²⁶. To determine which factor controls nuclear availability and chromatin incorporation, we genetically engineered flies to express Dendra2-tagged H3/H3.3 chimeras at the endogenous H3.3A locus, keeping the H3.3B locus intact....These chimeras were all viable and fertile.” lines 127-131, 136

In addition we propose performing hatch rate assays for embryos from the chimeric embryos of S31A, SVM and ASVM to assess if there is any decrease in fecundity due to the presence of the chimeras.

Moreover, given that H3.3 is associated with actively transcribed genes, an RNA-seq analysis of chimeric embryos should be included to assess transcriptional changes linked to H3.3 incorporation.

This is an excellent suggestion and will definitely be a future project for the lab. However, to do this experiment correctly we will need to generate untagged chimeric lines that will (hopefully) allow for the full replacement of H3.3 with the chimeric histones instead of a single copy among 4. This is beyond the scope of this paper.

Figures 3 and 4 raise additional concerns about whether histone cluster transcription is altered in shkl mutant embryos. The authors propose that the shkl mutation affects the N/C ratio, yet it remains unclear whether this leads to changes in the transcription of histone clusters. Furthermore, since HIRA is a key chaperone for H3.3, it would be important to assess whether its levels or function are compromised in shkl mutants. To address these gaps, RT-qPCR or RNA-seq should be performed to quantify histone cluster transcription, and Western blot analysis should be used to determine if HIRA protein levels are affected.

The changes in the N/C ratio that are observed in the shkl mutant are within SINGLE embryo (differences in nuclear spacing). In these experiments we are comparing nuclei within a common cytoplasm that have different local nuclear densities (N/C ratios). Therefore, if Shkl

Revision Plan

were somehow affecting the transcription of histones or their chaperones we would expect all of the nuclei within the same mutant embryo to be equally affected since they are genetically identical and share a common cytoplasm. We do not directly compare the behavior of shkl embryos to wildtype except to demonstrate that there is no positional effect on the import of H3 and H3.3 across the length of the embryo in wildtype. To clarify our experimental system for these experiments we have added additional panels to Figure 4A and B that depict the number of neighbors for both control and Shkl embryos.

Nonetheless, to address the reviewer's concern that shkl may change the amount of H3 present in the embryo, we propose to conduct a western blot comparison of wildtype and shkl embryos using a pan-H3 antibody. There are no tools (antibodies or fluorescently tagged proteins) to assess HIRA protein levels in Drosophila. We therefore propose to perform RT-qPCR for HIRA in wildtype and shkl embryos.

A similar issue arises in Figure 5, where the authors claim that H3.3 incorporation is dependent on cell cycle state but do not sufficiently test whether this is linked to changes in HIRA levels. Given the importance of HIRA in H3.3 deposition, its levels should be examined in Slbp, Zelda, and Chk1 RNAi embryos to verify whether changes in H3.3 incorporation correlate with HIRA function. Without this, it is difficult to conclude that the observed effects are strictly due to cell cycle regulation rather than histone chaperone dynamics.

Since H3.3 incorporation is unaffected in the Slbp and Zelda-RNAi lines there is no reason to suspect a change in HIRA function. There are no available tools (antibodies or fluorescently tagged proteins) to directly measure HIRA protein in Drosophila. To test if changes in HIRA loading might contribute to the decreased H3.3 incorporation in the Chk1 mutant we propose to perform RT-qPCR for HIRA in wildtype and Chk1 embryos.

Several figures require additional statistical analyses to support the claims made. In Figure 1B, statistical testing should be included to validate the reported differences. Figure 1C-D states that "H3.3 accumulation reduces more slowly than H3," yet there is no quantitative comparison to substantiate this claim. Similarly, Figure 1E presents the conclusion that "These changes in nuclear import and incorporation result in a less dramatic loss of the free nuclear H3.3 pool than previously seen for H3," despite the fact that H3 data are not included in this figure. The conclusions drawn from these data need to be supported with appropriate statistical comparisons and more precise descriptions of what is being measured.

For Figure 1B (now 2B) we do not feel that it is appropriate to directly compare the intensities of the H3-Dendra2 construct expressed from the pseudo-endogenous locus to the H3.3 and chimeric proteins expressed from the H3.3A locus as they were imaged using different settings and therefore we do not feel that direct statistical tests are appropriate. Rather, we plot the two histones on the same graph normalized to their own NC10 values so that the trend in their decrease over time may be compared. The statistical tests for H3.3 compared to the chimeras which were originally in the supplemental table have been added to Figure 3 (formerly figure 2).

Revision Plan

It is important to note that in this directly comparable situation the ASVM mutant (whose trends closely mirror H3) is highly statistically distinct from H3.3.

We have added a note to the legend of the new Figure 1 explaining this which reads:

“Note that statistical comparisons between the two Dendra2 constructs have not been done as they were expressed from different loci and imaged under different experimental settings.”

For Figure 1C-D (now 2C-D) we have removed this claim from the text. We were referring to the plateau in nuclear import for H3 that is less dramatic in H3.3, but this is more carefully discussed in the next paragraph and its addition at that point generated confusion. The text now reads:

“To further assess how nuclear uptake dynamics changed during these cycles, we tracked total nuclear H3 and H3.3 in each cycle (Figure 2C-D). Since nuclear export is effectively zero, we attribute the increase in total H3.3 over time solely to import and therefore the slope of total H3.3 over time corresponds to the import rate. Though the change in initial import rates between NC10 and NC13 are similar between the two histones (Figure S1F), we observed a notable difference in their behavior in NC13. H3 nuclear accumulation plateaus ~5 minutes into NC13, whereas H3.3 nuclear accumulation merely slows (Figure 2C-D).” lines 109-116

For Figure 1E (now 2E), to address the difference between H3 and H3.3 free pools we have added the previously published values to the text and changed the phrasing from “less dramatic” to “less complete”. The sentence now reads:

“These changes in nuclear import and incorporation result in a less complete loss of the free nuclear H3.3 pool (~70% free in NC11 to ~30% in NC13) than previously seen for H3 (~55% free in NC11 to ~20% in NC13)” lines 116-119

Figure 2 presents additional concerns regarding data interpretation. The comparisons between H3.3 and H3.3S31A to H3 and H3.3SVM/ASVM lack statistical analysis, making it difficult to determine the significance of the observed differences.

As discussed above, it is not appropriate to directly compare H3 to H3.3 and the chimeras at the H3.3A locus since they are expressed from different promoters and imaged with different settings. The ANOVA comparisons between all of the constructs in the H3.3A locus can be found in the supplemental table. We have also added the statistical significance between each chimera and H3.3 within a cell cycle to the figure. Including the full set of comparisons for all genotypes and timepoints makes the figure nearly impossible to interpret, but they remain available in the supplemental table.

Revision Plan

The disappearance of H3.3 from mitotic chromosomes in Figure 2E is also not explained. If this phenomenon is functionally relevant, the authors should provide a mechanistic interpretation, or at the very least, discuss potential explanations in the text. In Figures 2F-H, the reasoning behind comparing the nuclear intensity of H3.3 to H3 in Hira mutants is unclear. To properly assess the role of HIRA in H3.3 chromatin accumulation, a more appropriate comparison would be between wild-type H3.3 and H3.3 levels in Hira knockdown embryos.

As explained in the text and depicted in Figure 3D (formerly 2D), the HIRA^{ssm} mutant is a point mutation that prevents observable H3.3 chromatin incorporation, but not nuclear import. This is what is depicted in Figure 3E (formerly 2E). The loss of H3.3 from mitotic chromatin is due to the inability to incorporate H3.3 into chromatin as expected for a HIRA mutant. We have edited the figure 3 legend to make this more clear. It now reads:

“Hira^{ssm} mutation nearly abolishes the observable H3.3 on mitotic chromatin (E).”

In Figure 3F (formerly 2F-H) we ask what happens to H3 chromatin incorporation when there is almost no incorporation of H3.3 due to the HIRA mutation. In this mutant there is so little H3.3 incorporation that we cannot quantify H3.3 levels on mitotic chromatin (see the new Figure 1B for the stage where chromatin levels are quantified). This experiment was done to test if H3.3^{ASVM} (expressed at the H3.3A locus) is incorporated into chromatin in embryos lacking the function of H3.3’s canonical chaperone. We have edited the text to make this more clear:

“Since the chromatin incorporation of the H3/H3.3 chimeras appears to depend on their chaperone binding sites, we asked if impairing the canonical H3.3 chaperone, Hira, would affect the incorporation of H3.3^{ASVM} expressed from the H3.3A locus.”lines 158-160

A broader concern is that the authors only test HIRA as a histone chaperone but do not consider alternative chaperones that could influence H3.3 deposition. Since multiple chaperone systems regulate histone incorporation, it would strengthen the conclusions if additional chaperones were tested.

Since HIRA^{ssm} reduced H3.3-Dendra2 incorporation to nearly undetectable levels (Figure 3E) we believe that it is the primary H3.3 incorporation pathway during this period of development. Therefore, we believe that removing HIRA function is a sufficient test of the dependence of H3.3^{ASVM} on the major H3.3 chaperone at this time. Although it would be interesting to fully map how all H3 and H3.3 chimera constructs respond to all histone chaperone pathways, we believe that this is beyond the scope of this manuscript.

Additionally, the manuscript does not include any validation of the RNAi knockdown efficiencies used throughout the study. This raises concerns about whether the observed phenotypes are truly due to target gene depletion or off-target effects. RT-qPCR or Western blot analyses should be performed to confirm knockdown efficiency.

Revision Plan

Both the Zelda and slbp-RNAi lines used for knockdowns have been used and validated in the early fly embryo in previously published works ((Yamada et al., 2019), (Duan et al., 2021), (O'Haren et al., 2025), (Chari et al, 2019)) and the phenotypes that we observe in our embryos are consistent with the published data including altered cell cycle durations (Figure S4C) and lack of cellularization/gastrulation. We note that the zelda RNAi phenotypes are also highly consistent with the effects of Zelda germline clones. To validate that slbp-RNAi knocks down histones we included a western blot for Pan-H3 in slbp-RNAi embryos that demonstrates a large effect on total H3 levels (Figure S4A). To further demonstrate the phenotypic effects of the slbp-RNAi we have added supplemental movies (Videos 4 and 5).

To fully characterize the RNAi efficiency under our conditions we propose to perform RT-qPCR for slbp in slbp-RNAi and Zelda in Zelda-RNAi compared to control (w) RNAi embryos.

Finally, the section discussing "H3.3 incorporation depends on cell cycle state, but not cell cycle duration" is unclear. The term "cell cycle state" is vague and should be explicitly defined. Does this refer to a specific phase of the cell cycle, changes in chromatin accessibility, or another regulatory mechanism?

The term cell cycle state is deliberately vague. We know that Chk1 regulates many aspects of cell cycle progression and cannot determine from our data which aspect(s) of cell cycle regulation by Chk1 are important for H3.3 incorporation. Our data indicate that it is not simply interphase duration as we originally hypothesized. We have expanded our discussion section to underscore some aspects of Chk1 regulation that we speculate may be responsible for the change in H3.3 behavior.

“Chk1 mutants decrease H3.3 incorporation even before the cell cycle is significantly slowed. Cell cycle slowing has been previously reported to regulate the incorporation of other histone variants in Drosophila¹⁵. However, our results indicate that cell cycle state and not duration per se, regulates H3.3 incorporation. In most cell types, the primary role of Chk1 is to stall the cell cycle to protect chromatin in response to DNA damage. Therefore, Chk1 activity directly or indirectly affects the chromatin state in a variety of ways. We speculate that Chk1’s role in regulating origin firing may be particularly important in this context^{73,74}. Late replicating regions and heterochromatin first emerge during ZGA, and Chk1 mutants proceed into mitosis before the chromatin is fully replicated^{22,23,25,71}. Since H3.3 is often associated with heterochromatin, the decreased H3.3 incorporation in Chk1 mutants may be an indirect result of increased origin firing and decreased heterochromatin formation^{73,74}.” lines 287-298

Reviewer #2 (Significance (Required)):

This manuscript investigates the regulation of H3.3 incorporation during zygotic genome

Revision Plan

activation (ZGA) in *Drosophila*, proposing that the nuclear-to-cytoplasmic (N/C) ratio plays a central role in this process. While the study is conceptually interesting, several concerns arise regarding the lack of proper control experiments and the clarity of the writing. The manuscript is difficult to follow due to vague descriptions, insufficient distinctions between established knowledge and novel findings, and a lack of rigorous statistical analyses. These issues need to be addressed before the study can be considered for publication.

Reviewer #3 (Evidence, reproducibility and clarity (Required)):

Summary:

Based on previous findings of the changing ratios of histone H3 to its variant H3.3, the authors test how H3.3 incorporation into chromatin is regulated for ZGA. They demonstrate here that H3 nuclear availability drops and replacement by H3.3 relies on chaperone binding, though not on its typical chaperone Hira. Furthermore, they show that nuclear-cytoplasmic (N/C) ratios can influence this histone exchange likely by influencing cell cycle state.

We thank the reviewer for their thoughtful comments. We note that our data ARE consistent with H3.3 incorporation depending on Hira through its chaperone binding site.

Major comments:

1. The claims are largely supported by the data but I think a couple more experiments could help bolster the claims about cell cycle and chk1 regulation.

a. Creating a phosphomimetic of the chk1 phosphorylation site on H3.3 to see if it can overcome the defects seen in chk1 mutants

b. Assessing heterochromatin of embryos without chk1 (or ASVM mutants) for example, by looking at H3K9me3 levels

The first experiments could take several months if the flies haven't already been generated by the authors but the second should be quicker.

a. This is an excellent experimental suggestion which is bolstered by the fact that in frogs H3.3 S31A cannot rescue H3.3 morpholino during gastrulation, but H3.3S31D can (Sitbon et al, 2020). However, to correctly conduct this experiment would require generating and validating multiple additional endogenous H3.3 replacement lines, likely without a fluorescent tag as they can interfere with histone rescue constructs in most species. As the reviewer notes, this would take several months of work (we have not generated the critical flies yet) and may not yield a satisfying answer since there are reports that H3.3 may be dispensable in flies aside from as a source of H3-type histone outside of S-phase (Hödl and Bassler, 2012). While we hope to continue experiments along these lines in the future we feel that this is beyond the scope of the current manuscript.

Revision Plan

b. To address this we propose to stain for H3K9me3 in wildtype and Chk1^{-/-} embryos. Since the ASVM line is not a full replacement of all H3.3 we think that staining for H3K9me3 in this line is unlikely to yield a detectable difference.

2. It would also be interesting to see what the health of the flies with some mutations in this paper are beyond the embryo stage if they are viable (e.g., development to adulthood, fertility etc.)

a. the SVM, ASVM mutations

b. the hira + ASVM mutations

The authors might already have this data but if not they have the flies and it shouldn't take long to get these data.

a. To address this concern we propose to conduct hatch rate assays for embryos from the Dendra tagged H3.3, S31A, SVM, ASVM flies. However, we do note that in our experiments only one copy of the H3.3A locus was mutated and tagged with Dendra2 leaving one copy of H3.3A and both copies of H3.3B untouched to ensure normal development as tagging all copies of histone genes can lead to lethality.

b. All Hira mutants develop as haploids due to the inability to decondense the sperm chromatin (which is dependent on Hira). This leads to one extra division to restore the N/C ratio prior to cell cycle slowing and ZGA. These embryos go on to gastrulate and die late in development after cuticle formation (presumably due to their decreased ploidy) (Loppin et al., 2000). The addition of ASVM into the Hira background does not appear to rescue the ploidy defect as these embryos also undergo the extra division (Figure 3H). We are therefore confident that these embryos will not hatch. We have added the information about the development of Hira mutant to the text as follow:

“These embryos develop as haploids and undergo one additional syncytial division before ZGA (NC14). Hira^{ssm} embryos develop otherwise phenotypically normally through organogenesis and cuticle formation, but die before hatching⁵⁷.” lines 164-167

3. In the discussion section, can the authors speculate on how they think H3.3 ASVM is getting incorporated if not through Hira. Are there other known H3 variant chaperones, or can the core histone chaperone substitute?

We have expanded our discussion to include the the following:

“In the case of the chimeric histone proteins the incorporation behavior was dependent on the chaperone binding site. For example, H3.3^{ASVM} import and incorporation was similar to H3 in control embryos and H3.3^{ASVM} was still incorporated in Hira^{ssm} mutants. This is consistent with the chaperone binding site determining the chromatin incorporation pathway and suggests that H3.3^{ASVM} likely interacts with H3 chaperones such as Caf1.” lines 280-285

Revision Plan

Minor comments:

While the paper is well written, I found the figures very confusing and difficult to interpret. Comments here are meant to make it easier to interpret.

1. Fig 1 and most of the paper would benefit from a schematic of early embryo transitions labelled with time and stages of cell cycle to make interpreting data easier

This is an excellent suggestion! We have added a new figure (Figure 1) to explain both the biological system and the way that we measured many properties in this paper.

2. Fig 1- same green color is used for nuclear cycle 12 and for H3.3 making it confusing when reading graphs. Please check other figures where there is a similar use of color for two different things

We have changed the colors so that they are more distinct.

3. Fig 1C,D might benefit more from being split up into 3 graphs by cell cycle with H3 and H3.3 plotted on the same graphs rather than the way it is now

We do not feel that it is appropriate to directly compare the intensities of the H3-Dendra2 construct expressed from the pseudo-endogenous locus to the H3.3 and chimeric proteins expressed from the H3.3A locus as they were imaged using different settings. These curves can be directly compared within a construct and we can evaluate their trends over time, but the normalized values should not be directly compared in the way that would be encouraged by plotting the data as suggested.

4. Line 130-133: can they also comment on the different between SVM and ASVM. It seems like SVM might be even worse than ASVM (Fig 2C). Is this related to chk1 phosphorylation?

We think that this is a property of the mixed chimeras since S31A is also imported less efficiently than H3.3 (though we cannot be sure without further experiments). We have added this explanation to the text:

“We speculate that chimeric histone proteins (H3.3^{S31A} and H3.3^{SVM}) are not as efficiently handled by the chaperone machinery as species that are normally found in the organism including H3.3^{ASVM} which is protein-identical to H3.” lines 150-152

5. Fig 2F-G: It is very difficult to compare between histones when they are on different graphs, please consider putting H3, H3.3 and H3.3ASVM in a histogram background on the same graph.

We have done this in the new Figure 3F.

Revision Plan

6. Fig 3- move G to become A and then have A and B.

We have restructured this figure to include the nuclear density map of control in response to a comment from Reviewer 1. Although not exactly what the reviewer has envisioned, we hope that this adds clarity to the figure.

7. The initial slope graphs in 4D, E, H and I are not easy to understand and would benefit from an explanation in the legend.

We have edited the legend of Figure 5D (formerly 4D) and S1F which now read:

“Initial slopes of nuclear import curves (change in total nuclear intensity over time for the first 5 timepoints) ...”

In addition we have updated the methods to include:

“Import rates were calculated by using a linear regression for the total nuclear intensity over time for the first 5 timepoints in the nuclear import curves.” lines 471-473, methods

Reviewer #3 (Significance (Required)):

This paper addresses an important and understudied question- how do histones and their variants mediate chromatin regulation in the early embryo before zygotic genome activation? The authors follow up on some previous findings and provide new insights using clever genetics and cell biology in *Drosophila melanogaster*. However, the authors do not directly look at chromatin structural changes using existing genomic tools. This may be beyond the scope of this work but would make for a nice addition to strengthen their claims if they can implement these chromatin accessibility techniques in the early embryo.

Histones affect a majority of biological processes and understanding their role in the early embryo is key to understanding development. I believe this study applies to a broad audience interested in basic science. However, I do think the authors might benefit from a more broad discussion of their results to attract a broad readership.

Dear Prof. Amodeo,

Thank you for the transfer of your reviewed manuscript to EMBO reports and for your revision plan. I am sorry for my delayed reply, I was attending a conference and traveling the past days.

I have looked at your revision plan now and I agree with your suggestions for how to revise your study. I would thus like to invite you to revise your manuscript with the understanding that the referee concerns must be fully addressed and their suggestions taken on board. Please address all referee concerns in a complete point-by-point response. Acceptance of the manuscript will depend on a positive outcome of a second round of review. It is EMBO reports policy to allow a single round of major revision only and acceptance or rejection of the manuscript will therefore depend on the completeness of your responses included in the next, final version of the manuscript.

We realize that it is difficult to revise to a specific deadline. In the interest of protecting the conceptual advance provided by the work, we recommend a revision within 3 months (20th Jun 2025). Please discuss the revision progress ahead of this time with the editor if you require more time to complete the revisions.

- 1) A data availability section providing access to data deposited in public databases is missing. If you have not deposited any data, please add a sentence to the data availability section that explains that.
- 2) Your manuscript contains statistics and error bars based on $n=2$. Please use scatter blots in these cases. No statistics should be calculated if $n=2$.

5) a complete author checklist, which you can download from our author guidelines . Please insert information in the checklist that is also reflected in the manuscript. The completed author checklist will also be part of the RPF.

6) Please note that all corresponding authors are required to supply an ORCID ID for their name upon submission of a revised manuscript (. Please find instructions on how to link your ORCID ID to your account in our manuscript tracking system in our Author guidelines

7) Before submitting your revision, primary datasets produced in this study need to be deposited in an appropriate public database (see <https://www.embopress.org/page/journal/14693178/authorguide#datadeposition>). Please remember to provide a reviewer password if the datasets are not yet public. The accession numbers and database should be listed in a formal "Data Availability" section placed after Materials & Method (see also <https://www.embopress.org/page/journal/14693178/authorguide#datadeposition>). Please note that the Data Availability Section is restricted to new primary data that are part of this study. * Note - All links should resolve to a page where the data can be

accessed. *

10) Regarding data quantification (see Figure Legends:

<https://www.embopress.org/page/journal/14693178/authorguide#figureformat>)

- the name of the statistical test used to generate error bars and P values,
- the number (n) of independent experiments (please specify technical or biological replicates) underlying each data point,
- the nature of the bars and error bars (s.d., s.e.m.),
- If the data are obtained from n Program fragment delivered error ``Can't locate object method "less" via package "than" (perhaps you forgot to load "than"?) at //ejpvfs23/sites23b/embor_www/letters/embor_decision_rc_revise_and_rereview.txt line 56.' 2, use scatter blots showing the individual data points.

12) All Materials and Methods need to be described in the main text using our 'Structured Methods' format, which is required for all research articles. According to this format, the Methods section includes a Reagents and Tools Table (listing key reagents, experimental models, software and relevant equipment and including their sources and relevant identifiers) followed by a Methods and Protocols section describing the methods using a step-by-step protocol format. The aim is to facilitate adoption of the methodologies across labs. More information on how to adhere to this format as well as a downloadable template (.docx) for the Reagents and Tools Table can be found in our author guidelines:

An example of a Method paper with Structured Methods can be found here: <https://www.embopress.org/doi/full/10.1038/s44320-024-00037-6#sec-4>

I look forward to seeing a revised form of your manuscript when it is ready.

Esther Schnapp, PhD

Senior Editor
EMBO reports

Manuscript number: EMBOR-2025-61506

Corresponding author(s): Amanda Amodeo

1. General Statements

*We thank the reviewers for the time and care that went into their comments. We have provided our response in two forms. First, we have placed each of our responses to individual comments into the categories (new revisions, previous revisions, not planned). Second, we have also **provided the same comments in their original order at the end of the document** so that the reviewers can find all of their own comments in one place. Throughout the response we use **black for reviewer** comments, **blue for our responses at the time of transfer**, **green for responses added at resubmission**, and **red for new text added to the manuscript**. We hope that this makes re-assessing the work as efficient as possible.*

2. Description of new revisions

Reviewer #1

If the embryos appeared "reasonably healthy" (line 113) after slbp RNAi, how do the authors know that the RNAi was effective, especially in THESE embryos, given siblings had clear and drastic phenotype? This is especially critical given that the authors find no effect on H3.3 incorporation after slbp RNAi (and presumably H3 reduction), but this result would also be observed if the slbp RNAi was just not effective in these embryos.

We apologize for the confusion caused by our word choice. The "healthy" slbp-RNAi embryos had measurable phenotypes consistent with histone depletion that we have reported previously (Chari et al, 2019) including cell cycle elongation and early cell cycle arrest (Figure EV5E). However, they did not have the catastrophic mitosis observed in more severely affected embryos. We agree with the reviewer that a concern of this experiment is that the less severely affected embryos likely have more remaining RD histones including H3. To address this we also tested H3.3 incorporation in the embryos that fail to progress to later cell cycles in the cycles that we could measure. Even in these more severely affected embryos we were not able to detect a change in H3.3 incorporation relative to controls (lines 242-246 and Fig EV5B). Unfortunately, it is impossible to conduct the ideal experiment, which would be a complete removal of H3 since this is incompatible with oogenesis and embryo survival.

To address this confusion we have added supplemental videos of control, moderately affected and severely affected SLBP-RNAi embryos as movies 3-5 and modified the text to read:

"All embryos that survive through at least NC12, had elongated cell cycles in NC12 and 60% arrested in NC13 as reported previously indicating the effectiveness of the knockdown (Fig EV5E, Movie EV3-5)³⁹. In these embryos, H3.3 incorporation is largely unaffected by the reduction in RD H3 (Fig 6B)." lines 239-242

Finally, to characterize the range of SLBP knockdown in the RNAi embryos we propose to do single embryo RT-qPCRs for SLBP mRNA for multiple individual embryos. This will provide a measure of the range knockdown that we observed in our H3.3 movies.

We have completed the RT-qPCR experiments to measure the variability of the SLBP RNAi knockdown and found that all embryos exhibit a substantial reduction in the amount of SLBP RNA (corresponding to 126 to 424 times less Slbp RNA compared to control embryos). These data are included in the new extended figure EV5C.

How is H3.3 expression controlled? Is it possible that H3.3 biosynthesis is affected in Chk1 mutants?

To address this question we propose to perform RT-qPCR for H3.3A and H3.3B as well as Hira in the Chk1 mutant. Unfortunately, we do not have antibodies that reliably distinguish between H3 and H3.3 in our hands (despite literature reports), but we will also perform a pan-H3 immunostaining in the Chk1 embryos to measure how the total H3-type histone pool changes as a result of the loss of Chk1.

We performed RT-qPCR for H3.3A, H3.3B and Hira in NC12 Chk1 mutant, H3.3-Dendra2 embryos (same genotype that we imaged) and found no statistically significant differences for any of these components compared to H3.3-Dendra2 controls. These data are now added to EV5 and give us confidence that the differences in H3.3 behavior observed in the Chk1 embryos are not due to changes in H3.3 production at the RNA level. We have added the following line to the text:

“Although these mutants have normal H3.3 mRNA deposition (Fig EV5F-H), the lack of checkpoints causes an unusually rapid NC13” lines 263-264

We also performed pan-H3 immunostaining in the Chk1 mutant background and found that the total H3 on chromatin in NC12 was much more variable than in control embryos. Nonetheless, we were unable to detect a statistically significant difference in the total amount of all H3-type histones on chromatin. We have added the following sentence to the results:

“However, the total H3-type histone deposition on chromatin was much more variable in NC12 Chk1 mutant embryos (Fig EV5K,M).” lines 269-270

Fig 5:

F: OPTIONAL Experimental request: Here I would like to see H3 as a control.

This is a very good suggestion, and we are currently imaging H3-Dendra2 in the Chk1 background. However, our preliminary results suggest that there may be some synthetic early lethality between the tagged H3-Dendra2 and Chk1 since these embryos are much less healthy than H3.3-Dendra2 Chk1 embryos or Chk1 with other reporters. In addition, we have observed a much higher level of background fluorescence in this cross than in the H3-Dendra2 control. We are uncertain if we will be able to obtain usable data from this experiment, but will continue to try to find conditions that allow us to analyze this data.

Unfortunately, the synthetic lethality prevented us from obtaining a sufficient number of embryos to complete this experiment.

As an orthogonal approach to answer the question, we will perform immunostaining with a pan-H3 antibody in Chk1 mutant embryos to measure total H3 levels under these conditions. Since the majority of H3-type histone is H3.2 and we know how H3.3 changes, this staining will give us insight into the dynamics of H3 in Chk1 mutant embryos.

We have completed the proposed pan-H3 immunostaining. At the same time we have also quantified the amount of DNA using Hoechst. To conduct these experiments we needed to capture nuclei during mitosis since we know that a large percent of the total H3 in the nucleus during interphase is not bound to DNA. Since mitosis in these cycles is only ~2-3 minutes in length, mitotic embryos make up a very small percent of the total embryos collected.

We found no statistically significant differences in the amount of DNA in NC12 in Chk1 mutant vs yw control embryos. There was no statistically significant decrease in the amount of pan-H3 staining in the Chk1 embryos compared to controls, however pan-H3 staining was much more variable in the mutant. We have incorporated these results into the new figure EV5J-M. We have also added the following sentences to the text:

“The decreased H3.3 incorporation was likely not due to DNA underreplication since Hoechst staining shows no significant decrease in NC12 (Fig EV5J,L). However, the total H3-type histone deposition on chromatin was much more variable in NC12 Chk1 mutant embryos (Fig EV5K,M).” lines 267-270

Reviewer #2

In Figure 2, the manuscript introduces chimeric embryos expressing modified histone variants, but their developmental viability is not addressed. It is essential to determine whether these embryos survive and whether they exhibit any phenotypic consequences such as altered hatching rates, defects in nuclear division, or developmental arrest.

Tagging histones is often deleterious to organismal health. In Drosophila there are two H3.3 loci (H3.3A and H3.3B). In all of our chimera experiments we have left the H3.3B and one copy of the H3.3A locus unperturbed to provide a supply of untagged H3.3. This allows us to study H3.3 and chimera dynamics without compromising organism health. All of our chimeras are viable and fertile with no obvious morphological defects. We have added the following sentences to the text to clarify this point:

“There are ~100 copies of H3 in the Drosophila genome, but only 2 of H3.3 (H3.3A and H3.3B)²⁶. To determine which factor controls nuclear availability and chromatin incorporation, we genetically engineered flies to express Dendra2-tagged H3/H3.3 chimeras at the endogenous

H3.3A locus, keeping the H3.3B locus intact....These chimeras were all viable and fertile.” lines 128-132, 137

In addition we propose performing hatch rate assays for embryos from the chimeric embryos of S31A, SVM and ASVM to assess if there is any decrease in fecundity due to the presence of the chimeras.

We have now included a 5 day hatch rate assay for all of the H3.3A chimera lines compared to H3.3-Dendra2 and yw (wildtype) embryos. There is no clear defect in the fertility of any of the lines. These data are included in EV2I and table EV9.

Figures 3 and 4 raise additional concerns about whether histone cluster transcription is altered in shkl mutant embryos. The authors propose that the shkl mutation affects the N/C ratio, yet it remains unclear whether this leads to changes in the transcription of histone clusters. Furthermore, since HIRA is a key chaperone for H3.3, it would be important to assess whether its levels or function are compromised in shkl mutants. To address these gaps, RT-qPCR or RNA-seq should be performed to quantify histone cluster transcription, and Western blot analysis should be used to determine if HIRA protein levels are affected.

The changes in the N/C ratio that are observed in the shkl mutant are within SINGLE embryo (differences in nuclear spacing). In these experiments we are comparing nuclei within a common cytoplasm that have different local nuclear densities (N/C ratios). Therefore, if Shkl were somehow affecting the transcription of histones or their chaperones we would expect all of the nuclei within the same mutant embryo to be equally affected since they are genetically identical and share a common cytoplasm. We do not directly compare the behavior of shkl embryos to wildtype except to demonstrate that there is no positional effect on the import of H3 and H3.3 across the length of the embryo in wildtype. To clarify our experimental system for these experiments we have added additional panels to Figure 4A and B that depict the number of neighbors for both control and Shkl embryos.

Nonetheless, to address the reviewer’s concern that shkl may change the amount of H3 present in the embryo, we propose to conduct a western blot comparison of wildtype and shkl embryos using a pan-H3 antibody. There are no tools (antibodies or fluorescently tagged proteins) to assess HIRA protein levels in Drosophila. We therefore propose to perform RT-qPCR for Hira in wildtype and shkl embryos.

We have added the western blot for pan-H3 (including quantification) and RT-qPCR of Hira in the Shkl background to EV3D-F. There is no statistically significant difference in total H3 protein or Hira mRNA in this background compared to H3-Dendra embryos without the Shkl mutation. We have added the following sentence to the text:

“Although the nuclear densities within the embryos are altered, the shkl mutation does not affect the deposition of total H3 proteins or Hira mRNA (Figure EV3D-F).” lines 187-188

We note that it has come to our attention that a *Drosophila* Hira antibody has recently been generated, but we were not able to obtain it in time for this resubmission.

A similar issue arises in Figure 5, where the authors claim that H3.3 incorporation is dependent on cell cycle state but do not sufficiently test whether this is linked to changes in HIRA levels. Given the importance of HIRA in H3.3 deposition, its levels should be examined in *Slbp*, *Zelda*, and *Chk1* RNAi embryos to verify whether changes in H3.3 incorporation correlate with HIRA function. Without this, it is difficult to conclude that the observed effects are strictly due to cell cycle regulation rather than histone chaperone dynamics.

*Since H3.3 incorporation is unaffected in the Slbp and Zelda-RNAi lines there is no reason to suspect a change in Hira function. There are no available tools (antibodies or fluorescently tagged proteins) to directly measure Hira protein in Drosophila. To test if changes in HIRA loading might contribute to the decreased H3.3 incorporation in the Chk1 mutant **we propose to perform RT-qPCR for Hira in wildtype and Chk1 embryos.***

We performed RT-qPCR for H3.3A, H3.3B and Hira in NC12 Chk1 mutant, H3.3-Dendra2 embryos (same genotype that we imaged) and found no statistically significant differences for any of these components compared to H3.3-Dendra2 controls. These data are now added to EV5. We have added the following line to the text:

“Although these mutants have normal H3.3 mRNA deposition (Fig EV5F-H), the lack of checkpoints causes an unusually rapid NC13” lines 263-264

We note that it has come to our attention that a *Drosophila* Hira antibody has recently been generated, but we were not able to obtain it in time for this resubmission.

Additionally, the manuscript does not include any validation of the RNAi knockdown efficiencies used throughout the study. This raises concerns about whether the observed phenotypes are truly due to target gene depletion or off-target effects. RT-qPCR or Western blot analyses should be performed to confirm knockdown efficiency.

Both the Zelda and slbp-RNAi lines used for knockdowns have been used and validated in the early fly embryo in previously published works ((Yamada et al., 2019), (Duan et al., 2021), (O’Haren et al., 2025), (Chari et al, 2019)) and the phenotypes that we observe in our embryos are consistent with the published data including altered cell cycle durations (Figure EV5E) and lack of cellularization/gastrulation. We note that the zelda RNAi phenotypes are also highly consistent with the effects of Zelda germline clones. To validate that slbp-RNAi knocks down histones we included a western blot for Pan-H3 in slbp-RNAi embryos that demonstrates a large effect on total H3 levels (Figure EV5A). To further demonstrate the phenotypic effects of the slbp-RNAi we have added supplemental movies (Movies EV4 and EV5).

To fully characterize the RNAi efficiency under our conditions **we propose to perform RT-qPCR for slbp in slbp-RNAi and Zelda in Zelda-RNAi compared to control (w) RNAi embryos.**

We have conducted the requested RT-qPCRs to validate the knockdown of Slbp and zelda. We see an average of ~200 fold reduction of Slbp RNA in the Slbp-RNAi line and zelda is reduced to undetectable levels by qPCR in % of our zelda-RNAi samples which prevents quantification. These results have been added to figure EV5C-D.

Reviewer #3

Major comments:

1. The claims are largely supported by the data but I think a couple more experiments could help bolster the claims about cell cycle and chk1 regulation.

a. Creating a phosphomimetic of the chk1 phosphorylation site on H3.3 to see if it can overcome the defects seen in chk1 mutants

b. Assessing heterochromatin of embryos without chk1 (or ASVM mutants) for example, by looking at H3K9me3 levels

The first experiments could take several months if the flies haven't already been generated by the authors but the second should be quicker.

a. This is an excellent experimental suggestion which is bolstered by the fact that in frogs H3.3 S31A cannot rescue H3.3 morpholino during gastrulation, but H3.3S31D can (Sitbon et al, 2020). However, to correctly conduct this experiment would require generating and validating multiple additional endogenous H3.3 replacement lines, likely without a fluorescent tag as they can interfere with histone rescue constructs in most species. As the reviewer notes, this would take several months of work (we have not generated the critical flies yet) and may not yield a satisfying answer since there are reports that H3.3 may be dispensable in flies aside from as a source of H3-type histone outside of S-phase (Hödl and Bassler, 2012). While we hope to continue experiments along these lines in the future we feel that this is beyond the scope of the current manuscript.

*b. To address this **we propose to stain for H3K9me3 in wildtype and Chk1[±] embryos.** Since the ASVM line is not a full replacement of all H3.3 we think that staining for H3K9me3 in this line is unlikely to yield a detectable difference.*

We have completed the requested staining, however, the results are difficult to interpret. We chose to focus on mitosis of NC12 since that is the earliest timepoint where we see a detectable difference in H3.3 chromatin incorporation and the reported underreplication of DNA in NC13 would complicate analysis in that cycle. Our analysis was confounded by very low signal to noise in our staining even after optimization. This problem was worse in the Chk1 embryos than controls. Upon further investigation, these results are consistent with previous reports by CUT&Tag which indicate that H3K9me3 levels are very low on chromatin prior to NC14 in the fly

embryo (Atinbayeva et al, 2024). We have therefore included the following sentence to the discussion, but prefer not to include our staining data in the manuscript as we do not believe that quantification of these images will yield reliable results. We include representative images below for the reviewer's perusal.

"However, it is unlikely that the effect of Chk1 on H3.3 incorporation is directly due to loss of heterochromatin since the H3K9me3 mark only becomes prominent at NC13 in wildtype embryos⁷⁴." lines 307-309

2. It would also be interesting to see what the health of the flies with some mutations in this paper are beyond the embryo stage if they are viable (e.g., development to adulthood, fertility etc.)

- a. the SVM, ASVM mutations
- b. the hira + ASVM mutations

The authors might already have this data but if not they have the flies and it shouldn't take long to get these data.

a. To address this concern we propose to conduct hatch rate assays for embryos from the Dendra tagged H3.3, S31A, SVM, ASVM flies. However, we do note that in our experiments only one copy of the H3.3A locus was mutated and tagged with Dendra2 leaving one copy of

H3.3A and both copies of H3.3B untouched to ensure normal development as tagging all copies of histone genes can lead to lethality.

We have now included a 5 day hatch rate assay for all of the H3.3A chimera lines compared to H3.3-Dendra2 and yw (wildtype) embryos. There is no clear defect in the fertility of any of the lines. These data are included in EV2I and table EV9.

b. All Hira mutants develop as haploids due to the inability to decondense the sperm chromatin (which is dependent on Hira). This leads to one extra division to restore the N/C ratio prior to cell cycle slowing and ZGA. These embryos go on to gastrulate and die late in development after cuticle formation (presumably due to their decreased ploidy) (Loppin et al., 2000). The addition of ASVM into the Hira background does not appear to rescue the ploidy defect as these embryos also undergo the extra division (Figure 3H). We are therefore confident that these embryos will not hatch. We have added the information about the development of Hira mutant to the text as follow:

“These embryos develop as haploids and undergo one additional syncytial division before ZGA (NC14). Hira^{ssm}embryos develop otherwise phenotypically normally through organogenesis and cuticle formation, but die before hatching⁵⁸.” lines 165-168

3. Description of the revisions from previous submission

Reviewer #1

Evidence, reproducibility and clarity

Summary:

Bhatt et al. seek to define factors that influence H3.3 incorporation in the embryo. They test various hypotheses, pinpointing the nuclear/cytoplasmic ratio and Chk1, which affects cell cycle state, as influencers. The authors use a variety of clever Drosophila genetic manipulations in this comprehensive study. The data are presented well and conclusions reasonably drawn and not overblown. I have only minor comments to improve readability and clarity. I suggest two OPTIONAL experiments below.

We thank the reviewer for their positive and helpful comments.

Major comments:

We found this manuscript well written and experimentally thorough, and the data are meticulously presented. We have one modification that we feel is essential to reader understanding and one experimental concern:

The authors provide the photobleaching details in the methodology, but given how integral this measurement is to the conclusions of the paper, we feel that this should be addressed in clear prose in the body of the text. The authors explain briefly how nuclear export is assayed, but not import (line 99). Would help tremendously to clarify the methods here. This is especially important as import is again measured in Fig 4. This should also be clarified (also in the main body and not solely in the methods).

We have added the following sentences to the main body of the text to clarify how photobleaching and import were assayed.

“We note that these differences are not due to photobleaching as our measurements on imaged and unimaged embryos indicate that photobleaching is negligible under our experimental conditions (see methods, Fig EV1G-H).” lines 100-102

and

“Since nuclear export is effectively zero, we attribute the increase in total H3.3 over time solely to import and therefore the slope of total H3.3 over time corresponds to the import rate.” lines 111-113

In addition we have clarified how import was calculated to figure legends in Figure 5D (formerly 4D) and S1F which now read:

“Initial slopes of nuclear import curves (change in total nuclear intensity over time for the first 5 timepoints) ...”

We also added the following explanation of how nuclear import rates were calculated to the methods section:

“Import rates were calculated by using a linear regression for the total nuclear intensity over time for the first 5 timepoints in the nuclear import curves.” lines 492-493, methods

If the embryos appeared "reasonably healthy" (line 113) after slbp RNAi, how do the authors know that the RNAi was effective, especially in THESE embryos, given siblings had clear and drastic phenotype? This is especially critical given that the authors find no effect on H3.3 incorporation after slbp RNAi (and presumably H3 reduction), but this result would also be observed if the slbp RNAi was just not effective in these embryos.

We apologize for the confusion caused by our word choice. The “healthy” slbp-RNAi embryos had measurable phenotypes consistent with histone depletion that we have reported previously (Chari et al, 2019) including cell cycle elongation and early cell cycle arrest (Figure EV5E). However, they did not have the catastrophic mitosis observed in more severely affected embryos. We agree with the reviewer that a concern of this experiment is that the less severely affected embryos likely have more remaining RD histones including H3. To address this we also tested H3.3 incorporation in the embryos that fail to progress to later cell cycles in the cycles that we could measure. Even in these more severely affected embryos we were not able to detect a change in H3.3 incorporation relative to controls (lines 242-246 and Fig EV5B). Unfortunately, it is impossible to conduct the ideal experiment, which would be a complete removal of H3 since this is incompatible with oogenesis and embryo survival.

To address this confusion we have added supplemental videos of control, moderately affected and severely affected SLBP-RNAi embryos as movies 3-5 and modified the text to read:

“All embryos that survive through at least NC12, had elongated cell cycles in NC12 and 60% arrested in NC13 as reported previously indicating the effectiveness of the knockdown (Fig EV5E, Movie EV3-5)³⁹. In these embryos, H3.3 incorporation is largely unaffected by the reduction in RD H3 (Fig 6B).” lines 239-242

Minor comments:

Introduction:

Consider using "replication dependent" (RD) rather than "replication coupled." Both are used in the field, but RD parallels RI ("replication independent").

We thank the reviewer for this suggestion. We have made the text edits to change "replication coupled" (RC) to "replication dependent" (RD) throughout the manuscript.

Would help for clarity if the authors noted that H3 is equivalent to H3.2 in Drosophila. Also it is relevant that there are two H3.3 loci as the authors knock mutations into the H3.3A locus, but leave the H3.3B locus intact. The authors should clarify that there are two H3.3 genes in the Drosophila genome.

We have changed the text as follows to increase clarity as suggested:

“Similarly, we have previously shown that RD H3.2 (hereafter referred to as H3) is replaced by RI H3.3 during these same cycles, though the cause remains unclear²⁹” lines 52-54

“There are ~100 copies of H3 in the Drosophila genome, but only 2 of H3.3 (H3.3A and H3.3B)²⁶. To determine which factor controls nuclear availability and chromatin incorporation, we genetically engineered flies to express Dendra2-tagged H3/H3.3 chimeras at the endogenous H3.3A locus, keeping the H3.3B locus intact.” lines 128-132

Please add information and citation (line 58): H3.3 is required to complete development when H3.2 copy number is reduced (PMID: 37279945, McPherson et al. 2023)

We have added the suggested information. The text now reads

“Nonetheless, H3.3 is required to complete development when H3 copy number is reduced⁵⁴.” lines 61-62

Results:

Embryo genotype is unclear (line 147): Hira^[ssm] haploid embryos inherit the Hira mutation maternally? Are Hira homozygous mothers crossed to homozygous fathers to generate these embryos, or are mothers heterozygous? This detail should be in the main text for clarity.

The Hira mutants are maternal effect. We crossed homozygous Hira^{ssm} females to their hemizygous Hira^{ssm} or FM7C brothers. However, the genotype of the male is irrelevant since the

Hira phenotype prevents sperm pronuclear fusion and therefore there is no paternal contribution to the embryonic genotype. We have clarified this point in the text:

“We generated embryos lacking functional maternal Hira using Hira^{ssm-185b} (hereafter Hira^{ssm}) homozygous mothers which have a point mutation in the Hira locus⁵⁸.” lines 161-163

Line 161: Shkl affects nuclear density, but it also appears from Fig 3 to affect nuclear size? The authors do not address this, but it should at least be mentioned.

We thank the reviewer for the astute observation. More dense regions of the Shkl embryos do in fact have smaller nuclei. We believe that this is a direct result of the increased N/C ratio since nuclear size also falls during normal development as the N/C ratio increases. We have added a new figure 1 in which we more carefully describe the events of early embryogenesis in flies including a quantification of nuclear size and number in the pre-ZGA cell cycles (Figure 1C). We also note the correlation of nuclear size with nuclear density in the text:

“During the pre-ZGA cycles (NC10-13), the maximum volume that each nucleus attains decreases in response to the doubling number of nuclei with each division (Fig 1C).” lines 87-89

“To test this, we employed mutants in the gene Shackleton (shkl) whose embryos have non-uniform nuclear densities and therefore a gradient of nuclear sizes across the anterior/posterior axis (Fig 4A-B, Movie EV1-2)⁵⁹” lines 181-184

The authors often describe nuclear H3/H3.3 as chromatin incorporated, but these image-based methods do not distinguish between chromatin-incorporated and nuclear protein.

To distinguish between chromatin incorporated and nuclear free histone we have exploited the fact that histones that are not incorporated into DNA freely diffuse away from the chromatin mass during mitosis while those that are bound into nucleosomes remain on chromatin during this time. In our previous study we showed that H3-Dendra2 that is photoconverted during mitosis remains stably associated with the mitotic chromatin through multiple cell cycles (Shindo and Amodeo, 2019) strengthening our use of this metric. To help clarify this point as well as other methodological details we have added a new Figure 1B which documents the time points at which we make various measurements within the lifecycle of the nucleus. We also edited the text to read:

“We have previously shown that with each NC, the pool of free H3 in the nucleus is depleted and its levels on chromatin during mitosis decrease (Fig 1D, EV1C-D)²⁹. In contrast, H3.3 mitotic chromatin levels increase during the same cycles (Fig 1D, EV1C-D)²⁹.” lines 91-94

I very much appreciate how the authors laid out their model in Fig 3 and then used the same figure to explain which part of the model they are testing in Figs 4 and 5. This is not a critique—we can complement too!

Thank you!

O'Haren 2024 (PMID: 39661467) did not find increased Pol II at the HLB after zelda RNAi (line 227). Might also want to mention here that zelda RNAi does not result in changes to H3 at the mRNA level (O'Haren 2024), as that would confound the model.

We thank the reviewer for the suggestion. We have removed the discussion of Pol II localization and replaced it with the information about histone mRNA :

“zelda controls the transcription of the majority of Pol II genes during ZGA but disruption of zelda does not change RD histone mRNA levels⁶⁸⁻⁷¹.” lines 252-254

Discussion:

Should discuss results in context of McPherson et al. 2023 (PMID: 37279945), who showed that decreasing H3.2 gene numbers does not increase H3.3 production at the mRNA or protein levels.

We expanded our discussion to include the following:

“Given the fact that H3.3 pool size does not respond to H3 copy number in other Drosophila tissues⁵⁴, our results suggest that H3.3 incorporation dynamics are likely independent of H3 availability.” lines 286-288

The Shackleton mutation is a clever way to alter N/C ratio, but the authors should point out that it is difficult (impossible?) to directly and cleanly manipulate the N/C ratio. For example, Shkl mutants seem to also have various nuclear sizes.

As discussed above, we think that nuclear size is a direct response to the N/C ratio. We have added the following sentence to the discussion as well as a citation to a paper which discusses how the N/C ratio might contribute to nuclear import in early embryos to the discussion:

“This may be due to N/C ratio-dependent changes in nuclear import dynamics which may also contribute to the observed changes in nuclear size across the shkl embryo⁷⁸.” lines 318-320

Figures:

While I appreciate the statistical summaries in tables, it is still helpful to display standard significance on the figures themselves.

We have added statistical comparisons in Figure 3 (formerly Figure 2). We do not feel that it is appropriate to directly compare the intensities of the H3-Dendra2 construct expressed from the pseudo-endogenous locus to the H3.3 and chimeric proteins expressed from the H3.3A locus as they were imaged using different settings. Although we plot H3 on the same graph as the other proteins to allow for ease of comparison of their trends over time it is not appropriate to directly compare their normalized intensities which including statistical tests would encourage. We have added a note to the legend of Figure 1 explaining this which reads:

“Note that statistical comparisons between the two Dendra2 constructs have not been done as they were expressed from different loci and imaged under different experimental settings.”

Fig 1:

A: Is it possible to label panels with the nuclear cycle? *We have done this.*

E: The comparison in the text is between H3.3 and H3, but only H3.3 data is shown. I realize that it is published prior, but the comparison in figure would be helpful.

We have added the previously published values to the text.

“These changes in nuclear import and incorporation result in a less complete loss of the free nuclear H3.3 pool (~70% free in NC11 to ~30% in NC13) than previously seen for H3 (~55% free in NC11 to ~20% in NC13)” lines 117-120

Fig 2:

A: A very helpful figure. Slightly unclear that the H3 that is not Dendra tagged is at the H3.3 locus. Also unclear that the H3.3A-Dendra2 line exists and used as control, as is not shown in figure.

Should show H3 and H3.3 controls (Figure S2)

We have edited the figure to add Dendra2 to all of the constructs and made clear the location of each construct including adding the landing site for H3-Dendra2. We have also cited Figure EV1 in the legend which contains a more detailed diagram of the integration strategy.

F/H- As the comparison is between H3 and ASVM, it would help to combine these data onto the same graph. As the color is currently used unnecessarily to represent nuclear cycle, the authors could use their purple/pink color coding to represent H3/ASVM. *We have combined these data onto a single graph as requested and changed the colors appropriately. We have not added statistical comparisons to this graph as we again believe that they would be inappropriate.*

In the legend of Fig 2 the authors write "in the absence of Hira." Technically, there is only a point mutation in Hira. It is not absent.

Good catch! We have changed this to “in Hira^{ssm} mutants”.

Fig 3:

G: Please show WT for comparison. Can use data in Fig 3A. *We have added the color-coded number of neighbor embryo representations for WT and Shkl embryos underneath the example embryo images in 4A-B (formerly 3A-B,G).*

Model in H is very helpful (complement)! *Thank you.*

Fig 4:

D/E/H/I: What does "min volume" mean on the X axis?

Since the uneven N/C ratio in the shkl embryos results in a wavy cell cycle pattern there is no single time point where we can calculate the number of neighbors for the whole embryo (since not all nuclei are in the same cell cycle at a given point). Therefore, we had to choose a criterion

for when we would calculate the number of neighbors for each nucleus. We chose nuclear size as a proxy for nuclear age since nuclear size increases throughout interphase (see new figure 1B). So, the minimum volume is the newly formed nucleus in a given cell cycle. We also tested other timepoints for the number of neighbors (maximum nuclear volume, just before nuclear envelope breakdown and midway between these two points) and found similar results. We chose to use minimum volume in this paper because this is the time point when the nucleus is growing most quickly and nuclear import is at its highest. We have added the following explanation to the methods:

“For shkl embryos, as the nuclear cycles are asynchronous, nuclear divisions start at different timepoints within the same cell cycle and the nuclear density changes as the neighboring nuclei divide. Therefore, the total intensity traces were aligned to match their minimum volumes (as shown in Fig 1B) to T0” lines 505-508, methods

And the following detail to the figure legend:

“...plotted by the number of nuclear neighbors at their minimum nuclear volume...” Figure 5 legend

We also added a depiction of the lifecycle of the nucleus in which we marked the minimum volume as the new Figure 1B.

Reviewer #2 (Evidence, reproducibility and clarity (Required)):

This manuscript investigates the regulation of H3.3 incorporation during zygotic genome activation (ZGA) in *Drosophila*, proposing that the nuclear-to-cytoplasmic (N/C) ratio plays a central role in this process. While the study is conceptually interesting, several concerns arise regarding the lack of proper control experiments and the clarity of the writing. The manuscript is difficult to follow due to vague descriptions, insufficient distinctions between established knowledge and novel findings, and a lack of rigorous statistical analyses. These issues need to be addressed before the study can be considered for publication.

We thank the reviewers for their careful reading of this manuscript. We have sought to clarify the concerns regarding clarity through numerous text edits detailed below. We did include ANOVA analysis for all of the relevant statistical comparisons in the supplemental table. However, to increase clarity we have also added some statistical comparisons in the main figures. We note that we do not feel that it is appropriate to directly compare the intensities of the H3-Dendra2 construct expressed from the pseudo-endogenous locus to the H3.3 and chimeric proteins expressed from the H3.3A locus as they were imaged using different settings. Although we plot H3 on the same graph as the other proteins to allow for ease of comparison of their trends over time it is not appropriate to directly compare their normalized intensities which including statistical tests would encourage. We have added a note to the legend of the new Figure 1 explaining this which reads:

“Note that statistical comparisons between the two Dendra2 constructs have not been done as they were expressed from different loci and imaged under different experimental settings.”

Major Concerns

The manuscript would benefit from a clearer introduction that explicitly distinguishes between previously known mechanisms of histone regulation during ZGA and the novel contributions of this study. Currently, the introduction lacks sufficient background on early embryonic chromatin regulation, making it difficult for readers unfamiliar with the field to grasp the significance of the findings. The authors should also be more precise when discussing the timing of ZGA. While they state that ZGA occurs after 13 nuclear divisions, it is well established that a minor wave of ZGA begins at nuclear cycle 7-8, whereas the major wave occurs after cycle 13. Clarifying this distinction will improve the manuscript's accessibility to a broader audience.

We have added a new figure 1 to make the timing and nuclear behaviors of the embryo during ZGA in Drosophila more clear. We have also added information about how the chromatin changes during Drosophila ZGA in the following sentence:

“In Drosophila, these changes include refinement of nucleosomal positioning, partitioning of euchromatin and heterochromatin and formation of topologically associated domains^{20,21,23,25}.”
lines 39-41

We have clarified the major and minor waves of ZGA in the introduction and results by adding the following sentences to the introduction and results respectively:

“In most organisms ZGA happens in multiple waves but the chromatin undergoes extensive remodeling to facilitate bulk transcription during the major wave of ZGA (hereafter referred to as ZGA)¹⁸⁻²⁴.” *lines 36-39*

“In Drosophila, ZGA occurs in 2 waves. The minor wave starts as early as the 7th cycle, while major ZGA occurs after 13 rapid syncytial nuclear cycles (NCs) and is accompanied by cell cycle slowing and cellularization (Fig 1A-B).” *lines 85-87*

We hope that these changes help to reduce confusion and make the paper more accessible. However, we are happy to add additional information if the reviewer can provide specific points which require further attention.

One of the primary weaknesses of this study is the lack of adequate control experiments. In Figure 1, the authors suggest that the levels of H3 and H3.3 are influenced by the N/C ratio, but it is unclear whether transcription itself plays a role in these dynamics. To properly test this, RNA-seq or Western blot analyses should be performed at nuclear cycles 10 and 13-14 to compare the levels of newly transcribed H3 or H3.3 against maternally supplied histones. Without such data, the authors cannot rule out transcriptional regulation as a contributing factor.

In the pre-ZGA cell cycles the vast majority of protein including histones is maternally loaded. Gunesdogan et al. (2010) showed that the zygotic RD histone cluster nulls survive past NC14 (well past ZGA) with no discernible defects indicating that maternal RD histone supplies are sufficient for normal development during the cell cycles under consideration. Therefore, new transcription of replication coupled histones is not needed for apparently normal development during this period. Moreover, we have done the western blot analysis using a Pan-H3 antibody as suggested by the reviewer in our previously published paper (Shindo and Amodeo, 2019 supplemental figure S3A-B) and found that total H3-type histone proteins only increase moderately during this period of development, nowhere near the rate of the nuclear doublings. We have added the following sentence to clarify this point.

“These divisions are driven by maternally provided components and the total amount of H3 type histones do not keep up with the pace of new DNA produced²⁹.” lines 89-91

We have also previously done RNA-seq on wild-type embryos (and those with altered maternal histone levels) (Chari et al 2019). In this RNA-seq (like most RNA-seq in flies) we used poly-A selection and therefore cannot detect the RD histone mRNAs (which have a stem-loop instead of a poly-A tail). We have plotted the mRNA concentrations for both H3.3 variants from that dataset below for the reviewers reference (we have not included this in the revised manuscript). The total H3.3 mRNA levels are nearly constant from egg laying (NC0- these are from unfertilized embryos) until after ZGA (NC14). These data combined with the westerns discussed above give us confidence that what we are observing is the partitioning of large pools of maternally provided histones with only a relatively small contribution of new histone synthesis.

Several figures require additional statistical analyses to support the claims made. In Figure 1B, statistical testing should be included to validate the reported differences. Figure 1C-D states that "H3.3 accumulation reduces more slowly than H3," yet there is no quantitative comparison to substantiate this claim. Similarly, Figure 1E presents the conclusion that "These changes in nuclear import and incorporation result in a less dramatic loss of the free nuclear H3.3 pool than previously seen for H3," despite the fact that H3 data are not included in this figure. The

conclusions drawn from these data need to be supported with appropriate statistical comparisons and more precise descriptions of what is being measured.

For Figure 1B (now 2B) we do not feel that it is appropriate to directly compare the intensities of the H3-Dendra2 construct expressed from the pseudo-endogenous locus to the H3.3 and chimeric proteins expressed from the H3.3A locus as they were imaged using different settings and therefore we do not feel that direct statistical tests are appropriate. Rather, we plot the two histones on the same graph normalized to their own NC10 values so that the trend in their decrease over time may be compared. The statistical tests for H3.3 compared to the chimeras which were originally in the supplemental table have been added to Figure 3 (formerly figure 2). It is important to note that in this directly comparable situation the ASVM mutant (whose trends closely mirror H3) is highly statistically distinct from H3.3.

We have added a note to the legend of the new Figure 1 explaining this which reads:

“Note that statistical comparisons between the two Dendra2 constructs have not been done as they were expressed from different loci and imaged under different experimental settings.”

For Figure 1C-D (now 2C-D) we have removed this claim from the text. We were referring to the plateau in nuclear import for H3 that is less dramatic in H3.3, but this is more carefully discussed in the next paragraph and its addition at that point generated confusion. The text now reads:

“To further assess how nuclear uptake dynamics changed during these cycles, we tracked total nuclear H3 and H3.3 in each cycle (Fig 2C-D). Since nuclear export is effectively zero, we attribute the increase in total H3.3 over time solely to import and therefore the slope of total H3.3 over time corresponds to the import rate. Though the change in initial import rates between NC10 and NC13 are similar between the two histones (Fig EV1F), we observed a notable difference in their behavior in NC13. H3 nuclear accumulation plateaus ~5 minutes into NC13, whereas H3.3 nuclear accumulation merely slows (Fig 2C-D).” lines 110-117

For Figure 1E (now 2E), to address the difference between H3 and H3.3 free pools we have added the previously published values to the text and changed the phrasing from “less dramatic” to “less complete”. The sentence now reads:

“These changes in nuclear import and incorporation result in a less complete loss of the free nuclear H3.3 pool (~70% free in NC11 to ~30% in NC13) than previously seen for H3 (~55% free in NC11 to ~20% in NC13)” lines 117-120

The disappearance of H3.3 from mitotic chromosomes in Figure 2E is also not explained. If this phenomenon is functionally relevant, the authors should provide a mechanistic interpretation, or at the very least, discuss potential explanations in the text. In Figures 2F-H, the reasoning behind comparing the nuclear intensity of H3.3 to H3 in Hira mutants is unclear. To properly assess the role of HIRA in H3.3 chromatin accumulation, a more appropriate comparison would be between wild-type H3.3 and H3.3 levels in Hira knockdown embryos.

As explained in the text and depicted in Figure 3D (formerly 2D), the HIRA^{ssm} mutant is a point mutation that prevents observable H3.3 chromatin incorporation, but not nuclear import. This is what is depicted in Figure 3E (formerly 2E). The loss of H3.3 from mitotic chromatin is due to the inability to incorporate H3.3 into chromatin as expected for a HIRA mutant. We have edited the figure 3 legend to make this more clear. It now reads:

“Hira^{ssm} mutation nearly abolishes the observable H3.3 on mitotic chromatin (E).”

In Figure 3F (formerly 2F-H) we ask what happens to H3 chromatin incorporation when there is almost no incorporation of H3.3 due to the HIRA mutation. In this mutant there is so little H3.3 incorporation that we cannot quantify H3.3 levels on mitotic chromatin (see the new Figure 1B for the stage where chromatin levels are quantified). This experiment was done to test if H3.3^{ASVM} (expressed at the H3.3A locus) is incorporated into chromatin in embryos lacking the function of H3.3's canonical chaperone. We have edited the text to make this more clear:

“Since the chromatin incorporation of the H3/H3.3 chimeras appears to depend on their chaperone binding sites, we asked if impairing the canonical H3.3 chaperone, Hira, would affect the incorporation of H3.3^{ASVM} expressed from the H3.3A locus.”lines 159-161

Finally, the section discussing "H3.3 incorporation depends on cell cycle state, but not cell cycle duration" is unclear. The term "cell cycle state" is vague and should be explicitly defined. Does this refer to a specific phase of the cell cycle, changes in chromatin accessibility, or another regulatory mechanism?

The term cell cycle state is deliberately vague. We know that Chk1 regulates many aspects of cell cycle progression and cannot determine from our data which aspect(s) of cell cycle regulation by Chk1 are important for H3.3 incorporation. Our data indicate that it is not simply interphase duration as we originally hypothesized. We have expanded our discussion section to underscore some aspects of Chk1 regulation that we speculate may be responsible for the change in H3.3 behavior.

“Chk1 mutants decrease H3.3 incorporation even before the cell cycle is significantly slowed. Cell cycle slowing has been previously reported to regulate the incorporation of other histone variants in Drosophila¹⁵. However, our results indicate that cell cycle state and not duration per se, regulates H3.3 incorporation. In most cell types, the primary role of Chk1 is to stall the cell cycle to protect chromatin in response to DNA damage. Therefore, Chk1 activity directly or indirectly affects the chromatin state in a variety of ways. For example, Chk1 mutants lose the mitosis-specific phosphorylation of H3 earlier in anaphase than wildtype embryos⁷⁵. We speculate that Chk1's role in regulating origin firing may be particularly important in this context^{76,77}. Late replicating regions and heterochromatin first emerge during NC13, and Chk1 mutants proceed into mitosis before the chromatin is fully replicated in NC13^{22,23,25,72,74}. Since H3.3 is often associated with late-replicating heterochromatic regions, the decreased H3.3

incorporation in Chk1 mutants may be an indirect result of increased origin firing^{76,77}.” lines 295-307

Reviewer #3 (Evidence, reproducibility and clarity (Required)):

Summary:

Based on previous findings of the changing ratios of histone H3 to its variant H3.3, the authors test how H3.3 incorporation into chromatin is regulated for ZGA. They demonstrate here that H3 nuclear availability drops and replacement by H3.3 relies on chaperone binding, though not on its typical chaperone Hira. Furthermore, they show that nuclear-cytoplasmic (N/C) ratios can influence this histone exchange likely by influencing cell cycle state.

We thank the reviewer for their thoughtful comments. We note that our data ARE consistent with H3.3 incorporation depending on Hira through its chaperone binding site.

3. In the discussion section, can the authors speculate on how they think H3.3 ASVM is getting incorporated if not through Hira. Are there other known H3 variant chaperones, or can the core histone chaperone substitute?

We have expanded our discussion to include the the following:

“In the case of the chimeric histone proteins, the incorporation behavior was dependent on the chaperone binding site. For example, H3.3^{ASVM} import and incorporation were similar to H3 in control embryos and H3.3^{ASVM} was still incorporated in Hira^{ssm} mutants. This is consistent with the chaperone binding site determining the chromatin incorporation pathway and suggests that H3.3^{ASVM} likely interacts with H3 chaperones such as Caf1.” lines 288-293

Minor comments:

While the paper is well written, I found the figures very confusing and difficult to interpret. Comments here are meant to make it easier to interpret.

1. Fig 1 and most of the paper would benefit from a schematic of early embryo transitions labelled with time and stages of cell cycle to make interpreting data easier

This is an excellent suggestion! We have added a new figure (Figure 1) to explain both the biological system and the way that we measured many properties in this paper.

2. Fig 1- same green color is used for nuclear cycle 12 and for H3.3 making it confusing when reading graphs. Please check other figures where there is a similar use of color for two different things

We have changed the colors so that they are more distinct.

4. Line 130-133: can they also comment on the different between SVM and ASVM. It seems like SVM might be even worse than ASVM (Fig 2C). Is this related to chk1 phosphorylation?

We think that this is a property of the mixed chimeras since S31A is also imported less efficiently than H3.3 (though we cannot be sure without further experiments). We have added this explanation to the text:

“We speculate that chimeric histone proteins (H3.3^{S31A} and H3.3^{SVM}) are not as efficiently handled by the chaperone machinery as species that are normally found in the organism including H3.3^{ASVM} which is protein-identical to H3.” lines 151-153

5. Fig 2F-G: It is very difficult to compare between histones when they are on different graphs, please consider putting H3, H3.3 and H3.3ASVM in a hirasm background on the same graph.

We have done this in the new Figure 3F.

6. Fig 3- move G to become A and then have A and B.

We have restructured this figure to include the nuclear density map of control in response to a comment from Reviewer 1. Although not exactly what the reviewer has envisioned, we hope that this adds clarity to the figure.

7. The initial slope graphs in 4D, E, H and I are not easy to understand and would benefit from an explanation in the legend.

We have edited the legend of Figure 5D (formerly 4D) and S1F which now read:

“Initial slopes of nuclear import curves (change in total nuclear intensity over time for the first 5 timepoints) ...”

In addition we have updated the methods to include:

“Import rates were calculated by using a linear regression for the total nuclear intensity over time for the first 5 timepoints in the nuclear import curves.” lines 492-493, methods

4. Description of analyses that authors prefer not to carry out

Reviewer #1

OPTIONAL experimental suggestion: The experiments in Figure 4 and 5 are clever. One would expect that H3 levels might exhaust faster in embryos lacking all H3.2 histone genes

(Gunesdogan, 2010, PMID: 20814422), allowing a comparison testing the H3 availability > H3.3 incorporation portion of the hypothesis without manipulating the N/C ratio. This might also result in a more consistent system than slbp RNAi (below).

We thank the reviewer for the experimental suggestion. We also considered this experimental manipulation to decrease RD histone H3.2. We chose not to do this experiment because in the Gunesdogan paper they show that the zygotic HisC nulls have normal development until after NC14 (unlike the maternal SLBP-RNAi that we used) suggesting that maternal H3.2 supplies do not become limiting until after the stages under consideration in our paper. Maternal HisC-nulls are, of course, impossible to generate since histones are essential.

Figures:

While I appreciate the statistical summaries in tables, it is still helpful to display standard significance on the figures themselves.

We have added statistical comparisons in Figure 3 (formerly Figure 2). We do not feel that it is appropriate to directly compare the intensities of the H3-Dendra2 construct expressed from the pseudo-endogenous locus to the H3.3 and chimeric proteins expressed from the H3.3A locus as they were imaged using different settings. Although we plot H3 on the same graph as the other proteins to allow for ease of comparison of their trends over time it is not appropriate to directly compare their normalized intensities which including statistical tests would encourage. We have added a note to the legend of Figure 1 explaining this which reads:

“Note that statistical comparisons between the two Dendra2 constructs have not been done as they were expressed from different loci and imaged under different experimental settings.”

Fig 1:

B: Statistics required - caption suggests statistics are in Table S2, but why not put on graph? *Please see the explanation above for why we do not feel that it is appropriate to perform this comparison.*

C/D: Would be helpful if authors could plot H3/H3.3 on same graph because what we really need to compare is NC13 between H3/H3.3 (and statistics between these curves) *Please see the explanation above for why we do not feel that it is appropriate to perform this comparison. These curves can be directly compared within a construct and we can evaluate their trends over time, but the normalized values should not be directly compared in the way that would be encouraged by plotting the data as suggested.*

Fig 4:

B/C/F/G: The authors use a point size scale to represent the number of nuclei, but the graphs are so overlaid that it is not particularly useful. Is there a better way to display this dimension? *We chose to represent the data in this way so that the visual impact of each line is*

representative of the amount of data (number of nuclei in each bin) that underlies it. This helps to prevent sparsely populated outlier bins at the edges of the distribution from dominating the interpretation of the data. If the reviewer has a suggestion for a better way to visualize this information we would welcome their suggestion, but we cannot think of a better way at this time.

Reviewer #2

Moreover, given that H3.3 is associated with actively transcribed genes, an RNA-seq analysis of chimeric embryos should be included to assess transcriptional changes linked to H3.3 incorporation.

This is an excellent suggestion and will definitely be a future project for the lab. However, to do this experiment correctly we will need to generate untagged chimeric lines that will (hopefully) allow for the full replacement of H3.3 with the chimeric histones instead of a single copy among 4. This is beyond the scope of this paper.

Figure 2 presents additional concerns regarding data interpretation. The comparisons between H3.3 and H3.3S31A to H3 and H3.3SVM/ASVM lack statistical analysis, making it difficult to determine the significance of the observed differences.

As discussed above, it is not appropriate to directly compare H3 to H3.3 and the chimeras at the H3.3A locus since they are expressed from different promoters and imaged with different settings. The ANOVA comparisons between all of the constructs in the H3.3A locus can be found in the supplemental table. We have also added the statistical significance between each chimera and H3.3 within a cell cycle to the figure. Including the full set of comparisons for all genotypes and timepoints makes the figure nearly impossible to interpret, but they remain available in the supplemental table.

A broader concern is that the authors only test HIRA as a histone chaperone but do not consider alternative chaperones that could influence H3.3 deposition. Since multiple chaperone systems regulate histone incorporation, it would strengthen the conclusions if additional chaperones were tested.

Since HIRA^{ssm} reduced H3.3-Dendra2 incorporation to nearly undetectable levels (Figure 3E) we believe that it is the primary H3.3 incorporation pathway during this period of development. Therefore, we believe that removing HIRA function is a sufficient test of the dependence of H3.3^{ASVM} on the major H3.3 chaperone at this time. Although it would be interesting to fully map how all H3 and H3.3 chimera constructs respond to all histone chaperone pathways, we believe that this is beyond the scope of this manuscript.

Reviewer #3

3. Fig 1C,D might benefit more from being split up into 3 graphs by cell cycle with H3 and H3.3 plotted on the same graphs rather than the way it is now

We do not feel that it is appropriate to directly compare the intensities of the H3-Dendra2 construct expressed from the pseudo-endogenous locus to the H3.3 and chimeric proteins expressed from the H3.3A locus as they were imaged using different settings. These curves can be directly compared within a construct and we can evaluate their trends over time, but the normalized values should not be directly compared in the way that would be encouraged by plotting the data as suggested.

Dear Prof. Amodeo

Thank you for the submission of your revised manuscript to EMBO reports. We have now received the full set of referee reports that is copied below. Since my colleague Esther Schnapp is currently traveling, I have taken over the handling of your manuscript temporarily.

As you will see, all referees are very positive about the study and request only minor changes to clarify text and figures.

From the editorial side, there are also a few things that we need before we can proceed with the official acceptance of your study.

- Please reduce the number of keywords to 5.
- Please rename the Conflict of Interest section to Disclosure and Competing Interests Statement and place it after the Acknowledgments.
- Regarding the Author Contributions, we now use CRediT to specify the contributions of each author in the journal submission system. Therefore, please remove the Author Contributions from the manuscript file and make sure that the author contributions in our online manuscript tracking system are correct and up-to-date. The information you specified in the system will be automatically retrieved and typeset into the article. You can enter additional information in the free text box provided, if you wish.
- Please update the references to the alphabetical Harvard style. The abbreviation 'et al' should be used if there are more than 10 authors. DOIs should only be used for preprints and datasets that have not been published yet.
- McPherson et al, 2023 is a preprint. Please cite it like this in the text (preprint: McPherson et al, 2023) and add the tag [PREPRINT] to the end of the reference in the reference list. See also
- Funding information: Sondra and Charles Gilman Graduate Research Fellowship, currently listed in the Acknowledgments, also needs to be entered in our online manuscript tracking system.
- In the Author Checklist you chose "Not applicable" for most points in the section "Experimental study design and statistics" but nevertheless refer to Materials and Methods. Please reconcile this discrepancy, and in case you did use blinding or randomization, please add this information to the Methods section.
- Reagents and Tools Table: please remove the "Instructions" text from the file.
- Drosophila stocks used: you list them in the Methods section and in the Reagents and Tools table. I think the latter would be sufficient, as the information is otherwise redundant.
- You provide 10 EV tables in a Supplementary word file. You can either
(A) keep all tables in this Word file, but then add the nomenclature to "Appendix Table S#", call the entire file "Appendix" and add a table of content with page numbers. DATASET EV LEGENDS: 10 EV tables provided in one Word file; EV tables need to be uploaded as separate files
or
(B) keep the tables as EV tables, but then you need to upload each table individually as separate file.
- The movie legends need to be removed from the manuscript and each should be zipped up with its corresponding movie file (as a README.txt file) and then uploaded as a zip folder so that we have one folder per movie.
- Materials and methods should be Methods
- Please provide the exact p values in the legends of figures 3B, C; 4E, F; 6F, EV3 A, B; EV5 C
- Finally, EMBO Reports papers are accompanied online by
A) a short (1-2 sentences) summary of the findings and their significance,
B) 2-3 bullet points highlighting key results and
C) a schematic summary figure that provides a sketch of the major findings (not a data image).
Please provide the summary figure as a separate file in PNG or JPG format at a size of 550x300-600 pixels (width x height). Please note that the size is rather small and that text needs to be readable at the final size. Please send us this information along with the revised manuscript.

With kind regards,

Martina

=====

Referee #1:

I enjoyed this paper and I am satisfied with the response of the authors and recommend publication.

Referee #2:

I find the manuscript impeccably written. The authors have carefully considered how to display their data. Since the phenotypes can be subtle, the authors sometimes had to be creative in their analysis and presentation. The work opens up intriguing new links between Chk1, the cell cycle, and histone dynamics.

Tiny suggestions: I think a significance indicator is missing from Figure 6B (NC13). The protein "Zelda" is often capitalized.

Referee #3:

The authors have addressed my comments, and I believe the manuscript is now ready for publication.

Referee #1:

I enjoyed this paper and I am satisfied with the response of the authors and recommend publication.

Referee #2:

I find the manuscript impeccably written. The authors have carefully considered how to display their data. Since the phenotypes can be subtle, the authors sometimes had to be creative in their analysis and presentation. The work opens up intriguing new links between Chk1, the cell cycle, and histone dynamics.

Tiny suggestions: I think a significance indicator is missing from Figure 6B (NC13). The protein "Zelda" is often capitalized.

Referee #3:

The authors have addressed my comments, and I believe the manuscript is now ready for publication.

Rev_Com_number: N/a
New_manu_number: EMBOR-2025-61506V2
Corr_author: Amodeo

All editorial and formatting issues were resolved by the authors.

Prof. Amanda Amodeo
Dartmouth College
Biology
223 Class of 1978 Life Sciences Center
Dartmouth College
Hanover, NH 03755
United States

Dear Prof. Amodeo,

I am very pleased to accept your manuscript for publication in the next available issue of EMBO reports. Thank you for your contribution to our journal.
